# Controlling gene activation by enhancers through a drug-inducible topological insulator

**Taro Tsujimura[1,2]†\*, Osamu Takase[1,2], Masahiro Yoshikawa[1,2], Etsuko Sano[1,2], Matsuhiko Hayashi[3], Kazuto Hoshi[4,5], Tsuyoshi Takato[4,5], Atsushi Toyoda[6], Hideyuki Okano[2], Keiichi Hishikawa[1,2]\***

[1]Department of iPS Cell Research & Epigenetic Medicine, Keio University School of Medicine, Tokyo, Japan; [2]Department of Physiology, Keio University School of Medicine, Tokyo, Japan; [3]Apheresis and Dialysis Center, Keio University School of Medicine, Tokyo, Japan; [4]Division of Tissue Engineering, University of Tokyo Hospital, Tokyo, Japan; [5]Department of Oral and Maxillofacial Surgery, University of Tokyo Hospital, Tokyo, Japan; [6]Department of Genomics and Evolutionary Biology, National Institute of Genetics, Mishima, Japan

**Abstract** While regulation of gene-enhancer interaction is intensively studied, its application remains limited. Here, we reconstituted arrays of CTCF-binding sites and devised a synthetic topological insulator with tetO for chromatin-engineering (STITCH). By coupling STITCH with tetR linked to the KRAB domain to induce heterochromatin and disable the insulation, we developed a drug-inducible system to control gene activation by enhancers. In human induced pluripotent stem cells, STITCH inserted between *MYC* and the enhancer down-regulated *MYC*. Progressive mutagenesis of STITCH led to a preferential escalation of the gene-enhancer interaction, corroborating the strong insulation ability of STITCH. STITCH also altered epigenetic states around *MYC*. Time-course analysis by drug induction uncovered deposition and removal of H3K27me3 repressive marks follows and reflects, but does not precede and determine, the expression change. Finally, STITCH inserted near *NEUROG2* impaired the gene activation in differentiating neural progenitor cells. Thus, STITCH should be broadly useful for functional genetic studies.

**\*For correspondence:**
taro.tsujimura@keio.jp (TT);
hishikawa-tky@umin.ac.jp (KH)

**Present address:** †Institute for the Advanced Study of Human Biology (WPI-ASHBi), Kyoto University, Kyoto, Japan

## Introduction

Interaction of genes and enhancers is greatly affected by architectural proteins that bind to chromatin and organize folding of the genome (*Dekker et al., 2017*). Most notably, CTCF mediates loop formation of chromatin in association with a cohesin complex, which physically bundles two distant loci of the genomic DNA (*Parelho et al., 2008*; *Phillips-Cremins et al., 2013*; *Wendt et al., 2008*). The genome-wide contact maps of chromatin show that the CTCF-binding sites often demarcate boundaries of so-called contact domains or topologically associating domains (TADs), where chromatin association takes place more preferentially inside than outside (*Dixon et al., 2012*; *Phillips-Cremins et al., 2013*; *Rao et al., 2014*). The looping between two CTCF-binding sites is mostly established where they are in the converging orientations with each other (*de Wit et al., 2015*; *Guo et al., 2015*; *Rao et al., 2014*; *Vietri Rudan et al., 2015*). Loss of cohesin or CTCF resulted in disappearance of contact domains (*Gassler et al., 2017*; *Nora et al., 2017*; *Rao et al., 2017*; *Schwarzer et al., 2017*; *Wutz et al., 2017*). According to the extrusion model, the cohesin ring extrudes the chromatin fiber from a site of loading and pauses at a CTCF-binding site that is oriented towards the ring (*Fudenberg et al., 2016*; *Sanborn et al., 2015*). This model is widely accepted as the underlying mechanism for the formation of the loops and contact domains.

On the other hand, several studies have shown that the CTCF boundaries limit the action ranges of enhancers and thus restrict the enhancer targets to genes within the same contact domains as the enhancers reside in *Dowen et al. (2014)*; *Lupiáñez et al. (2015)*; *Symmons et al. (2014)*; *Tsujimura et al. (2015)*; *Tsujimura et al. (2018)*. These results are interpreted that CTCF demarcates contact domains, which then serve as entity to restrict or facilitate gene-enhancer interaction within themselves (*Schoenfelder and Fraser, 2019*). In the above studies, however, the gene-enhancer regulation was investigated primarily with respect to CTCF/cohesin and their binding sites in the genome, but not directly to the contact domains. Therefore, it remains elusive if contact domains per se have instructive roles in gene-enhancer interaction, or CTCF/cohesin directly regulates the interaction separately from creating contact domains.

Nonetheless, considering the apparent importance of CTCF, engineering the genome based on the CTCF function can add a new layer to the techniques of artificially controlling gene expression. The classical insulator element identified in the chicken *β-globin* locus (cHS4), which harbors a CTCF-binding site (*Bell et al., 1999*), has been utilized in heterologous systems (*Bessa et al., 2014*). However, the mechanistic investigation of these elements was limited. Therefore, the general utility of these elements as a tool was not very evident. In this respect, re-examining synthetic CTCF binding elements in light of the current understanding of chromatin regulation is desired to explore the utility of CTCF for genome engineering.

Also, a recent study showed that the SETDB1 repressive complex negatively regulates CTCF binding probably through heterochromatin formation involving KRAB zinc-finger proteins around the binding sites at the clustered protocadherin locus (*Jiang et al., 2017*). Currently, the generality of CTCF regulation by heterochromatin formation is unclear. Besides, it is not shown how such epigenetic change would affect the enhancer blocking activity of CTCF binding regions. Nonetheless, the possibility of artificially controlling CTCF binding is quite attractive in terms of genome engineering.

The *Tfap2c-Bmp7* locus in mice is partitioned into two contact domains by a region termed TZ (*Tsujimura et al., 2015*; *Tsujimura et al., 2018*). The TZ also limits target ranges of enhancers at the locus (*Tsujimura et al., 2015*). The TZ consists of two arrays of CTCF-binding sites in divergent orientations with each other. Serial mutagenesis has shown that this configuration underlies the strong ability of the TZ to block chromatin contacts (*Tsujimura et al., 2018*). Taking advantage of the well-characterized nature of the TZ, in this study, we developed a new system to control the interaction between a gene and an enhancer. We first reconstituted the CTCF-binding sites of the TZ as a short DNA cassette, which successfully functioned as an enhancer blocker. Further, we added a feature that enables epigenetically controlling the blocking activity of the cassette in a drug-inducible manner. Here we describe the system, demonstrate its utility to study gene regulation by enhancers, and discuss the future potential of the system.

## Results

### STITCH blocks the interaction of *MYC* with the enhancer when inserted in between

To newly develop an artificial genomic insulator cassette to switch on and off the gene-enhancer interaction, we reconstituted arrays of binding sites of CTCF derived from the TZ present at the mouse *Tfap2c-Bmp7* locus. The TZ consists of seven binding sites of CTCF: they are L1, L2, L3, L4, R1, R2, and R3, arrayed in this order from the *Tfap2c* side to the *Bmp7* side (*Figure 1A*; *Tsujimura et al., 2018*). L1-L4 are oriented towards *Tfap2c* and collectively referred to as L, while R1-R3 are towards *Bmp7* and referred to as R. The seven sites are constantly called as peaks of CTCF binding in different cell types by ChIP-seq (Chromatin immunoprecipitation followed by sequencing) with cross-linking. However, native-ChIP (nChIP) failed to detect CTCF binding at L1 and L4, suggesting the binding there is weak or indirect (*Tsujimura et al., 2018*). We extracted the 178 or 179 bp DNA sequences carrying the motif sequences for CTCF binding and concatenated them as a short DNA cassette. We embedded the core sequence of the tetracycline operator (tetO) at four different positions within the cassette. tetO is bound by the tetracycline repressor (tetR), but not in the presence of doxycycline (DOX), and thus allows recruitment of a linked effector protein to the cassette in a drug-dependent manner (*Gossen and Bujard, 1992*). We also put a puromycin-resistant gene (*PURO'*) sandwiched by two loxP sites for the sake of efficient targeting (*Figure 1A*,

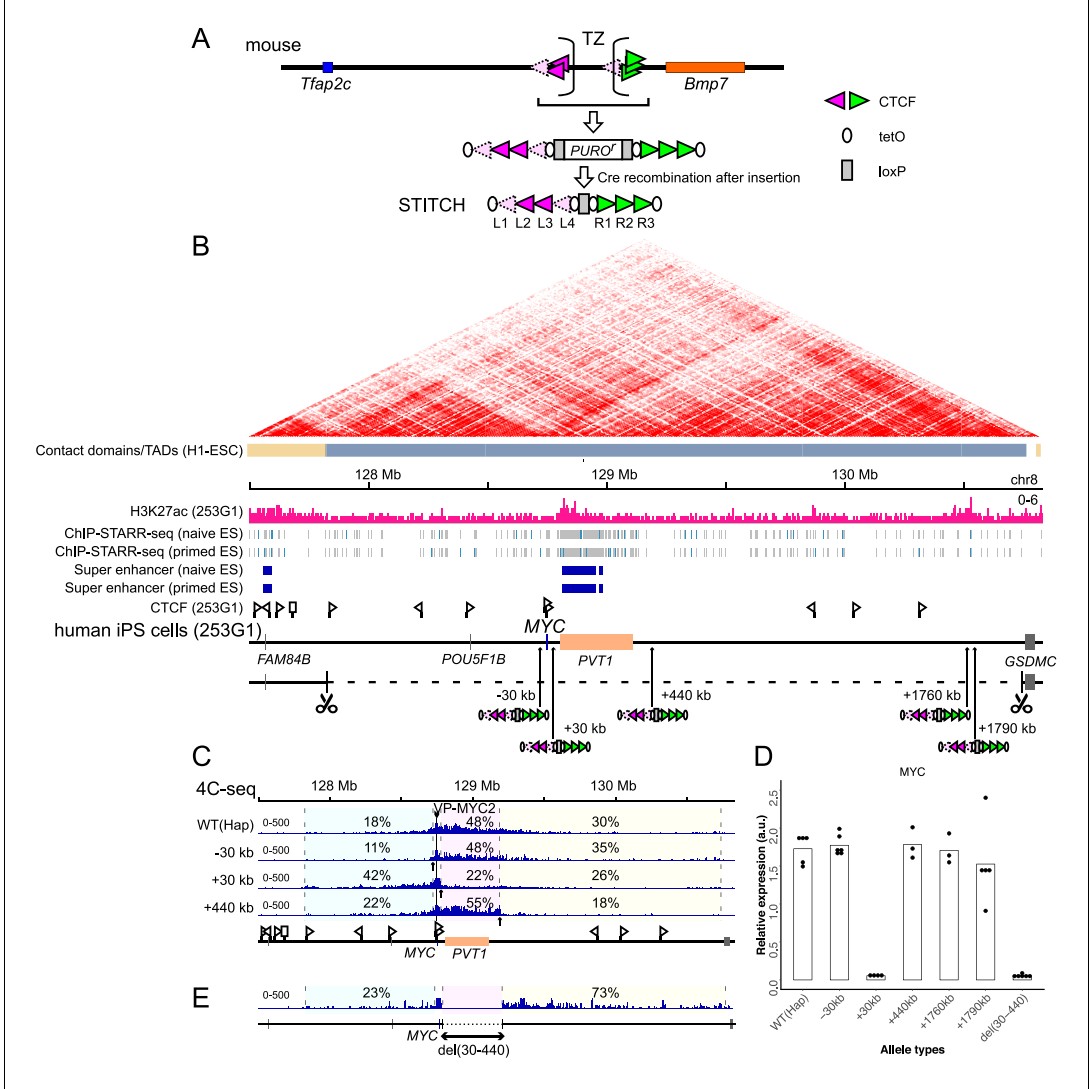

**Figure 1.** Serial insertion of STITCH around *MYC* localized the enhancer. (**A**) Design of STITCH and scheme of inserting the cassette. After recombination of the two loxP sites (rectangles), the puromycin resistant gene is removed. The orientations of the CTCF binding motifs are represented by the orientations and colors of the triangles. Note that binding of CTCF at L1 and L4 was detected by nChIP neither in the endogenous locus of the mouse genome nor at STITCH in the *MYC* locus, as represented by the paled color (see *Figure 1—figure supplement 1C*). The ovals represent tetO. The sequences of these elements are shown in *Supplementary file 1B*. (**B**) The H3K27ac profile and the insertion sites of STITCH around *MYC* in the human iPS cells. The Hi-C map and the contact domains in human ESCs are shown at the top (*Dixon et al., 2015*). The Hi-C contact map was generated with the 3D Genome Browser (http://3dgenome.org) (*Wang et al., 2018*). The ChIP-STARR-seq profiles and annotated super-enhancer regions in human naïve and primed ES cells (*Barakat et al., 2018*) are also depicted. The triangle flags indicate the positions and orientations of the CTCF binding sequences identified in this study. Note that the algorithm that we used could not determine the binding motif of one site represented by a rectangle flag. The 3 Mb region deleted from one of the two alleles to make 'Hap' is indicated by the dashed line, flanked by scissors that indicate the target sites of CRISPR/Cas9. The numbers in the insertion names indicate the distance from *MYC*. (**C**) The 4C-seq profiles from VP-MYC2 of the wild type (Hap) and STITCH-30kb, +30kb, and +440kb alleles. (**D**) Relative *MYC* expression levels normalized with *ACTB* expression in the different alleles. Each dot represents replicate clones (see Materials and methods for details). The bars represent their means. (**E**) The 4C-seq profile of del(30-440) from VP-MYC2. The numbers indicate the ratios of sequence reads mapped to given intervals within the locally haploid 3 Mb region around *MYC* except for the 10 kb region from the viewpoint fragment (**C, E**).

The online version of this article includes the following source data and figure supplement(s) for figure 1:

**Source data 1.** 4C-seq read counts in the given intervals.
**Figure supplement 1.** 4C-seq profiles of STITCH insertion clones.
**Figure supplement 2.** The scheme to delete the +(30-440)kb region.

*Supplementary file 1B*). We expected that the CTCF-binding sites of the cassette would recruit CTCF and function as a topological insulator and that the tetO/tetR system would enable epigenetically modifying the insulation activity. We named the cassette as <u>S</u>ynthetic <u>T</u>opological <u>I</u>nsulator with <u>T</u>etO for <u>Ch</u>romatin-engineering (STITCH) (*Figure 1A*).

*MYC* is highly expressed in human pluripotent stem cells (*Knoepfler, 2008*). As *MYC* expression is regulated by long-range enhancers in various cell types, we thought that *MYC* expression in the stem cells should also be dependent on long-range enhancers (*Bahr et al., 2018*; *Cho et al., 2018*; *Dave et al., 2017*; *Herranz et al., 2014*; *Hnisz et al., 2013*; *Lovén et al., 2013*; *Pulikkan et al., 2018*; *Shi et al., 2013*; *Sur et al., 2012*; *Uslu et al., 2014*; *Zhang et al., 2016*). Hence, we used the human induced pluripotent stem cell (iPSC) line 253G1, which was generated via retroviral transduction of *OCT4*, *KLF4*, and *SOX2* but without *MYC*, to test the functionality of STITCH (*Nakagawa et al., 2008*). A previous study called a large contact domain around *MYC* in human embryonic stem cells (ESCs) spanning almost 3 Mb (*Dixon et al., 2015*; *Figure 1B*). A super-enhancer region is annotated within the neighboring long non-coding RNA (lncRNA) gene *PVT1* in ESCs based on ChIP-seq for histone H3 lysine 27 acetylation (H3K27ac), the enhancer associated histone modification, and ChIP-STARR-seq (self-transcribing active regulatory region sequencing, after chromatin immunoprecipitation) (*Barakat et al., 2018*; *Figure 1B*). Similarly, a super-enhancer was annotated within the same region in the mouse ESCs (*Witte et al., 2015*). We also confirmed the broad deposition of H3K27ac around there in the human iPSC line (*Figure 1B*).

Since the diploidy would hamper the following genome editing procedures, we first deleted one allele of the 3 Mb region around *MYC* as described before (*Tsujimura et al., 2018*) to make the locus locally haploid, and termed the clone as 'Hap' (*Figure 1B*). Then we inserted STITCH into five different positions of the remaining allele of the locus: 'STITCH+30kb', 'STITCH+440kb', 'STITCH +1760kb' and 'STITCH+1790kb' have the STITCH insertions away from the *MYC* promoter for the indicated distances to the telomeric side of the q arm of the chromosome (the right side on the map, *Figure 1B*); 'STITCH-30kb', at the 30 kb upstream from *MYC* (the left side, *Figure 1B*). STITCH +30kb and STITCH+440kb flank the super-enhancer and *PVT1*. STITCH+1760kb and STITCH +1790kb flank a peak of H3K27ac (*Figure 1B*).

We first performed 4C-seq (Circular chromatin conformation capture assay followed by deep sequencing) from the *MYC* promoter as a viewpoint to see how STITCH impacts on the chromatin conformation. We designed two sets of primers around the *MYC* promoter as viewpoints of 4C-seq (VP-MYC1 and VP-MYC2, see *Figure 1—figure supplement 1A*). In the wild type allele of Hap, *MYC* mainly contacts with the *PVT1* region and around (*Figure 1C*, *Figure 1—figure supplement 1B*). In STITCH+30kb, STITCH+440kb, and STITCH-30kb, the contacts were blocked at the inserted positions of STITCH as expected (*Figure 1C*, *Figure 1—figure supplement 1B*). We then extracted RNA from the cells and measured the *MYC* expression levels with quantitative PCR (qPCR). We found that only STITCH+30kb strongly down-regulated the *MYC* expression, while the others did not (*Figure 1D*). These results suggest that the region between STITCH+30kb and STITCH+440kb (+(30-440)kb region) possesses the enhancer for the *MYC* expression. We made a deletion clone of the region, termed del(30-440) (*Figure 1E*, *Figure 1—figure supplement 2*). While the 4C contact profile of *MYC* extended further away from the deleted region (*Figure 1E*, *Figure 1—figure supplement 1B*), *MYC* was strongly down-regulated by the deletion (*Figure 1D*), showing that the region contains the responsible enhancer. nChIP-seq in STITCH+30kb confirmed that each of the binding sites of STITCH, except L1 and L4, was bound by CTCF as in the endogenous mouse genome, showing that these CTCF bindings are recapitulated regardless of the genomic context in human iPSCs (*Figure 1—figure supplement 1C*). Thus, STITCH recruits CTCF and blocks the gene-enhancer interaction when located in between as an insulator.

Of note, the nChIP-seq also identified endogenous sites directly bound by CTCF. In this study, we performed in total six nChIP-seq, including the two replicates from STITCH+30kb (*Figure 1—figure supplement 1C*) and the other following four that are two replicates from two different conditions (see Figure 5F, Figure 5—figure supplement 1). We collected peaks that are called at least in two out of the six experiments as reliable binding sites of CTCF for this study. Then we mapped the sites and orientations (*Figure 1B,C*). As indicated, *MYC* carries two CTCF-binding sites directed to the right side near the promoter region. These sites may account for the directional bias of the *MYC* contact towards the right side in WT(Hap) (*Figure 1C*). At the left side border of the large contact

domain of the locus, a CTCF-binding site oriented to the right side was detected. The contact of *MYC* in STITCH+30kb appears to extend up to this boundary (*Figure 1C*).

## Insulation and deletion of the enhancer resulted in similar transcriptome profiles

We employed RNA-seq to understand how the insulation (STITCH+30kb) and deletion (del(30-440)) of the enhancer affect the transcriptome of the cells through the down-regulation of *MYC* (*Figure 2*, *Figure 2—figure supplement 1*). Of note, deletion and duplication of the whole *PVT1* genic region, as well as the knockdown experiment via RNAi, has suggested a role for the *PVT1* lncRNA in *MYC* activation (*Tseng et al., 2014*). However, clearly distinguishing if it is the transcribed RNA or the associated enhancer regions that regulate *MYC* could be complicated (*Bassett et al., 2014*). Indeed, a recent study shows that inhibition of the *PVT1* transcription does not impact on *MYC* expression in a cancer cell line (*Cho et al., 2018*). This study instead showed that the *PVT1* promoter modulates *MYC* expression as a competitor for enhancer activity, which may indicate that the transcribed RNA is a byproduct. Comparing the two mutations in this study might also clarify the role of *PVT1* as lncRNA or a *cis*-regulator.

We prepared libraries from three replicates (as for Hap the parental clone and two derived sub-clones; as for STITCH+30kb and del(30-440), three different clones isolated upon the Cre recombination, respectively) for each configuration. Consistently with the qPCR assay (*Figure 1D*), strong down-regulation of *MYC* was confirmed in both STITCH+30kb and del(30-440) (*Figure 2A*, *Figure 2—figure supplement 1A*). *PVT1* expression was not altered in STITCH+30kb (*Figure 2A*, *Figure 2—figure supplement 1A*). We did not observe other detectable expression changes around the *MYC* locus in either STITCH+30kb or del(30-440) (*Figure 2A*). We computed the log2 fold changes of the transcriptome with the shrinking algorithm implemented in DESeq2 (*Love et al., 2014*; *Figure 2B–D*) and applied the results to Gene Set Enrichment Analysis (GSEA) (*Subramanian et al., 2005*) against the hallmark gene sets (HALLMARK50) in the Molecular Signatures Database (MSigDB) (*Liberzon et al., 2015*). Strikingly, the down-regulated genes in both STITCH+30kb and del(30-440) are highly enriched with known *MYC* target genes, showing that the down-regulation of *MYC* by both mutations is large enough to affect its target transcriptome. The other enriched categories are also well shared by the two mutations, highlighting the similarity in the transcriptomic change.

With threshold of log2 fold change <0.5 and p-adjusted <0.05, STITCH+30kb and del(30-440) had 218 and 68 down-regulated genes, and 494 and 137 up-regulated genes, respectively (*Figure 2B,C*). Among those, large fractions (36 and 92 genes, for down- and up-regulation, respectively) were common between the two alleles (*Figure 2—figure supplement 1B,C*). Importantly, the comparison between STITCH+30kb and del(30-440) called much less number (64) of differentially expressed genes (*Figure 2D*). Moreover, del(30-440) exhibited a rather milder effect on the transcriptome than STITCH+30kb (*Figure 2B and C*, *Figure 2—figure supplement 1B,C*). These data suggest that *PVT1* has little impact on the transcriptome as *trans*-acting lncRNA if any.

It should be noted that the insulation showed a stronger effect than the deletion of the enhancer. STITCH+440kb did not show almost any effect on the *MYC* expression level (*Figure 1C,D*), indicating that the region beyond +440 kb does not contribute to the activation of *MYC* when the locus is intact. However, upon the deletion of the enhancer, contacts of *MYC* greatly extended beyond +440 kb (*Figure 1C,E*, and *Figure 1—figure supplement 1B*). Therefore it might be possible that *MYC* can be slightly activated by regions with some enhancer activity located beyond +440 kb that do not associate with *MYC* in the normal context, which may account for the milder outcome of del (30-440) than STITCH+30kb.

The results of the GSEA indicate the possible functional roles of *MYC*. The categories enriched in down-regulated genes include those in which *MYC* has been implicated by previous studies such as cell cycle progression (*Bretones et al., 2015*), unfolded protein response (*Shajahan-Haq et al., 2014*), TCA cycle (*Anderson et al., 2018*), mTORC1 signaling (*Liu et al., 2017*; *Yue et al., 2017*), and cholesterol synthesis (*Hofmann et al., 2015*; *Figure 2E*). Also, gene ontology (GO) enrichment analysis shows that the commonly down-regulated genes in STITCH+30kb and del(30-440) are highly enriched with genes encoding regulators of ribosome assembly, which are known target groups of *MYC* in various systems (*Hofmann et al., 2015*; *Uslu et al., 2014*; *van Riggelen et al., 2010*; *Zeller et al., 2006*), as well as those involved in cholesterol metabolism similarly as above (*Figure 2—*

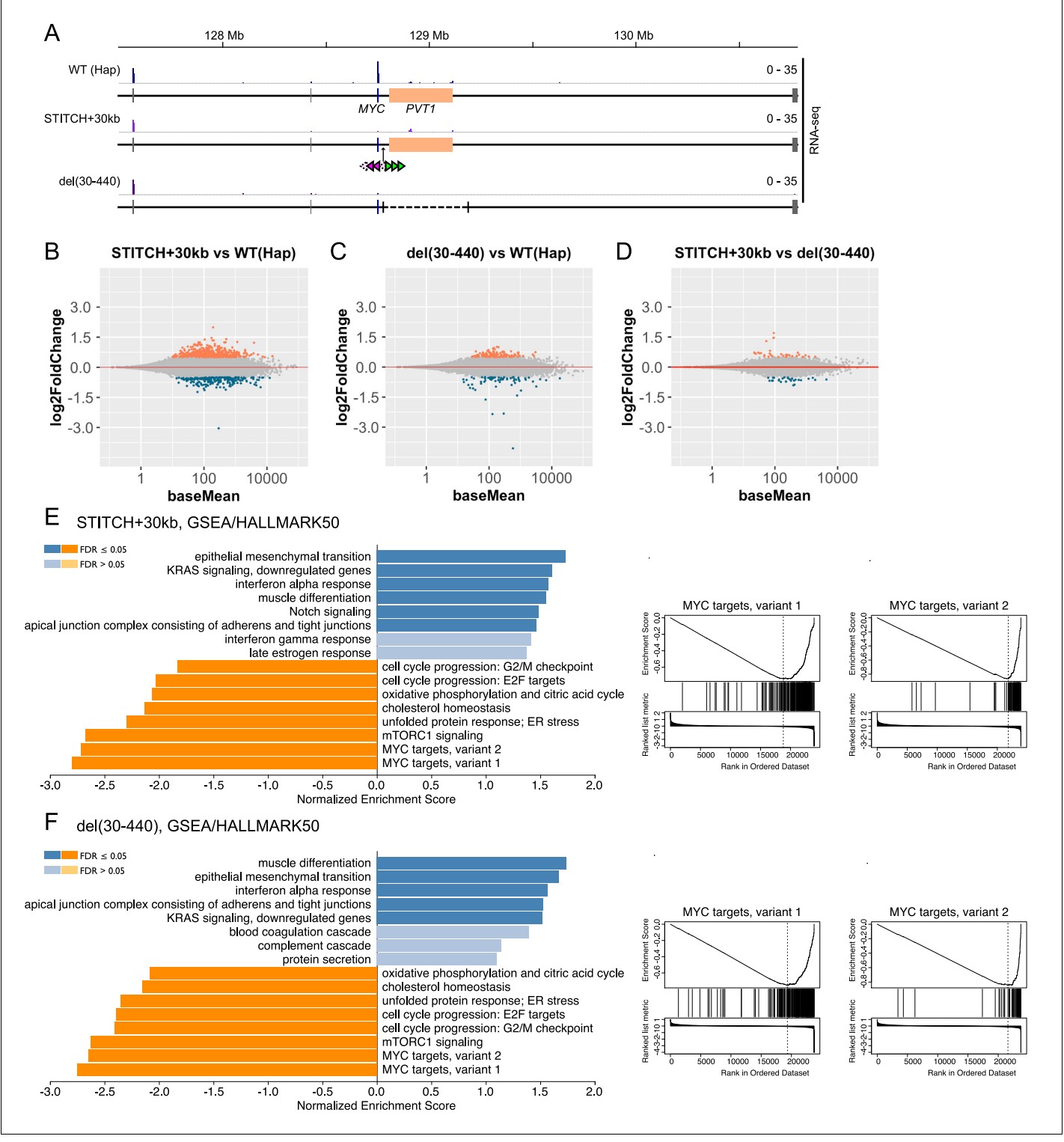

**Figure 2.** Transcriptome analysis of Hap, STITCH+30kb and del(30-440). (**A**) Tracks of RNA-seq from Hap, STITCH+30kb, and del(30-440) around the *MYC* locus. (**B–D**) MA-plots of RNA-seq to compare STITCH+30kb vs. Hap (**B**), del(30-440) vs. Hap (**C**) and STITCH+30kb vs. del(30-440) (**D**). Differentially expressed genes (adjusted p-values<0.05, log2 fold changes > 0.5) are marked by colors (orange for up-regulated genes and dark blue for down-regulated ones). (**E and F**) Enriched categories among HALLMARK50 (*Liberzon et al., 2015*) by GSEA (*Liao et al., 2019*) (left) and the enrichment plots against the categories MYC targets variant 1 and 2 (right) in STITCH+30kb (**E**) and del(30-440) (**F**).

The online version of this article includes the following source data and figure supplement(s) for figure 2:

**Source data 1.** RNA-seq read counts and the results of the DESeq2 analyses.

*Figure 2 continued on next page*

*Figure 2 continued*

**Figure supplement 1.** Transcriptome analysis of Hap, STITCH+30kb, and del(30-440).

*figure supplement 1B*). Our results strengthen the link between *MYC* and these biological processes.

## Titrating blocking activity of STITCH by serial mutations of the CTCF-binding sites

The divergent configuration of CTCF-binding sites establishes boundaries of contact domains in the genome, while those directed to only one side are also capable of partitioning the chromatin into two domains, namely as loop and exclusion domains (*Guo et al., 2015*; *Sanborn et al., 2015*). In fact, deletion and inversion of either of the two CTCF binding arrays, L or R, impaired, but still kept, the blocking activity of the TZ at the endogenous locus in the mouse ESCs (*Tsujimura et al., 2018*). However, it has been unclear how these differences in the CTCF configuration would impinge on gene activation by enhancers. In the present study, to understand how important the arrangement of the CTCF-binding sites is for STITCH to block the chromatin contact and the gene activation, we made deletion of each CTCF array, L (delL) and R (delR), inversion of R (invR), deletion of the middle five binding sites from L2 to R2 (del(L2-R2)), and deletion of the six sites but for R3 (del(L1-R2)) in STITCH+30kb. We also obtained deletion and inversion of the whole of STITCH (del(L1-R3) and inv (L1-R3), respectively) (*Figure 3A*, *Figure 3—figure supplement 1*).

The *MYC* expression levels in delL and delR were slightly increased from the original STITCH allele (*Figure 3A*). invR also increased it but to a lesser extent (*Figure 3A*). del(L2-R2) and del(L1-R2) up-regulated the expression even more, but much less than the wild type Hap allele (*Figure 3A*). The *MYC* expression in del(L1-R3) was comparable to that of Hap, showing that the gene activation could be safely recovered upon removal of the CTCF-binding sites (*Figure 3A*). inv(L1-R3) exhibited the same degree of repression as STITCH+30kb, showing that STITCH blocks enhancer activation regardless of the orientation of the insertion as a whole (*Figure 3A*).

We next examined the 4C contact profiles of the *MYC* promoter in these mutation alleles (*Figure 3B–J*, *Figure 3—figure supplements 2–4*). The contact frequency with the +(30-440)kb enhancer region was changed depending on the configuration (*Figure 3B*, *Figure 3—figure supplement 2A*). The original STITCH and inv(L1-R3) most strongly reduced the contacts. invR showed slightly more of contacts there, but not as much as delL and delR. These results indicate firstly that the divergent configuration is the strongest way to block contacts, and secondly that the more CTCF binds there, the more strongly it blocks contacts (*Figure 3B*, *Figure 3—figure supplement 2A*). This observation is very consistent with the previous study about the endogenous TZ in the mouse ESCs (*Tsujimura et al., 2018*). del(L2-R2) and del(L1-R2) further recovered the contact frequency (*Figure 3B*). Thus, the gene expression level and the contact frequency are well correlated. The Spearman's rank correlation coefficients were 0.92 and 0.90 for VP-MYC1 (*Figure 3—figure supplement 2B*) and VP-MYC2 (*Figure 3I*), respectively. We noted that the expression level fits with a power-law model with the contact frequency of the +(30-440)kb region with a scaling exponent of 4.1–4.3 (*Figure 3I*, *Figure 3—figure supplement 2B*). It is particularly notable that even the del(L1-R2) efficiently blocks the gene activation only with the remaining one CTCF-binding site R3, but not much the contact.

To investigate into how the inserted STITCH impacts on chromatin conformation of the locus, we next performed 4C-seq from viewpoints flanking the insertion site (VP-STITCH-left and VP-STITCH-right) (*Figure 3—figure supplement 3*) in WT(Hap), STITCH+30kb, delR, and delL. The different compositions of the CTCF-binding sites in these mutants may affect the folding directionality of the flanking sites locally, as shown in previous studies (*de Wit et al., 2015*; *Guo et al., 2015*; *Tsujimura et al., 2018*). The flanking regions of the mouse TZ exhibit diverging directionality of chromatin folding (*Tsujimura et al., 2018*). This divergence is a typical hallmark feature of boundaries of contact domains (*Dixon et al., 2012*). We, therefore, calculated folding directionalities at each viewpoint (VP-MYC1, -MYC2, -STITCH-left, and -STITCH-right) as difference of read counts between the left and right intervals for given distances (1 Mb, 500 kb, or 100 kb) normalized by the sum of them (*Figure 3—figure supplement 3B*).

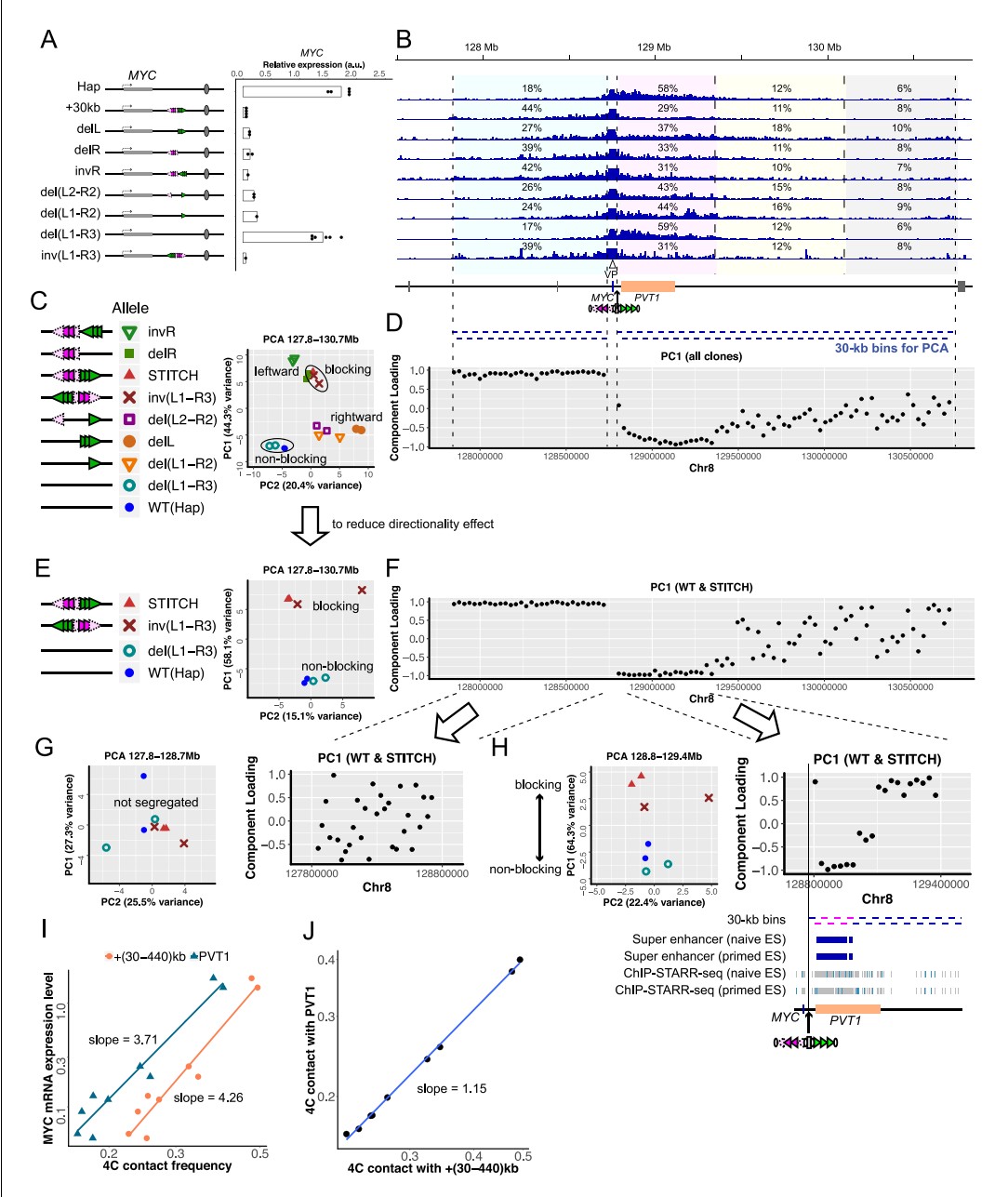

**Figure 3.** *MYC* expression and 4C-seq profiles in serially mutated STITCH alleles. (**A**) Configurations of CTCF-binding sites of mutated STITCH alleles and a plot showing their *MYC* expression levels. Each dot represents replicate clones (see Materials and methods for details). Note that the data of Hap and STITCH+30kb are the same as *Figure 1D*. Bars indicate means of the replicates. (**B**) 4C-seq profiles from VP-MYC2 in the different alleles. The numbers indicate the ratios of the mapped reads to the indicated regions within the 3 Mb region, except for the 10 kb region from the viewpoint. Below the coordinate map, blue bars indicate bins (each 30 kb) for PCA in (**C–H**) and *Figure 3—figure supplement 4*. (**C**) PCA plot of all the clones using the normalized counts in all the bins of the whole locus. (**D**) Component loadings of PC1 in the PCA in (**C**) are plotted along the coordinate for each bin. (**E, F**) The PCA plot only with the non-blocking alleles, the original STITCH, and inv(L1-R3) using the bins of the whole locus (**E**), and the corresponding PC1 component loading plots (**F**). (**G, H**) The PCA plot with the same subset clones as (**E**) (left), and the corresponding PC1 component-loading plots (right) using the re-normalized counts in the bins of the left 900 kb region (**G**) or the right 600 kb region (**H**). Below the component-loading plot in (**H**), tracks of the super-enhancers and ChIP-STARR-seq plots reported in *Barakat et al. (2018)* are depicted along with the 30 kb bins of the right 600 kb region. The six bins with the lowest values of component loadings in (**H**) are depicted with pink. (**I**) A log-log plot of the *MYC* expression levels against the 4C contact frequencies of VP-MYC2 in the +(30-440)kb region (orange) and the *PVT1* region (dark blue) for each clone. Note the difference between the two slopes. (**J**) A log-log plot of the 4C contact frequencies of VP-MYC2 in the *PVT1* region against the +(30-440)kb region.

*Figure 3 continued on next page*

*Figure 3 continued*

The online version of this article includes the following source data, source code and figure supplement(s) for figure 3:

**Source code 1.** Source Code File.
**Source code 2.** Source Code File_4CMYCcount.txt.
**Source code 3.** Source Code File_4CMYCcolor.txt.
**Source code 4.** Source Code File_4CMYCshape.txt.
**Source data 1.** 4C-seq read counts in the given intervals.
**Figure supplement 1.** CRISPR editing to make partial deletion and inversion alleles of STITCH.
**Figure supplement 2.** 4C-seq profiles of STITCH mutants.
**Figure supplement 3.** The directionality of chromatin folding around STITCH.
**Figure supplement 4.** The 4C-seq PCA plot of STITCH+30kb and the mutant clones without the non-blocking alleles (Hap and del(L1-R3)) (left) and the component loadings of PC1 (right) for the whole locus.

The Hap allele without the STITCH insertion exhibits overall rightward directionality from VP-MYC1/2 till VP-STITCH-left/right (*Figure 3—figure supplement 3B*). This tendency might be associated with the presence of the two CTCF-binding sites directed to the right side near the *MYC* promoter (*Figure 1*, *Figure 3—figure supplement 3A*). The insertion of STITCH introduced a skewed change of the directionality across the insertion site. The rightward directionality at VP-STITCH-right was even more enhanced, while those at VP-MYC1/2 and VP-STITCH-left were decreased to neutral (*Figure 3—figure supplement 3B*). In delR, the directionality at VP-STITCH-right became less prominent than the intact STITCH allele, while the directionality at both VP-MYC1/2 and VP-STITCH-left was again neutral (*Figure 3—figure supplement 3B*). By contrast, in delL, the rightward directionality was kept or slightly enhanced at both VP-STITCH-left and -right, while the directionality at VP-MYC1/2 was marginally reduced from the wild type allele (*Figure 3—figure supplement 3B*). These results suggest that the array L mainly orients the folding directionality at VP-MYC1/2 and VP-STITCH-left relatively towards the left side, while the array R enhances the rightward directionality at VP-STITCH-right. These relative transition patterns of the directionality across the insertion site are consistent with the case of the endogenous TZ (*Tsujimura et al., 2018*). However, it should be noted that the absolute divergence of folding directionality was not very evident around STITCH. Notably, the delL allele keeps the overall rightward directionality of chromatin folding across the region (*Figure 3—figure supplement 3B*). These results suggest that neither the diverging configuration of CTCF-binding sites nor the diverging directionality of chromatin folding is a prerequisite for enhancer blocking.

We note that VP-STITCH-left and VP-STITCH-right appear to have enhanced contacts with the left- and the right-side border of the large contact domains, respectively, which might represent the formation of loops by STITCH (*Figure 3—figure supplement 3A*). However, these contacts are not very striking compared to other recognizable contacting regions for both viewpoints (*Figure 3—figure supplement 3A*). Therefore, without deleting these regions, it is hard to specify loops, if any, that might be engaged in the STITCH functionality in this study. Also, more comprehensive analysis methods such as 5C (Chromatin Conformation Capture Carbon Copy) (*Dostie et al., 2006*) or Hi-C (*Lieberman-Aiden et al., 2009*) are required to fully describe the locus-wide conformational change induced by STITCH.

## Preferential association with the enhancer over non-enhancer regions upon CTCF removal

To understand more quantitatively and unbiasedly how the various configurations of the CTCF-binding sites at STITCH reshape the contact pattern of *MYC* along the locus, we performed the principal component analysis (PCA) for the 4C contact frequencies of 30 kb bins within a given region (*Figure 3B–H*, *Figure 3—figure supplement 4*), as inspired by its application in the Hi-C analysis to find the compartment domains (*Lieberman-Aiden et al., 2009*). We first analyzed the frequencies within the whole *MYC* locus for all of the alleles above (*Figure 3C,D*). The PCA plot well segregated the non-blocking alleles (Hap and del(L1-R3)) from the other blocking ones, especially the original STITCH, inv(L1-R3) along the PC1 axis (*Figure 3C*). To understand which bins of the locus contribute to this segregation, we plotted the component loadings for each bin along the genomic coordinate (*Figure 3D*). Component loadings are calculated as the product of the eigenvector and the square

root of the eigenvalue of the component. They correspond to the correlative coefficients of the original values of the bins and the component values. Therefore, component loadings indicate how much the values of each bin are reflected by the component. The component loadings of PC1 show that the segregation is mostly explained by lower and higher contact frequencies in the left side region, and higher and lower frequencies in the 570 kb region from the +30 kb site to the right side, of the non-blocking and the blocking alleles, respectively (*Figure 3D*).

We note that the different alleles are also arranged on the PCA plot according to the orientations of the CTCF binding motifs (leftward vs. rightward in *Figure 3C*). Both blocking effects of the mere presence of CTCF and directionality bias due to the orientations of the CTCF motifs seem to account for the segregation. To uncouple the two different effects, we performed PCA against subsets of the alleles. We first removed from the analysis the non-blocking alleles, Hap and del(L1-R3), to reduce the simple blocking effects and to enhance the directionality effect (*Figure 3—figure supplement 4*). Then, the alleles with leftward motifs were placed at the top, and the rest were at the bottom along PC1 on the plot (*Figure 3—figure supplement 4*). The component-loading plot indicates that the leftward alleles are more associated with the left side regions (*Figure 3—figure supplement 4*). These patterns are consistent with the above analysis showing that the array L reduces the rightward directionality of VP-MYC1/2 more than the array R (*Figure 3—figure supplement 3B*).

Next, to reduce the directionality effect, we used only the Hap/del(L1-R3), the original STITCH, and inv(L1-R3) clones for PCA. The PCA plot showed segregation between the non-blocking and blocking alleles along the PC1 axis (*Figure 3E*). The component-loading plot shows a clear split between the left 900 kb region and the right-side region at the STITCH insertion site (*Figure 3F*). The former region associates more with the blocking alleles, and the latter associates with the non-blocking alleles (*Figure 3E,F*).

To investigate if the left- or right-side regions contain sub-regions that specifically change contact patterns with *MYC* depending on the presence of STITCH, we then performed PCA for each of the left 900 kb region and the right 600 kb region with the subset clones (*Figure 3G,H*). The PCA plot for the left side did not show apparent segregation according to the CTCF composition (*Figure 3G*). By contrast, PCA for the right 600 kb region showed segregation between the blocking and non-blocking alleles (*Figure 3H*). These results indicate that the right 600 kb region contains bins that characteristically alter contact tendency with *MYC* depending on the presence of STITCH, while the left 900 kb region does not.

The pattern of the PC1 component loadings for the right side PCA was notable (*Figure 3H*). The association with the *PVT1* region, especially with the super-enhancer region (*Barakat et al., 2018*), accounts for the lower PC1 values of the Hap/del(L1-R3) clones, while that with the other remaining non-active regions accounts for STITCH/inv(L1-R3) (*Figure 3H*). These results suggest that *MYC* has preferential contacts with the super-enhancer/*PVT1* region more than with the other non-active regions in the absence of the CTCF insulation. We found that the power-law scaling of the *MYC* expression with the contact frequency with the *PVT1* region has a scaling exponent of 3.6–3.7, which is slightly less than with the +(30-440)kb region (*Figure 3I*, *Figure 3—figure supplement 2B*). Consistently, the contact with *PVT1* scales with that with the +(30-440)kb region, with an exponent factor 1.14–1.15, which is slightly higher than the linear correlation (*Figure 3J*, *Figure 3—figure supplement 2C*). Thus, titration of STITCH insulation revealed that the contact of *MYC* with the super-enhancer/*PVT1* region is enhanced more than the other non-active regions when the insulation is absent. In other words, the presence of CTCF insulation effectively impairs the gene-enhancer contact more than the contacts with neutral regions. A similar observation was also reported by a previous study (*Hou et al., 2008*). We think this kind of selective disruption of the gene-enhancer interaction may, at least in part, account for the discrepancy between the relatively small changes of the overall contact frequency and the drastic reduction of gene expression by the CTCF insulators here and in other genomic contexts.

## Epigenetic states of *MYC* well correlate with the gene activation by the enhancer

We next investigated how the STITCH insulation of the enhancer impinges on the epigenetic modifications of histones around *MYC* (*Figure 4*). Active transcription is associated with H3K4me3 at gene promoters, while repressed genes are often marked by H3K27me3. In the wild type allele, *MYC* is exclusively marked by H3K4me3, but not by H3K27me3. Upon the STITCH insulation, the H3K4me3

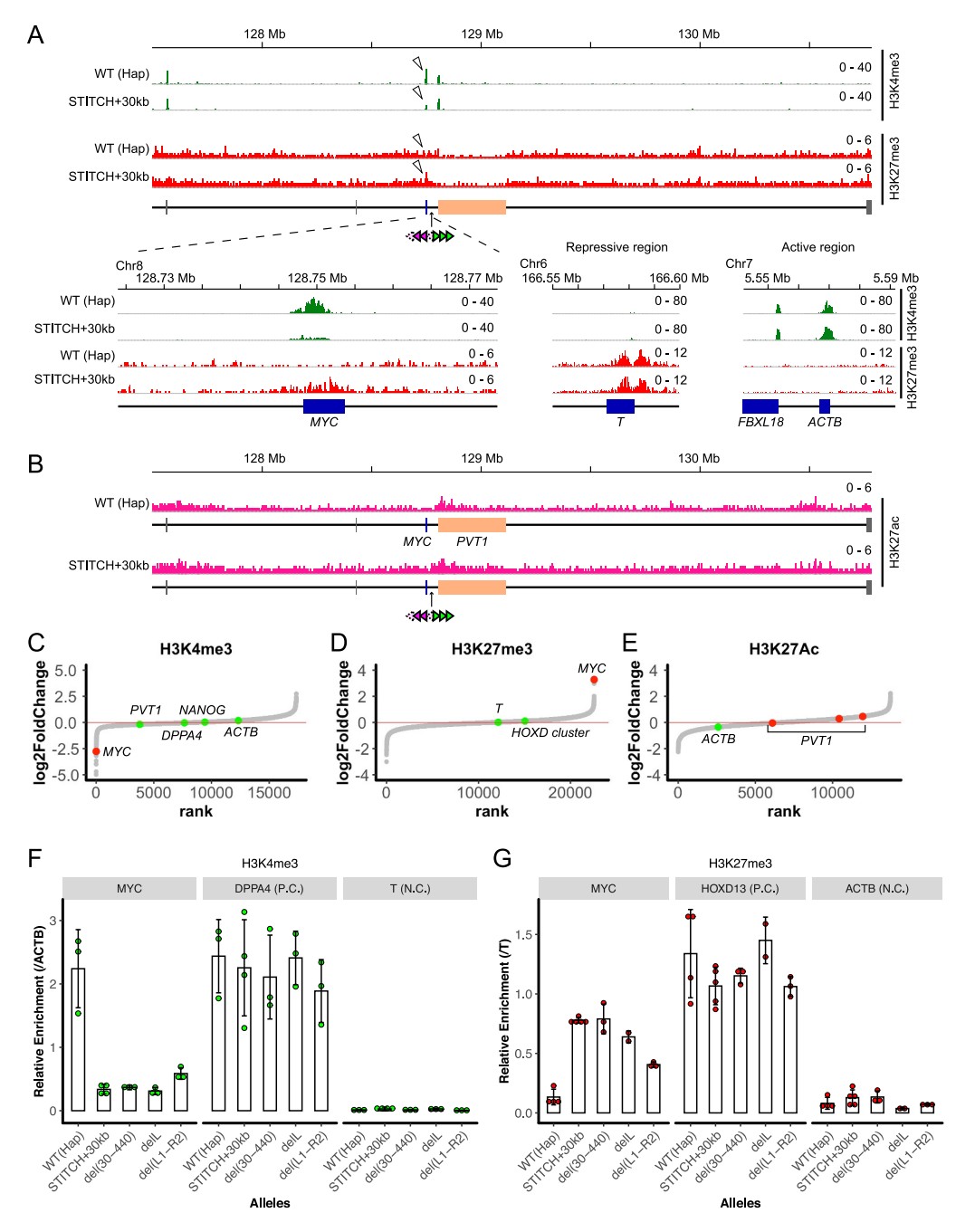

**Figure 4.** Epigenetic profile around *MYC* with and without STITCH. (**A**) nChIP-seq for H3K4me3 (green) and H3K27me3 (red) in the wild type (Hap) and STITCH+30kb clones. The magnified view around *MYC* is shown below, together with the typical repressive (*T*) and the active (*ACTB*) regions. (**B**) nChIP-seq for H3K27ac in Hap and STITCH+30kb. (**C–E**) The peaks of H3K4me3 (**C**), H3K27me3 (**D**), and H3K27ac (**E**) are ordered according to the normalized log2 fold changes in STITCH+30kb. The H3K4me3 and H3K27me3 peaks at *MYC* are depicted with red, and peaks at other representative genes are depicted with green, in C and D, respectively. Similarly, H3K27ac peaks within the *PVT1* genic region are depicted with red in E. (**F and G**) nChIP-qPCR for H3K4me3 (**F**) and H3K27me3 (**G**) in Hap, STITCH+30kb and the indicated mutant alleles of STITCH. The enrichment at *MYC* was normalized with those at *ACTB* (**F**) and *T* (**G**). We also quantified the relative enrichment at *DPPA4* and *T* for H3K4me3 (**F**), and *HOXD13* and *ACTB* for H3K27me3 (**G**), as positive and negative controls, respectively. The dots represent data from replicate experiments. The bars and the error bars indicate their means and the standard deviations (SD), respectively.

The online version of this article includes the following source data and figure supplement(s) for figure 4:

**Source data 1.** nChIP-seq read counts in the peaks for H3K4me3, H3K27me3, and H3K27ac.

*Figure 4 continued on next page*

*Figure 4 continued*

**Figure supplement 1.** ChIP-seq profiling of Hap and STITCH+30kb alleles.

deposition remained, but was markedly decreased. Instead, H3K27me3 was enriched. By contrast, the neighboring *PVT1* gene was strongly marked by H3K4me3 at the promoter in both conditions (*Figure 4A*, *Figure 4—figure supplement 1A,B*). Some typically active (*ACTB*, *NANOG*, *DPPA4*) and repressed (*T*, *HOXD13*) genes were constantly marked by either H3K4me3 or H3K27me3, respectively (*Figure 4A*, *Figure 4—figure supplement 1C*). Also, the H3K27ac mark around the super-enhancer region was similarly observed in both alleles (*Figure 4B*, *Figure 4—figure supplement 1A,B*). Among the peaks that were called in at least two out of the total four experiments (two from Hap and the other two from STITCH+30kb), the peaks at *MYC* were ranked as one of the top peaks exhibiting the largest fold change for both H3K4me3 (*Figure 4C*) and H3K27me3 (*Figure 4D*), while the H3K27ac peaks around *PVT1* did not change much (*Figure 4E*). These results show that the epigenetic change only occurred at *MYC* upon isolation from the enhancer by STITCH. We performed nChIP-qPCR to quantify the H3K4me4 and H3K27me3 levels at *MYC* in the alleles with the STITCH mutations (*Figure 4F,G*). We normalized the enrichment by that at *ACTB* and *T* (*Figure 4A*) to better compare different experiments for H3K4me3 and H3K27me3, respectively (*Figure 4F,G*). We found that *MYC* in the mutant alleles were epigenetically intermediate between the active and repressive states (*Figure 4F,G*). These results show that the histone marks around *MYC* vary depending on the association levels with the enhancer or the gene expression level.

## Induction of a heterochromatic state by tetR-KRAB impairs the STITCH insulation

The KRAB domain can induce heterochromatin formation around the tetO when linked to tetR (tetR-KRAB) and recruited there (*Deuschle et al., 1995*; *Groner et al., 2010*; *Sripathy et al., 2006*). If this leads to impairment of CTCF bindings as implicated in a previous study (*Jiang et al., 2017*), it would be possible to control the insulation ability of STITCH by DOX (*Figure 5A*). To test this, we integrated a transgene consisting of tetR-KRAB followed by DNA encoding the 2A peptide and the puromycin resistant gene (2A-PURO$^r$) with *piggyBac* transposition into the genome of a STITCH+30-kb clone, and established several cell lines that stably express it (*Figure 5B*). The expression levels of the transgene varied much among them (*Figure 5—figure supplement 1A*). Nonetheless, in all the cell lines tested, *MYC* expression was repressed in the presence of DOX but became activated after the removal of DOX (*Figure 5C*). Titration of the DOX concentration showed that 1 ng/ml is enough to achieve STITCH insulation in the tested clones with different expression levels of the transgene (*Figure 5D*, *Figure 5—figure supplement 1D*).

We performed nChIP-seq for H3K9me3, a mark representing the heterochromatin state, and for CTCF. When DOX was present, no H3K9me3 peak appeared around the inserted STITCH (*Figure 5E*, *Figure 5—figure supplement 1E–G*); instead, CTCF was strongly bound there (*Figure 5F*, *Figure 5—figure supplement 1E–I*). Accordingly, STITCH kept blocking the contacts of *MYC* towards *PVT1* (*Figure 5G*, *Figure 5—figure supplement 1M*). In the absence of DOX, however, H3K9me3 became highly enriched around STITCH (*Figure 5E*, *Figure 5—figure supplement 1E–G*). Concomitantly, the CTCF binding was strongly reduced, and the contact of *MYC* well extended to the enhancer region (*Figure 5F,G*, *Figure 5—figure supplement 1E–I,M*). We calculated the normalized fold changes of the read counts of the CTCF nChIP-seq mapped to each peak throughout the genome. Then, the arrays L and R of STITCH were the most significantly altered peaks by the removal of DOX (*Figure 5—figure supplement 1F*).

By contrast, induction of tetR linked to 3xFLAG with HA tag followed by 2A-PURO$^r$ neither affected CTCF binding at STITCH nor activated *MYC* in the STITCH+30kb clone (*Figure 5—figure supplement 1B,J*), showing that the KRAB domain is required to expel CTCF binding. We also confirmed that the STITCH before the Cre/loxP recombination harboring the *PURO$^r$* cassette, which should be bound by some transcription factors around the promoter for the expression, recruits CTCF and blocks *MYC* activation (*Figure 5—figure supplement 1A,K,L*), further arguing that binding of transcription factors does not impair CTCF binding. Also, integration of tetR-KRAB into a del (30-440) clone, which keeps two tetO sites at the +30 kb position, did not up-regulate *MYC* in the

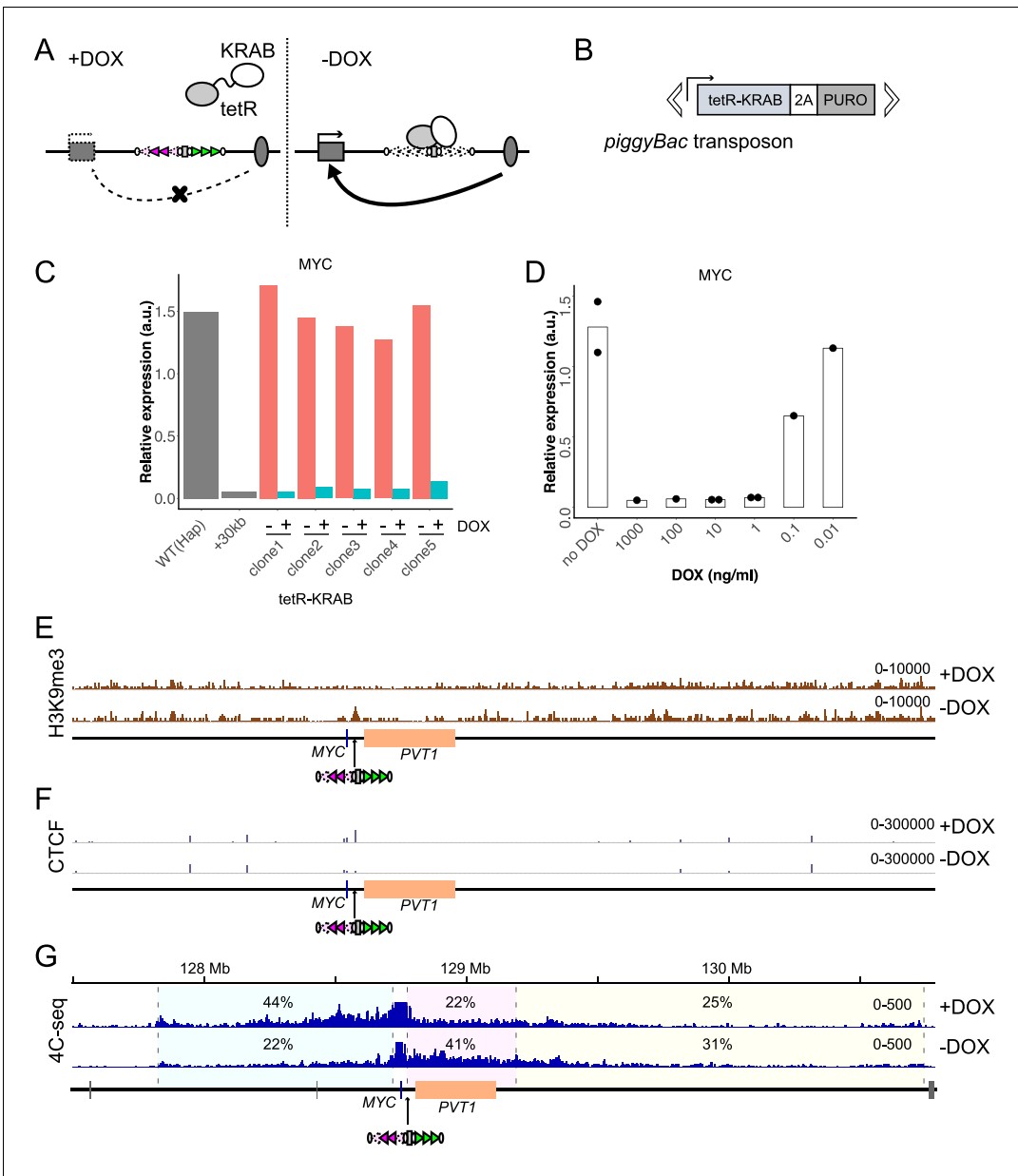

**Figure 5.** Drug-inducible control of STITCH insulation with tetR-KRAB. (**A**) DOX dependent binding to and dissociation from STITCH of tetR-KRAB. (**B**) The *piggyBac* transposon with the tetR-KRAB transgene followed by a sequence encoding 2A peptide and puromycin resistant gene. (**C**) The relative expression levels of *MYC* normalized to *ACTB* in five independent clones of STITCH/KRAB with and without DOX were compared to the expression levels of the ancestral Hap and STITCH+30kb clones from which the STITCH/KRAB clones were derived. (**D**) The *MYC* expression level in the clone 1 of STITCH/KRAB with different concentrations of DOX. The dots represent data from replicate experiments, and the bars indicate the means. (**E, F**) nChIP-seq tracks for H3K9me3 (**E**) and CTCF (**F**) of the clone one with and without DOX. The reads were mapped to a synthetic genomic DNA sequence around the *MYC* locus carrying the STITCH insert. (**G**) The 4C-seq tracks with and without DOX from VP-MYC2. The numbers indicate the ratios of sequence reads mapped to given intervals within the locally haploid 3 Mb region except for the 10 kb region from the viewpoint fragment.

The online version of this article includes the following source data and figure supplement(s) for figure 5:

**Source data 1.** 4C-seq read counts in the given intervals, and CTCF nChIP-seq read counts in the peaks.
**Figure supplement 1.** Heterochromatin induction by tetR-KRAB at STITCH.

absence of DOX (*Figure 5—figure supplement 1C*). These results show that the re-association with the enhancer upon KRAB-dependent displacement of CTCF led to the *MYC* activation by tetR-KRAB in the absence of DOX. Thus, the STITCH/KRAB system functions as a drug-inducible topological insulator to control gene activation by enhancers.

We next followed temporal changes of the system upon the addition and removal of DOX (*Figure 6*). The nChIP-qPCR for H3K9me3, the 4C-seq assays, and gene expression assays show that 16–24 hr, but not 8 hr, are sufficient to almost completely switch the STITCH insulation and *MYC* expression upon both removal and addition of DOX (*Figure 6A–E*). We tested how the switching of *MYC* expression would affect the cell proliferation and found that the addition of DOX (i.e., repression of *MYC*) for five days resulted in about 40% reduction of proliferated cells (*Figure 5—figure supplement 1N*).

## The epigenetic state of *MYC* follows and reflects the gene expression level

The H3K4me3 and H3K27me3 histone marks correlate well to the gene expression level (*Figure 4*, *Figure 4—figure supplement 1*). The rapid control of STITCH insulation with KRAB offers us an opportunity to investigate if the epigenetic changes precede the gene expression changes or not. Therefore, we also profiled the H3K4me3 and H3K27me3 levels around *MYC* at different time points up to 48 hr after the inductions. Interestingly, while the H3K4me3 mark returned to the levels expected from the gene expression levels within 24 hr after both removal and addition of DOX (*Figure 6F*), the H3K27me3 did not (*Figure 6G*). This result suggests that the change of the repressive histone mark follows, but does not precede, the gene expression change.

To test the hypothesis and confirm the reproducibility, we again sampled cells at time points of 24 and 72 hr after the addition/removal of DOX as well as cells that were kept either with or without DOX for more than one passage as the controls (*Figure 7A–F*). First, we confirmed that the *MYC* expression was up- and down-regulated within one day after removal and addition of DOX to the levels of the controls, respectively (*Figure 7A,B*). Then we performed nChIP-qPCR for both histone marks. Consistently to above, the deposition of H3K27me3 was significantly higher and lower in 24 hr than 72 hr and the controls after removal and addition of DOX, respectively (*Figure 7C,D*). By contrast, we did not see such significant differences for H3K4me3, suggesting that the active mark is more rapidly turned over than the repressive mark (*Figure 7E,F*).

These results suggest that the H3K27me3 mark per se only reflects, but does not determine, the gene expression level. To test this, we treated the cells with EPZ-6438 (EPZ), an inhibitor of Enhancer of zeste homolog 2 (EZH2), an enzymatic subunit of Polycomb Repressive Complex 2 (PRC2), which catalyzes methylation of H3K27 (*Knutson et al., 2013*). The addition of the inhibitor at 200 nM for two days was enough to mostly diminish the H3K27me3 mark at *MYC* (*Figure 7G*). This reduction of H3K27me3 did not result in significantly higher enrichment of the active H3K4me3 mark (*Figure 7H*). We compared the *MYC* expression levels in Hap, STITCH+30kb, and the mutant alleles of STITCH treated with EPZ or DMSO for three days (*Figure 7I*). The difference between the two treatments was not significant in any of the alleles. We next treated the STITCH/KRAB cells with EPZ or DMSO for two days, then removed DOX, and compared the *MYC* expression at different time points up to 24 hr after removal of DOX. The expression profiles showed no significant difference between the two, suggesting that the H3K27me3 mark does not affect the gene activation by the enhancer (*Figure 7J*).

## Blocking *NEUROG2* activation in differentiating neural progenitor cells with STITCH

We next tested the applicability of the STITCH/KRAB system to a different locus in a different cell-type. *NEUROG2* is a proneural gene expressed in neural progenitor cells (NPCs) (*Bertrand et al., 2002*). In the mouse embryonic brain, a stretch of the tissue-specific peaks of H3K27ac is present over the neighboring gene *Alpk1* (*ENCODE Project Consortium, 2012*), suggesting that these are the neural enhancers for *Neurog2* (*Figure 8—figure supplement 1*). NPCs can be efficiently derived from the human pluripotent stem cells by the dual SMAD inhibition (*Chambers et al., 2009*). A reported data shows that the differentiated NPCs with this method also exhibit prominent H3K27ac

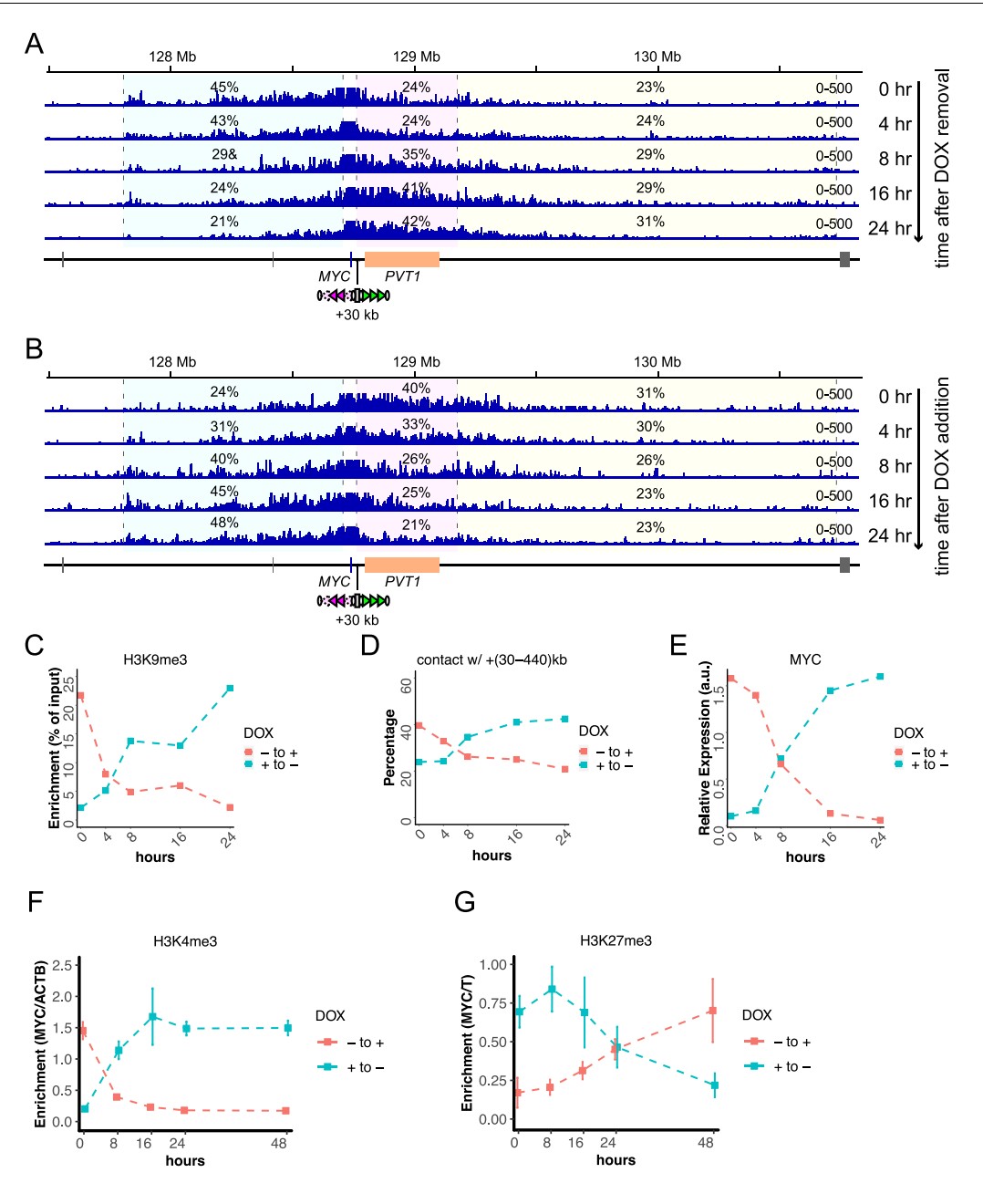

**Figure 6.** Temporal changes of STITCH insulation upon removal and addition of DOX. (**A**, **B**) The 4C-seq profiles in 0, 4, 8, 16, and 24 hr after removal (**A**) and addition (**B**) of DOX. The numbers indicate the ratios of sequence reads mapped to given intervals within the locally haploid 3 Mb region except for the 10 kb region from the viewpoint fragment. (**C–E**) Temporal changes of nChIP-qPCR for H3K9me3 at STITCH (**C**), 4C contact frequency with +(30–440)kb region from VP-MYC2 (**D**), the relative *MYC* expression level normalized to *ACTB* (**E**). (**F**, **G**) Temporal changes of relative enrichment of H3K4me3 at *MYC* normalized with that at *ACTB* (**F**), and relative enrichment of H3K27me3 at *MYC* normalized with that at *T* (**G**), up to 48 hr after removal and addition of DOX. We did not perform replicate experiments in (**A–E**). The nChIP-qPCR for H3K4me3 and H3K27me3 were performed for three replicate samples. The means and SDs are represented in the plots (**F**, **G**).

The online version of this article includes the following source data for figure 6:

**Source data 1.** 4C-seq read counts in the given intervals, and the results of nChIP-qPCR for H3K4me3 and H3K27me3.

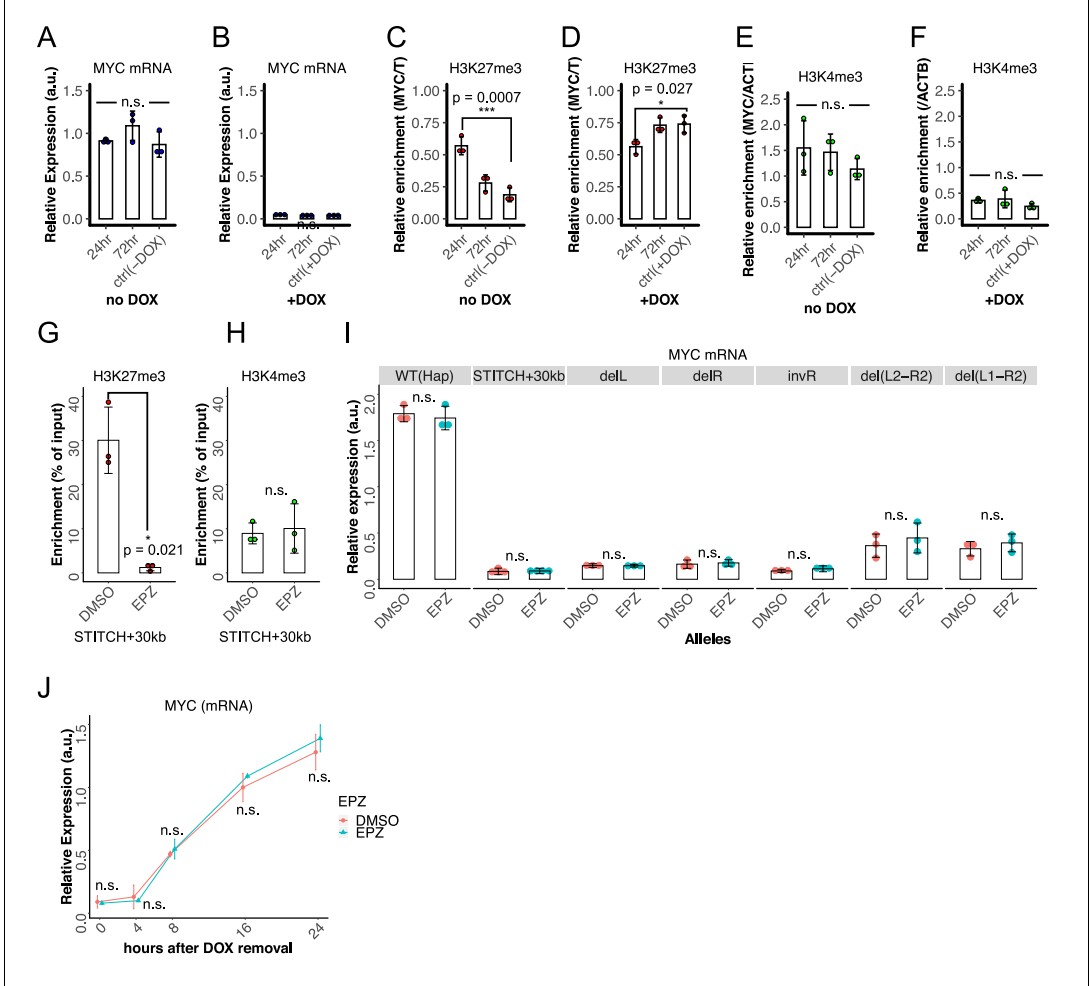

**Figure 7.** Delayed turnover of H3K27me3 enrichment after the gene expression change. (A–F) Relative *MYC* expression levels normalized to *ACTB* (A and B), relative H3K27me3 enrichment at *MYC* normalized to the enrichment at *T* (C and D) and relative H3K4me3 level at *MYC* normalized to that at *ACTB* (E and F) were measured at 24 hr (1 day) and 72 hr (3 days) after removal (A, C, E) or addition (B, D, F) of DOX in the STITCH/KRAB. The controls are the cells kept without (A, C, E) or with (B, D, F) DOX without switching for a few passages. The dots represent data from replicate experiments, the bars indicate their means, and the error bars indicate the SDs. *, *** and n.s. indicate p<0.05, p<0.001 and p>0.05, respectively, by one-way ANOVA. The p-values with Tukey's multiple-comparison post hoc test are indicated. (G, H) Enrichment of H3K27me3 (G) and H3K4me3 (H) at *MYC* after two days treatment with EPZ or DMSO in STITCH+30kb. The dots represent replicates, the bars indicate their means, and the error bars indicate the SDs. * and n.s. indicate p<0.05 and >0.05, respectively, by two-sided Welch's two-sample t-test. (I) Relative *MYC* expression levels in the Hap, STITCH+30kb, and the mutants of STITCH after three-days treatment of EPZ or DMSO. The dots represent replicates, and the bars indicate their means. (J) Temporal changes of relative *MYC* expression levels after DOX removal in the STITCH/KRAB. Before DOX was removed, cells were exposed to EPZ or DMSO for two days. Means and SDs of three replicate experiments were plotted. (I, J) n.s. indicates p>0.05, by one-sided Welch's two-sample t-test, in which the alternative hypothesis was that the mean of EPZ was greater than DMSO.

The online version of this article includes the following source data for figure 7:

**Source data 1.** *MYC* expression levels upon removal of DOX with DMSO or EPZ.

marks over *ALPK1* (*Xie et al., 2013*; *Figure 8A*), suggesting that these enhancers activate *NEUROG2* in vitro.

We inserted STITCH into the 65 kb downstream of *NEUROG2* near *ALPK1* in the iPS cells (the Hap clone), removed the *PURO*[r] cassette with Cre, and then integrated the tetR-KRAB-2A-PURO[r] with the *piggyBac* transposon. We term the resultant cells as NEUROG2/KRAB (*Figure 8A,B*). Here, STITCH was inserted only into one allele with the other one remaining intact. Also, after the *piggyBac* transposition, we did not clone single colonies, but just expanded the survived cells as a bulk for several passages in the presence of puromycin. Of note, the *MYC* expression levels in this cell population did not change by the absence and presence of DOX (*Figure 8—figure supplement 2A*),

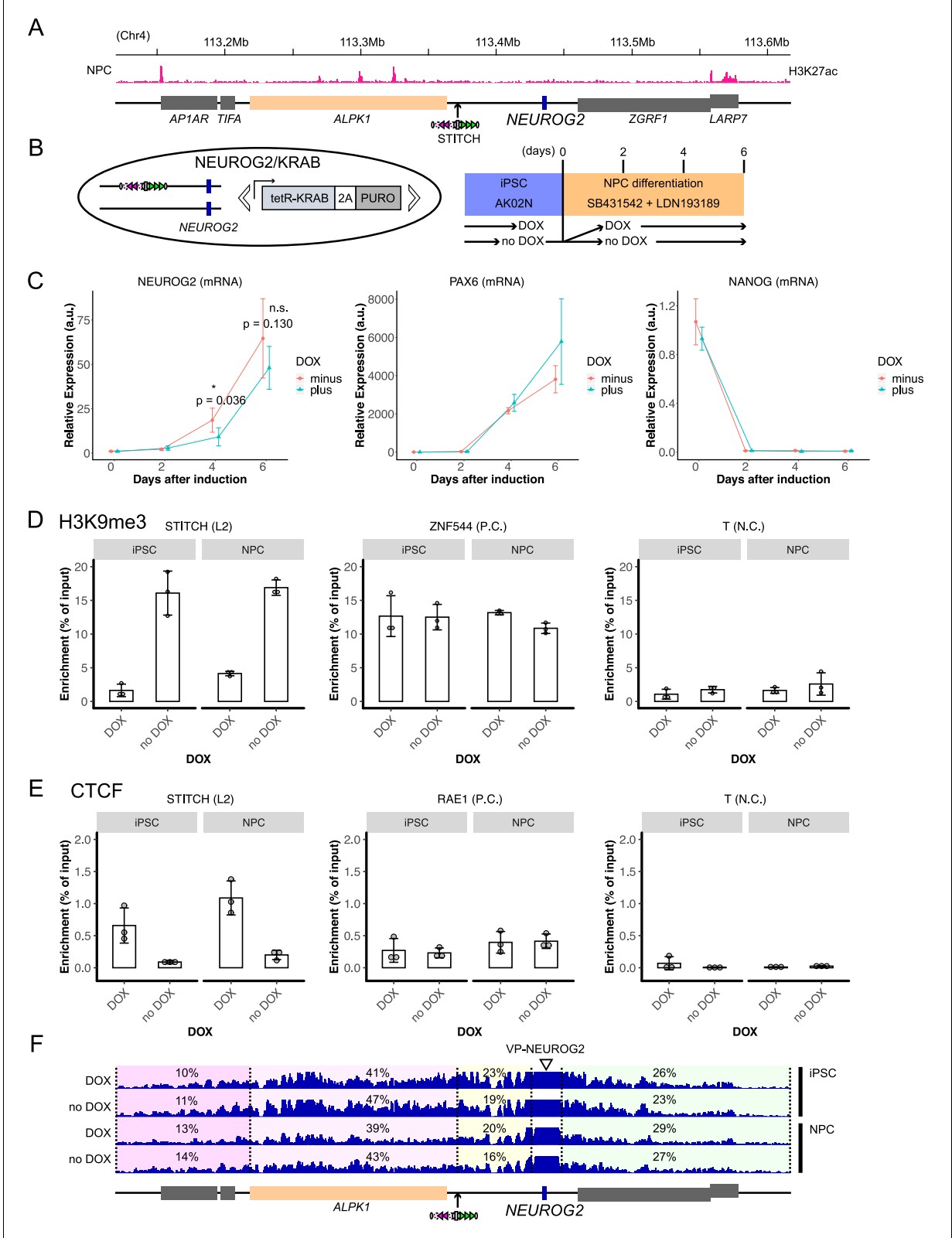

**Figure 8.** Testing STITCH/KRAB at the *NEUROG2* locus in the neural progenitor cells. (**A**) STITCH was inserted into the 65 kb downstream of *NEUROG2* near *ALPK1*. There seem active enhancers over the *ALPK1* genic region, as indicated by the previously reported H3K27ac profile in the NPCs (***Xie et al., 2013***). (**B**) The schematic illustration of the NEUROG2/KRAB cells (left) and the differentiation experiments (right). (**C**) The relative expression levels of *NEUROG2*, *PAX6*, and *NANOG* normalized by the *ACTB* expression levels during the differentiation process. Day 0 indicates iPSCs with or

*Figure 8 continued on next page*

*Figure 8 continued*

without DOX collected just before the start of the neural induction. The numbers of replicates were three for day 0 and four for days 2, 4, and 6. The means and the SDs of the replicate experiments are represented. The indicated p-values are obtained by one-sided Welch's two-sample t-test, where the alternative hypothesis was that the mean of DOX minus was greater than DOX plus. * and n.s. indicate p<0.05 and>0.05, respectively. (**D and E**) nChIP-qPCR for H3K9me3 (**D**) and CTCF (**E**) enrichment at STITCH (the L2 motif region) in iPSCs and NPCs (day6) with and without DOX. We also quantified the enrichment at *ZNF544* (**D**) and *RAE1* (**E**) regions as the positive controls for H3K9me3 and CTCF, respectively. We used the region around *T* as the negative control for both assays. (**F**) 4C-seq from VP-NEUROG2 in iPSCs and NPCs with and without DOX. The numbers indicate the ratios of the mapped reads in the given intervals.

The online version of this article includes the following source data, source code and figure supplement(s) for figure 8:

**Source code 1.** Source Code File_4CNGN2color.txt.
**Source code 2.** Source Code File_4CNGN2count.txt.
**Source code 3.** Source Code File_4CNGN2shape.txt.
**Source data 1.** Relative gene expression levels of *NEUROG2*, *PAX6*, *NANOG*, and *tetR* in differentiating NPCs, and 4C-seq read counts in the given intervals.
**Figure supplement 1.** The UCSC genome browser view of the mouse genome around *Neurog2* (mm9).
**Figure supplement 2.** Characterization of the NEUROG2/KRAB cells.
**Figure supplement 2—source code 1.** Source Code File.

confirming again that tetR-KRAB controls *MYC* expression only through the STITCH+30kb insertion (***Figure 5***).

We split the NEUROG2/KRAB cells derived from a single dish equivalently to different dishes, and then either did or did not add DOX upon the start of the differentiation into NPCs. The neural differentiation was achieved by the dual SMAD inhibition (***Chambers et al., 2009***) with SB-431542, an inhibitor for the SMAD2/3 pathway (***Inman et al., 2002***), and LDN-193189, an inhibitor for the SMAD1/5/8 pathway (***Cuny et al., 2008***; ***Figure 8B***). We then compared the expression levels of *NEUROG2*, as well as *PAX6* (NPCs marker) (***Chambers et al., 2009***) and *NANOG* (iPSCs marker) on days 2, 4 and 6 (***Figure 8C***). The induction diminished the *NANOG* expression already on day2 (***Figure 8C***). *PAX6* was strongly activated from day 4 (***Figure 8C***), showing that the cells were efficiently differentiated. *NEUROG2* was also activated from day 4 (***Figure 8C***). We tested if DOX treatment would decrease *NEUROG2* expression and found that *NEUROG2* was significantly less expressed in the cells with DOX than those without on day 4 (***Figure 8C***). On day 6, the tendency that the DOX treated cells express less *NEUROG2* was kept, though the difference was milder than day 4 and not statistically significant (***Figure 8C***). We realized that the expression level of tetR-KRAB was progressively decreased during the differentiation (***Figure 8—figure supplement 2B***), suggesting a part of the cells in culture might have experienced silencing of the transgene possibly due to the complete alteration of the epigenomic state. This silencing effect might be a reason why the difference between DOX plus and minus became smaller on day 6 (***Figure 8C***).

We compared the heterochromatin formation and CTCF binding at STITCH in the iPSCs and NPCs on day 6 between with and without DOX. In both cell types, tetR-KRAB induced H3K9me3 and expelled CTCF binding in the absence, but not in the presence of DOX (***Figure 8D,E***).

We next performed 4C-seq from a viewpoint at the *NEUROG2* promoter (VP-NEUROG2). Though the intact allele seems to mask the difference between the conditions a lot, contacts of *NEUROG2* with the *ALPK1* region beyond the STITCH insertion were constantly reduced by the addition of DOX in both iPSCs and NPCs (***Figure 8F***). PCA against the 4C-seq data segregated NPCs and iPSCs along PC1, and DOX plus and minus along PC2 (***Figure 8—figure supplement 2C***). Notably, the PC2 component loading plot unbiasedly exhibited the changing point exactly at the STITCH insertion site: the segregation between DOX plus and minus well correlates with contacts with the right- and the left-side regions from the insertion site, respectively (***Figure 8—figure supplement 2C***). We further performed 4C-seq using a viewpoint designed at the right edge of the inserted STITCH cassette (VP-R3) in the NPCs (***Figure 8—figure supplement 2D–G***). The addition of DOX strongly extended the contacts to further distances, as indicated by the ratio of reads between 100 and 200 kb distance region against those immediately within 100 kb region (***Figure 8—figure supplement 2E,F***). This change of chromatin conformation should reflect the extrusion-mediated contacts of the CTCF-binding sites at STITCH in the presence, but not the absence, of DOX (***Haarhuis et al., 2017***). Of note, the leftward extension of the contact indicates that the 4C-seq captures the effect of the

leftward CTCF array L1-L4 in the very close vicinity of VP-R3 (*Figure 8—figure supplement 2D*). Therefore, it was not surprising that the directionality of the chromatin folding of VP-R3 was not drastically biased towards the right side (*Figure 8—figure supplement 2E,G*). Overall, these results are consistent to the above observation that STITCH blocks the chromatin contacts of *NEUROG2*. Thus, the STITCH/KRAB system can be used in different loci in different cell types, strengthening its generality and robustness as a tool.

## Discussion

STITCH blocks the interaction of genes and enhancers when inserted in between as an insulator element (*Figure 9A,B*). Further combining this with the DOX control of tetR-KRAB achieved drug-inducible switching of the insulation (*Figure 9C*). Thus, the system adds a new layer to the toolkits for manipulating gene expression. Here, we first discuss the mechanism of the STITCH system and then the applicability of the system as a tool.

### Mechanism of the STITCH insulation and its control by heterochromatin induction

Though CTCF binding to L1 and L4 was not confirmed by the nChIP, the other five binding sites at STITCH were directly bound by CTCF (*Figure 1—figure supplement 1C* and *Figure 5—figure supplement 1G*). The delL and del(L1-R2) alleles only keep the direct binding sites of CTCF, and still show substantial insulation activity (*Figure 3*). Further, the insulation activity and the folding property is dependent on the orientation of the binding motifs (*Figure 3*), as in the endogenous TZ region (*Tsujimura et al., 2018*). Therefore, it should be safe to attribute the STITCH insulation primarily to the bindings of CTCF. Blocking of enhancer activity by heterologously inserted CTCF-binding sites is consistent with previous studies (*Bell et al., 1999*; *Hou et al., 2008*; *Liu et al., 2015*).

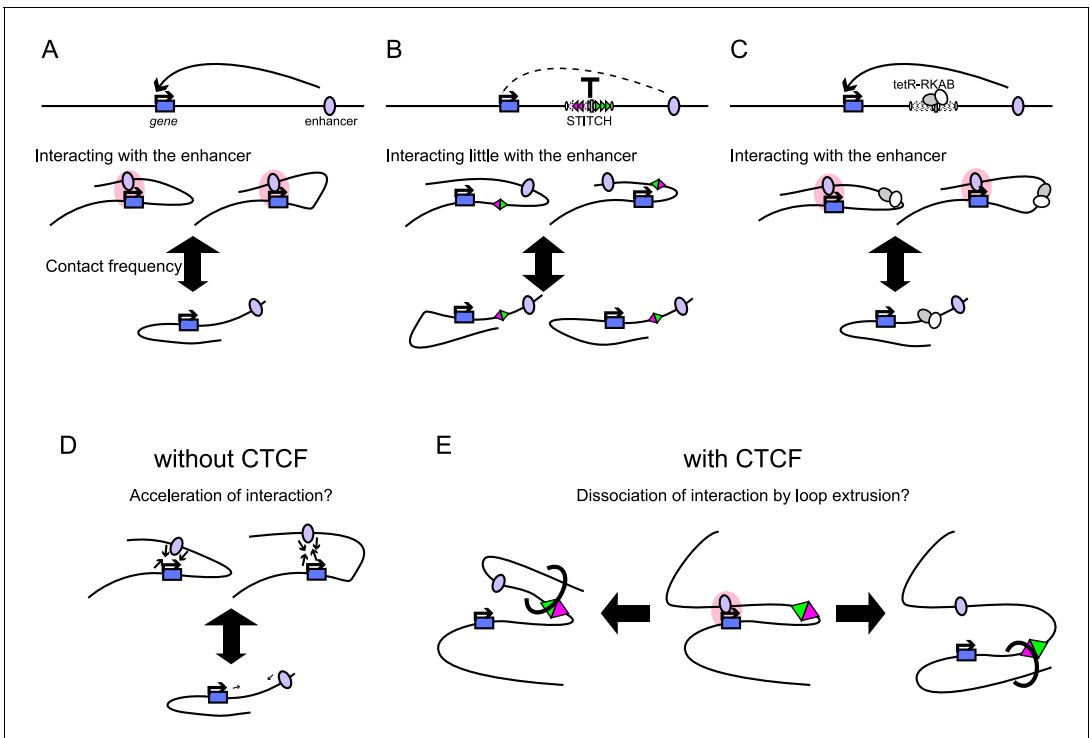

**Figure 9.** Summary of the STITCH system and models for the CTCF insulation. (**A, B**) Schematic illustration of how STITCH blocks the gene-enhancer interaction. STITCH insertion efficiently blocks the interaction, while it also alters the contact tendency of the locus though less prominently. (**C**) Upon the tetR-KRAB induction, the contact frequency becomes normal, and the gene-enhancer interaction is restored. (**D–E**) Models of how CTCF efficiently impairs the gene-enhancer interaction. There might be a mechanism that a slight increase/decrease of contact frequency leads to a drastic increase/decrease of the gene-enhancer interaction (**D**). Also. CTCF might actively disentangle the gene-enhancer interaction through loop extrusion (**E**).

CTCF establishes a boundary of contact domains through the function of blocking the extrusion of cohesin (*Fudenberg et al., 2016*; *Guo et al., 2015*; *Sanborn et al., 2015*). Numerous studies have shown that the domain boundaries with CTCF-binding sites insulate enhancer activation (*Dowen et al., 2014*; *Lupiáñez et al., 2015*; *Narendra et al., 2015*; *Symmons et al., 2014*; *Tsujimura et al., 2015*; *Tsujimura et al., 2018*). From these observations, it is vaguely accepted that CTCF limits the action range of enhancers through the formation of contact domains. However, whether the contact domains are the entity that regulates the gene-enhancer interaction in the context of CTCF insulation is elusive, because their direct causality was not shown so far. Recent comprehensive imaging of chromatin structure showed that the domain-like structures are frequently present across the boundary positions (*Bintu et al., 2018*), showing that the contact domains might be a mere averaged projection of highly variable chromatin structures. Our analysis based on the folding directionality, particularly of the delL allele, may indicate that CTCF can prevent the gene-enhancer interaction without demarcating contact domains (*Figure 3—figure supplement 3*). However, there are various ways to define contact domains (*Zufferey et al., 2018*). Moreover, applying 5C or Hi-C might be more appropriate to describe formation of contact domains than the present 4C-based analyses. Therefore, our study cannot conclude about the causality of the contact domains for the enhancer blocking activity of STITCH.

We instead compared contact profiles of *MYC* between different alleles deeply. First, we showed that the small changes (at most by half) of the contact frequency with the enhancer region lead to the drastic reduction of the *MYC* expression level by up to 20 folds (*Figures 1* and *3*, *Figure 1—figure supplement 1*, *Figure 3—figure supplement 2*). This fact may indicate that the disruption of gene activation should not only be attributed to the simple reduction of the contacts beyond the CTCF-binding sites.

Next, we compared the contact distribution only within the region beyond the STITCH insertion site (*Figure 3H–J*). Then we found that the contact of *MYC* with the super-enhancer/*PVT1* region was enhanced upon the stepwise loss of CTCF-binding sites of STITCH more than with the other non-active regions (*Figure 3H–J*). We think this result well explains, at least in part, how the gradual changes of contact frequency are translated into the skewed expression changes (*Figure 9A–C*).

The genome tends to be compartmentalized into two parts, active and repressive domains (*Lieberman-Aiden et al., 2009*). The depletion of CTCF or cohesin was shown to enhance the compartmentalization (*Nora et al., 2017*; *Rao et al., 2017*; *Schwarzer et al., 2017*). Along with this line, our observation can be interpreted that the preferential association of *MYC* with the super-enhancer obeys the same compartmentalization principle and that the CTCF binding interrupts this process. Then how does CTCF do so? Possibly there might be a mechanism that enhances aggregation of the active regions upon an increase of contact frequencies (*Figure 9D*). For example, the recently proposed phase separation model may explain it well (*Hnisz et al., 2017*). The increase of the overall contact frequency in the absence of CTCF-binding sites in between may boost the compartmentalization. Whether the loop extrusion process by cohesin would further help the association of *MYC* with the enhancer or not is unclear. A previous report has shown that the compartmentalization among super-enhancers is established even between different chromosomes upon depletion of cohesin, suggesting that the loop extrusion is not required for this process (*Rao et al., 2017*). In addition, or alternatively, CTCF per se, probably through anchoring the stabilized or dynamically extruding cohesin loops, might actively disrupt the compartmentalized association of *MYC* with the enhancer, when present in between (*Figure 9E*). Distinguishing the boost effect of the compartmentalization by the increase of contact frequency in the absence of CTCF (*Figure 9D*) and the interference effect of the compartmentalization by the loop extrusion in the presence of CTCF (*Figure 9E*) would be challenging.

The induction of tetR-KRAB impaired binding of CTCF at STITCH and restored the contacts of *MYC* with the enhancer over STITCH. This could be due to the formation of the heterochromatic states that were represented by the H3K9me3 deposition. The heterochromatic regions form dense nucleosomes, which may exclude binding of transcription factors (*Machida et al., 2018*). A previous study indirectly suggested that KRAB induction reduces the binding of CTCF (*Jiang et al., 2017*). Notably, our study shows that the formation of heterochromatin does not prevent the association between genes and enhancers. At the same *MYC* locus, recruitment of the KRAB domain to the *PVT1* promoter did not block the enhancer activation (*Cho et al., 2018*). It should be emphasized that the KRAB protein is generally considered as a repressor protein, and has also been widely used

to repress gene expression artificially. Our study clearly shows that in certain contexts, KRAB might be able to activate gene expression. The prevalence of this kind of regulation in the endogenous genomes needs to be studied in the future.

## STITCH as a novel tool for manipulating gene expression

In this study, we mainly applied the STITCH/KRAB to dissect gene regulation by long-range enhancers. The system has a unique advantage that it can target specifically only one locus without affecting much of the cellular and epigenetic states even around the enhancer region (*Figure 4*, *Figure 4—figure supplement 1*). This is in contrast to many other studies that depleted genes and proteins or induced cellular differentiation and signaling cascades. We think coupling the STITCH/KRAB system with live-imaging techniques and others should further contribute to understanding gene regulation by enhancers.

We also anticipate that STITCH can be a useful tool to disrupt gene function in a tissue-specific manner. Currently, this is predominantly achieved by the Cre-loxP system, which inevitably needs a suitable driver for Cre expression. However, STITCH disruption needs just one insertion between a gene and an enhancer. Our work exemplified that even the enhancers stretching over a vast region could be blocked. Controlling the insulation by KRAB can repeatedly switch on and off gene expression as desired and thus adds another degree of control.

The functionality of STITCH primarily relies on the binding of CTCF, as discussed above. Therefore, its generality should mostly depend on how robustly CTCF binds to STITCH and blocks the gene-enhancer interaction, and on how robustly the KRAB induction controls the binding of CTCF. The motif sequences recognized by CTCF at STITCH are derived from the TZ, to which CTCF consistently binds in various cell types in mice (*Tsujimura et al., 2018*). It was also shown that the TZ blocks chromatin contacts and gene-enhancer interactions in different contexts upon several balanced inversions in the mouse embryos (*Tsujimura et al., 2015*). We show in this study that the motif sequences robustly recruit CTCF in the same way as the TZ does even as a reconstituted DNA cassette in the genome of a different species, human, in both iPSCs and NPCs. We also confirmed that the tetR-KRAB induction expelled CTCF binding from STITCH regardless of the insertion sites and the cell types examined. Moreover, STITCH blocked chromatin contacts and the enhancer activities in these different contexts, as the endogenous TZ does in the mouse genome. These facts well argue that the STITCH system should be applicable robustly to various genomic and cellular contexts. As discussed above, however, there is still uncertainty in how CTCF interrupts the gene-enhancer interaction. Therefore, it cannot be excluded that STITCH might encounter cases where it does not affect the gene-enhancer interaction as expected, which might instead lead to uncovering yet unknown modes of genome regulation by CTCF.

## The *MYC* regulation

*MYC* is one of the four factors of the original cocktail to induce pluripotent stem cells (*Takahashi and Yamanaka, 2006*; *Takahashi et al., 2007*). In the STITCH/KRAB cells, the decrease of *MYC* expression led to a decreased proliferation rate (*Figure 5—figure supplement 1N*). It is well known that *MYC* accelerates cell proliferation in various systems, including cancers (*Bretones et al., 2015*). Further, our transcriptome analysis revealed that down-regulation of *MYC* leads to a decrease of genes involved in several cellular and metabolic processes that are also known to be targets of *MYC* in various cell types (*Figure 2*). These results suggest that in the iPSCs, *MYC* regulates cellular metabolism and proliferation through up-regulation of a specific set of target genes that are also shared by different types of cells, including cancer cells. Further digging into the function of *MYC* in our system should be fruitful in this sense.

## The H3K27me3 mark reflects the gene expression

The STITCH insulation not only down-regulated the gene expression but also affected the epigenetic states of *MYC* (*Figure 4*). We further investigated the temporal change and showed that the deposition of H3K27me3 only follows and reflects, but does not precede and affect gene expression changes. Perhaps this might seem contradictory to the prevailing notion of the histone mark as a repressor. However, the delayed change of the histone modification after the transcriptional change is consistent with previous reports showing the same relationship upon the global induction of

cellular stimuli (*Hosogane et al., 2013*; *Kashyap et al., 2011*), and with the mechanical property of the repressive state as an epigenetic memory (*Reinberg and Vales, 2018*). Also, accumulative evidence has shown that PRC2 has almost no effect on gene expression in a particular context (*Riising et al., 2014*). Yet, mutations in genes encoding PRC2 components have indicated that PRC2 has diverse and critical roles in organisms (*Schuettengruber and Cavalli, 2009*). Also, it was shown that PRC2 maintains gene silencing during the differentiation of mouse ES cells (*Riising et al., 2014*). To explain these observations, it has been proposed that the deposition of H3K27me3 raises the threshold for gene activation (*Comet et al., 2016*). However, the studies involving gene activation so far were carried out under the global induction of cellular stimuli. Therefore, it has not been clear if the H3K27me3 marks regulate gene activation locally as a resistance in cis or rather globally through effects on the cellular and epigenomic states. Our experiment showed that the presence of H3K27me3 makes no significant difference in *MYC* activation upon the local induction by the enhancer (*Figure 7J*), and thus challenged the above hypothesis. The role of this repressive histone mark needs to be further studied in future.

# Materials and methods

## Key resources table

| Reagent type (species) or resource | Designation | Source or reference | Identifiers | Additional information |
|---|---|---|---|---|
| Cell line (*Homo-sapiens*) | 253G1 induced pluripotent stem cells | RIKEN BRC | HPS0002: 253G1, RRID:CVCL_B518 | |
| Cell line (*Homo-sapiens*) | Hap | This paper | | 3 Mb deletion of an allele around *MYC*, in 253G1 cells |
| Cell line (*Homo-sapiens*) | STITCH-30kb | This paper | | STITCH insertion into 30 kb upstream of *MYC*, in Hap cells |
| Cell line (*Homo-sapiens*) | STITCH+30kb | This paper | | STITCH insertion into 30 kb downstream of *MYC*, in Hap cells |
| Cell line (*Homo-sapiens*) | STITCH+440kb | This paper | | STITCH insertion into 440 kb downstream of *MYC*, in Hap cells |
| Cell line (*Homo-sapiens*) | STITCH+1760kb | This paper | | STITCH insertion into 1760 kb downstream of *MYC*, in Hap cells |
| Cell line (*Homo-sapiens*) | STITCH+1790kb | This paper | | STITCH insertion into 1790 kb downstream of *MYC*, in Hap cells |
| Cell line (*Homo-sapiens*) | del(30-440) | This paper | | Deletion of +(30.440)kb region in Hap cells |
| Cell line (*Homo-sapiens*) | delL | This paper | | Deletion of the CTCF binding sites L1-L4 of STITCH in STITCH+30kb |
| Cell line (*Homo-sapiens*) | delR | This paper | | Deletion of the CTCF binding sites R1-R3 of STITCH in STITCH+30kb |
| Cell line (*Homo-sapiens*) | invR | This paper | | Inversion of the CTCF binding sites R1-R3 of STITCH in STITCH+30kb |
| Cell line (*Homo-sapiens*) | inv(L1-R3) | This paper | | Inversion of the whole STITCH in STITCH+30kb |
| Cell line (*Homo-sapiens*) | del(L1-R3) | This paper | | Deletion of the whole STITCH in STITCH+30kb |

*Continued on next page*

*Continued*

| Reagent type (species) or resource | Designation | Source or reference | Identifiers | Additional information |
|---|---|---|---|---|
| Cell line (*Homo-sapiens*) | del(L2-R2) | This paper | | Deletion of the CTCF binding sites L2-R2 of STITCH in STITCH+30kb |
| Cell line (*Homo-sapiens*) | del(L1-R2) | This paper | | Deletion of the CTCF binding sites L1-R2 of STITCH in STITCH+30kb |
| Cell line (*Homo-sapiens*) | STITCH+30kb/KRAB | This paper | | STITCH+30kb with piggy Bac integration of tetR-KRAB-2A-Puro^r |
| Cell line (*Homo-sapiens*) | STITCH+30kb/ tetR-3xFLAG-HA | This paper | | STITCH+30kb with piggyBac integration of tetR-3xFLAG-HA-2A-Puro^r |
| Cell line (*Homo-sapiens*) | STITCH+30kb with Puro^r | This paper | | STITCH+30kb with Puro^r inside STITCH |
| Cell line (*Homo-sapiens*) | del(30-440)/KRAB | This paper | | del(30-440) with piggyBac integration of tetR-KRAB-2A-Puro^r |
| Cell line (*Homo-sapiens*) | NEUROG2/KRAB | This paper | | STITCH insertion into the 65 kb downstream of NEUROG2 in Hap cells, with piggyBac integration of tetR-KRAB-2A-Puro^r |
| Transfected construct (*Escherichia virus P1*) | Cre Recombinase encoding mRNA | OZ Biosciences | Cat#MRNA32-20 | synthetic mRNA encoding Cre recombinase |
| Antibody | anti-CTCF (Rabbit polyclonal) | Millipore | Cat#07–729, RRID:AB_441965 | ChIP (1:88) |
| Antibody | anti-H3K4me3 (mouse monoclonal) | MAB Institute | Cat#MABI0304S, RRID:AB_11123891 | ChIP (1:147) |
| Antibody | anti-H3K27me3 (mouse monoclonal) | MAB Institute | Cat#MABI0323S, RRID:AB_11123929 | ChIP (1:220) |
| Antibody | anti-H3K9me3 (mouse monoclonal) | MAB Institute | Cat#MABI0318S | ChIP (1:176) |
| Antibody | anti-H3K27ac (mouse monoclonal) | MAB Institute | Cat#MABI0309S, RRID:AB_11126964 | ChIP (1:220) |
| Recombinant DNA reagent | pUC-STITCH (plasmid) | This paper | AddGene 129535 | A plasmid carrying STITCH with the homology arms with the MYC+30kb integreation site. *Supplementary file 1B* |
| Recombinant DNA reagent | pUC57-PB-PGK-tetR -KRAB-2A-Puro (plasmid) | This paper | AddGene 129536 | A piggyBac transposon vector encoding tetR-KRAB-2A-Puro^r under the PGK promoter. |
| Recombinant DNA reagent | pUC57-PB-PGK-tetR -3xFLAG-HA-2A-Puro (plasmid) | This paper | AddGene 129537 | A piggyBac transposon vector encoding tetR-3xFLAG-HA-2A-Puro^r under the PGK promoter. |
| Recombinant DNA reagent | Super PiggyBac Transposase Expression Vector | System Biosciences | Cat#PB210PA-1 | |
| Sequence-based reagent | Alt-R CRISPR tracrRNA | Integrated DNA Technologies | Cat#1072532 | |

*Continued on next page*

*Continued*

| Reagent type (species) or resource | Designation | Source or reference | Identifiers | Additional information |
|---|---|---|---|---|
| Sequence-based reagent | Alt-R CRISPR crRNA | Integrated DNA Technologies | | *Supplementary file 1A* |
| Sequence-based reagent | PCR primers | This paper | | *Supplementary file 1C-G* |
| Peptide, recombinant protein | Alt-R S.p. Cas9 Nuclease 3NLS | Integrated DNA Technologies | Cat#1074181 | |
| Peptide, recombinant protein | Dynabeads Protein G | Thermo Fisher Scientific | Cat# 10003D | |
| Peptide, recombinant protein | micrococcal nuclease | New England Biolabs | Cat#M0247S | |
| Peptide, recombinant protein | *Nla*III restriction enzyme | New England Biolabs | Cat#R0125 | 4C-seq Library Prep |
| Peptide, recombinant protein | *Dpn*II restriction enzyme | New England Biolabs | Cat#R0543 | 4C-seq Library Prep |
| Peptide, recombinant protein | T4 DNA ligase | Thermo Fisher Scientific | Cat#EL0014 | 4C-seq Library Prep |
| Peptide, recombinant protein | Tks Gflex DNA Polymerase | Takara | Cat#R060A | 4C-seq Library Prep |
| Commercial assay or kit | NEBNext Poly(A) mRNA Magnetic Isolation | New England Biolabs | Cat#E7490S | |
| Commercial assay or kit | NEXTflex Rapid RNA-Seq Kit | Bioo Scientific | Cat#NOVA-5238–01 | |
| Commercial assay or kit | NEBNext Ultra II DNA Library Prep with Sample Purification Beads | New England Biolabs | Cat#E7103S | |
| Chemical compound, drug | Doxycycline | Sigma Aldrich | Cat#D9891 | |
| Chemical compound, drug | EPZ-6438 | Adipogen Life Sciences | Cat#SYN-3045-M001 | |
| Chemical compound, drug | LDN-193189 | StemRD | | |
| Chemical compound, drug | SB-431542 | Tocris | Cat#1614 | |
| Software, algorithm | WebGestalt | PMID:31114916 | | http://www.webgestalt.org |
| Software, algorithm | DESeq2 | PMID:25516281 | | |
| Software, algorithm | topGO | PMID:16606683 | | |
| Software, algorithm | Bowtie2 | PMID:22388286 | | |
| Software, algorithm | FourCSeq | PMID:26034064 | | |
| Software, algorithm | HISAT2 | PMID:31375807 | | |

*Continued on next page*

*Continued*

| Reagent type (species) or resource | Designation | Source or reference | Identifiers | Additional information |
|---|---|---|---|---|
| Software, algorithm | HOMER | PMID:20513432 | | |
| Software, algorithm | HTSeq | PMID:25260700 | | |
| Software, algorithm | Integrated Genome Viewer | PMID:21221095 | | |
| Software, algorithm | GimmeMotifs | PMID:21081511 | | |
| Software, algorithm | SAMtools | PMID:19505943 | | |
| Software, algorithm | BEDtools | PMID:20110278 | | |
| Software, algorithm | R | CRAN | | |

## Cell culture

The human iPSC line 253G1 (*Nakagawa et al., 2008*) was kindly provided by Prof. Shinya Yamanaka through RIKEN BRC. We cultured the cells in the StemFit AK02N medium (ReproCELL, Cat#R-CAK02N) on dish coated with iMatrix-511 (ReproCELL, Cat#NP892-012) without feeder cells. We added Y-27632 (FUJIFILM Wako, Cat#036–24023) at the final concentration of 10 µM when seeding the cells on a dish. We used the 0.5x of TrypLE Select (Thermo Fisher Scientific K.K., Cat#12563–011) to dissociate the cells for passaging. The iPSCs were sampled for assays in their growth phase, well before the color of the medium turns yellow and cells reach near confluency.

To differentiate the iPSCs to NPCs, we let the iPSCs become almost confluent and then switch the medium to the neural induction medium consisting of 1:1 of DMEM/Ham's F-12 (FUJIFILM Wako, Cat#042–30795) and Neurobasal Plus Medium (Thermo Fisher Scientific K.K., Cat#A3582901), 1X GlutaMAX Supplement (Thermo Fisher Scientific K.K., Cat#35050061), 1X MEM Non-Essential Amino Acids Solution (Thermo Fisher Scientific K.K., Cat#11140050), 1X Penicillin-Streptomycin (Thermo Fisher Scientific K.K., Cat#15140122), 1X N-2 Supplement (Thermo Fisher Scientific K.K., Cat#17502048), 1X B-27 supplement (Thermo Fisher Scientific K.K., Cat#17504044), 0.1 mM 2-Mercaptoethanol (Sigma, Cat#M7522), 250 nM LDN-193189 (StemRD), and 10 µM SB-431542 (Tocris, Cat#1614). The medium was changed every or every other day up to day 6.

DOX (Sigma, Cat#D9891) was basically added at the final concentration of 10 ng/ml unless specifically indicated. When DOX was removed for time-course analysis, the concentration was first changed to 1 ng/ml one day before the start of removal. Then at the start of the removal, the cells were first washed with PBS (Thermo Fisher Scientific K.K., Cat#10010–049), and then fresh medium without DOX was supplied. Further two hours later, wash with PBS and replacement of medium was repeated to ensure the removal of DOX. EPZ (Adipogen Life Sciences, Cat#SYN-3045-M001) was used at the final concentration of 200 nM. For the DMSO controls, the same volume of DMSO as EPZ was added.

We verified the authenticity of the cells by confirming the presence of the pMX-KLF4 transgene in the 253G1 cells (*Nakagawa et al., 2008*; *Takahashi et al., 2007*) with PCR using a primer pair of 5'-CCCTCAAAGTAGACGGCATC-3' and 5'-GGTCTCTCTCCGAGGTAGGG-3'. We tested infection of mycoplasma with HiSense Mycoplasma PCR Detection Kit (CellSafe, Cat#HD-25), and confirmed they were negative.

## Genome editing

To delete the 3 Mb region of the *MYC* locus, we co-transfected the RNP complex of CRISPR/Cas9 targeting both edges of the deletion interval with Lipofectamine RNAiMAX Transfection Reagent (Thermo Fisher Scientific K.K., Cat#13778030) (*Figure 1B*). We assembled the RNP from Alt-R CRISPR crRNA (Integrated DNA Technologies, listed in *Supplementary file 1A*), Alt-R CRISPR tracrRNA, and Alt-R S.p. Cas9 Nuclease 3NLS (Integrated DNA Technologies, Cat#1072532 and

Cat#1074181, respectively), following the manufacturer's protocol. The target sequences of the guide RNAs are described in *Supplementary file 1A*. After the transfection, cells were sparsely re-plated on a dish. Grown colonies were picked up and expanded. The clones were screened for the correctly edited allele by PCR genotyping (see *Supplementary file 1D* for the primer sequences). We then confirmed the deletion by direct Sanger sequencing.

The STITCH vector targeting into the +30 kb position with the homology arm of 150 bp length at each side was synthesized by Integrated DNA Technologies (see *Supplementary file 1B* for the DNA sequences). We amplified the fragment by PCR (see *Supplementary file 1C* for the primer sequences) with Tks Gflex DNA Polymerase (Takara, Cat#R060A) and purified it. Then we transfected it into the cells with Lipofectamine 3000 Transfection Reagent (Thermo Fisher Scientific K.K., Cat#L3000001) together with the RNP complex of CRISPR/Cas9 targeting the insertion site as described above and transfected with Lipofectamine RNAiMAX Transfection Reagent (Thermo Fisher Scientific K.K., Cat#13778030). See *Supplementary file 1A* for the target sequences of the guide RNAs. The positive cells were first selected in the culture medium containing 0.2 mg/L puromycin. Then survived colonies were picked up and expanded. The correct insertion was confirmed by PCR and direct sequencing. We found a single nucleotide mutation within the R3 sequence in the clone that we obtained, which was far away from the core motif for CTCF binding for more than 30 bp. To insert STITCH into the other four sites at the *MYC* locus and the one at the *NEUROG2* locus, we attached 50 bp homology arms by PCR using the STITCH vector as the template (see *Supplementary file 1C* for the primer sequences) and performed the transfection as the same way as above. We screened puromycin resistant clones and then confirmed the insertion by PCR (see *Supplementary file 1D* for the primer sequences). These targeted cells were further transfected with Cre Recombinase encoding mRNA (OZ Biosciences, Cat#MRNA32-20) using Lipofectamine RNAiMAX Transfection Reagent (Thermo Fisher Scientific K.K., Cat#13778030) to remove the puromycin resistant cassette (*Figure 1A*). After the transfection, the cells were sparsely plated on a dish, and colonies were picked up after they formed. We screened positive clones by PCR (see *Supplementary file 1D* for the primer sequences).

To delete or invert the CTCF-binding sites within STITCH, we transfected CRISPR/Cas9 RNPs targeting the edges of the intervals of the deletion/inversion as described above (see *Figure 3—figure supplement 1A–D*). The target sequences of the guide RNAs are described in *Supplementary file 1A*. After transfection with the RNPs, the cells were sparsely seeded, and grown colonies were picked up. The mutations were first screened by PCR (see *Supplementary file 1D* for the primer sequences). Then the DNA sequences were confirmed by direct sequencing. While we tried to obtain the del(L2-R2) clones, we obtained the del(L1-R2) clone, probably due to the excessive excision at the cutting site (*Figure 3—figure supplement 1C*).

To make the del(30-440) allele, we inserted the selection cassette only (i.e., the two loxP sites sandwiching the Puromycin resistant gene inside) of the STITCH vector into the +440 kb position of a delL clone in the same way as above (*Figure 1—figure supplement 2*). The targeting fragment was prepared by two rounds of PCR from the STITCH vector (see *Supplementary file 1C* for the primer sequences). After correct integration, Cre Recombinase encoding mRNA (OZ Biosciences, Cat#MRNA32-20) was transfected, and the deletion allele was selected by PCR screening (see *Supplementary file 1D* for the primer sequences).

To obtain cells that stably express tetR-KRAB, we designed a plasmid vector of a *piggyBac* transposon carrying coding sequence for tetR-KRAB followed by that of the 2A peptide and the puromycin resistant gene (PURO[r]) under the promoter of human PGK gene. The plasmid was synthesized by GenScript. We also designed the *piggyBac* vector containing tetR-3xFlag-HA instead of the KRAB fragment, followed by the same 2A-PURO[r]. This plasmid was also synthesized by GenScript. We transfected the plasmids with Super PiggyBac Transposase Expression Vector (System Biosciences, Cat#PB210PA-1) using Lipofectamine 3000 Transfection Reagent, and screened positive clones under puromycin selection, as described above. We obtained and characterized several clones, but picked one (the clone one in *Figure 5A*) for the subsequent analysis of STITCH/KRAB. We did not isolate single colonies for the NEUROG2/KRAB cells after the *piggyBac* integration, but only expanded all the cells that survived in the presence of puromycin. Therefore the NEUROG2/KRAB cells should be composed of heterogeneous populations with different integration sites of the *piggyBac* cassette. The positive cells were expanded and maintained in the presence of puromycin at 0.1 mg/L.

## RNA extraction, cDNA synthesis, qPCR and library preparation for RNA-seq

RNA was extracted using the High-pure RNA isolation kit (Roche, Cat#11828665001) in the presence of the DNase I included in the kit for most of the study. We subsequently synthesized the cDNA with the High-Capacity cDNA Reverse Transcription Kit (Thermo Fisher Scientific K.K., Cat#4368813). We used KAPA SYBR Fast qPCR Kit (Kapa Biosystems, Cat#KK4621) as the reagent and the Applied Biosystems 7500 Fast Real-Time PCR System (Thermo Fisher Scientific K.K.) for the qPCR reaction for most of this study. We used RNeasy mini kit for the RNA extraction and the Viia 7 Real-Time PCR System (Thermo Fisher Scientific) with TB Green Premix Ex Taq II (Takara Bio, Cat#RR820A) for the qPCR reaction for the analysis presented in *Figures 7A–B* and *8C*, and *Figure 8—figure supplement 2A–B*. The primers used for qPCR assays are listed in *Supplementary file 1E*. To prepare libraries for RNA-seq, we first enriched mRNA using NEBNext Poly(A) mRNA Magnetic Isolation (New England Biolabs, Cat#E7490S). Then subsequently, we used NEXTflex Rapid RNA-Seq Kit (Bioo Scientific, Cat#NOVA-5238–01) for the library preparation with the oligo DNAs designed by ourselves (listed in *Supplementary file 1G*) as primers for the PCR reaction. The libraries were sequenced with HiSeq2500 System (Illumina) using HiSeq SR Rapid Cluster Kit v2-HS (Illumina, Cat#GD-402–4002) and HiSeq Rapid SBS Kit v2-HS 50 Cycle (Illumina, Cat#FC-402–4022).

## 4C-seq library preparation and sequencing

For a 4C-seq library prep, we collected c.a. 1 million cells and fixed them in 2% paraformaldehyde for 10 min at room temperature. Then the cells were lysed in lysis buffer (50 mM Tris (pH7.5), 150 mM NaCl, 5 mM EDTA, 0.5% NP-40, 1% Triton X-100, 1x complete proteinase inhibitors (Roche, Cat#11697498001); 1 ml), passed through a 23-gauge needle, pelleted and frozen in liquid nitrogen. After the cells were resuspended in $H_2O$ and CutSmart Buffer (New England Biolabs, Cat#B7204) and treated with 0.3% SDS and 2.5% Triton X100 at 37°C for 1 hr, respectively, we performed first digestion of the chromatin with 25 units of *Nla*III restriction enzyme (New England Biolabs, Cat#R0125) on a rotator at 37°C for overnight. After heat inactivation of the enzyme, 12.5 units of T4 DNA ligase (Thermo Fisher Scientific, Cat#EL0014) were applied for self-ligation of the digested chromatin. After de-crosslinking and purification, we carried out second digestion with 20 units of *Dpn*II restriction enzyme (New England Biolabs, Cat#R0543). Then the chromatin was again self-ligated with 12.5 units of T4 DNA ligase (Thermo Fisher Scientific, Cat#EL0014). We then performed the inverse PCR from the chromatin of the c.a. 1 million cells as the template to amplify the 4C library from a given viewpoint for 25 cycles with Tks Gflex DNA Polymerase (Takara, Cat#R060A). The primer sequences used for the 1st round of PCR are listed in *Supplementary file 1F*. We purified the DNA with High-pure PCR Product Purification Kit (Roche, Cat#11732676001) and performed the 2nd round of PCR to attach to the libraries adaptor and index sequences for the NGS analysis for eight cycles again with Tks Gflex DNA Polymerase (Takara, Cat#R060A). The DNA sequences of the adaptor/index primers are listed in *Supplementary file 1G*. The DNA was purified with High-pure PCR Product Purification Kit (Roche, Cat#11732676001). The final libraries were pooled and sequenced with the HiSeq2500 system, as described above, except for VP-R3. Note that the sequences were read from the side of *Nla*III for VP-MYC1, -STITCH-left, -STITCH-right, and -NEUROG2 and the *Dpn*II side for VP-MYC2. The sequencing for VP-R3 was performed with the iSeq 100 system using iSeq 100 i1 Reagent (Illumina, Cat# 20021533) in the paired-end mode. The reads from both *Nla*III and *Dpn*II sides were independently used for the subsequent analyses.

## nChIP for histone modifications and CTCF binding, qPCR, and library preparation for sequencing

For nChIP for histone modifications, cells were dissociated from the dish with TrypLE Select (Thermo Fisher Scientific K.K., Cat#12563–011), washed with PBS, and frozen as pellets. After resuspension in ChIP dilution buffer (20 mM Tris-HCl pH8.0, 150 mM NaCl, 2 mM EDTA, 1% Triton X-100), supplemented with 0.05% SDS, 3 mM CaCl2 and protease inhibitors, they were incubated on ice for 10 min, and incubated at 37°C for 2 min. We added 0.48 µl of micrococcal nuclease (NEB, Cat#M0247S) per 1.0 million cells, and incubated them at 37°C for 10 min. To stop the digestion reaction, EDTA and EGTA were added, so the final concentration was 10 mM and 20 mM, respectively. To solubilize the chromatin, we applied sonication with Ultrasonic Homogenizer UH-50 (SMT Co., Ltd.) for three

times of 20 s pulse and incubated them at 4°C for 1 hr. The solubilized chromatin after removal of the cell debris by centrifugation was incubated with antibodies at 4°C for overnight. We used 0.6, 0.4, 0.5 and, 0.6 µl of antibodies per 400,000 cells for H3K4me3, H3K27me3, H3K9me3, and H3K27ac (MAB Institute, Cat#MABI0304S, Cat#MABI0323S, Cat#MABI0318S, and Cat#MABI0309S), respectively. The chromatin with the antibodies was incubated with 6 µl of Dynabeads Protein G (Thermo Fisher Scientific, Cat# 10003D) for one hour. Then the beads were washed three times with ChIP dilution buffer supplemented with 0.05% SDS and subsequently twice with high-salt wash buffer (20 mM Tris-HCl pH8.0, 500 mM NaCl, 2 mM EDTA, 1% Triton X-100, 0.05% SDS). The chromatin was treated with RNase A (50 ng/µl) at 37°C for 15 min and then with Proteinase K (100 ng/µl) at 55°C for 1 hr in ChIP extraction buffer (20 mM Tris-HCl pH 8.0, 300 mM NaCl, 10 mM EDTA, 5 mM EGTA, 0.1% SDS). The DNA was precipitated with ethanol and eluted in 10 mM Tris-HCl pH 8.0 after removal of the beads. We performed the CTCF nChIP exactly as described before with the same polyclonal anti-CTCF antibody (Millipore, Cat#07–729) (*Tsujimura et al., 2018*). For qPCR assays, we used KAPA SYBR Fast qPCR Kit (Kapa Biosystems, Cat#KK4621) as the reagent and the Applied Biosystems 7500 Fast Real-Time PCR System (Thermo Fisher Scientific K.K.) as the platform for the most of this study. We also used the Viia 7 Real-Time PCR System (Thermo Fisher Scientific) with TB Green Premix Ex Taq II (Takara Bio, Cat#RR820A). To prepare nChIP-seq libraries, we used the NEBNext Ultra II DNA Library Prep with Sample Purification Beads (NEB, Cat#E7103S). We basically followed the protocol from the manufacturer but used partly oligo DNAs that we designed by ourselves for the PCR reaction as listed in *Supplementary file 1G,H*. The libraries were sequenced with the HiSeq2500, as described above.

## Data analysis of qPCR assay for gene expression levels

We first confirmed that the amplification efficiency is nearly 100% for all the primer pairs. Therefore, we used the ΔΔCt method to obtain the relative expression levels normalized to *ACTB*. As a reference sample, we used a large stock of cDNA prepared from the same iPSC line (253G1), which were cultured in a different condition from the present study (with feeder cells in a different medium), and always placed the reference sample in duplicates or triplicates in the same PCR plates, when measuring the Ct values of samples.

Replicates were defined differently for different experimental purposes. For STITCH insertions and del(30-440), replicates mean independent clones that were segregated after Cre transfection. The relative expression levels were measured for each clone and plotted in *Figure 1D*. For mutant clones of STITCH, replicates mean independent clones after CRISPR/Cas9 genome editing. The relative expression levels were measured for each clone and plotted in *Figure 3A*. We also obtained sub-clones from Hap and treated them as replicates in *Figure 1D* and *Figure 3A*. In *Figures 1D* and *3A*, the mean values of the replicates were also represented as bars. The relative expression levels of the STITCH mutants and the Hap clone in *Figure 3I and J*, and *Figure 3—figure supplement 2B and C* were the mean values of the replicates. We obtained five and three clones after the transfection of tetR-KRAB and tetR-3xFlag-HA transposons, respectively. The relative expression levels of *MYC* and the puromycin resistant gene were assayed for all of these clones in *Figure 5C* and *Figure 5—figure supplement 1A and B*. For the treatment of the STITCH/KRAB cells with DOX and EPZ, we used only one representative clone (the clone 1), and performed replicate experiments, which mean samples separately treated with drugs in different dishes (*Figures 5D* and *7A,B,I,J*). For the NEUROG2/KRAB cells, we obtained only one group of cells and performed replicate experiments for each condition. We performed one-way ANOVA with Tukey's multiple-comparison post hoc test to infer statistical significance between different conditions in *Figure 7A and B*, and one-sided Welch's two-sample t-test in *Figure 7I,J* for the statistical significance between the DMSO and EPZ treatments and in *Figure 8C* for the statistical significance between with and without DOX. The data were represented as graphs with the ggplot2 package in R.

## Cell proliferation assay

To compare cell proliferation rates between conditions with and without DOX, we first seeded equal volumes of cells in three replicates for each from a single population in the same medium without DOX. On the next day, we replaced the medium with a fresh one with or without DOX. After five days, the cell numbers were counted using a hemocytometer. We represented the relative

proliferation rates as normalized cell numbers divided by the mean number of cells in the DOX minus condition. The assay was performed for both the Hap and STITCH/KRAB clones. We performed two-sided Welch's two-sample t-test to infer the statistical significance between the two conditions.

## Data analysis of RNA-seq

We prepared and sequenced libraries from three replicate clones (see above) for each of Hap, STITCH+30kb, and del(30-440). We first combined separately sequenced reads of the same libraries from different lanes as fastq files. We mapped the sequences to the human genome (hg19) with HISAT2 (*Kim et al., 2019*). We made BedGraph tracks with HOMER (*Heinz et al., 2010*) and visualized them in Integrative Genomics Viewer (version 2.4.6) (IGV) (*Robinson et al., 2011*). The data ranges are indicated by counts per 10 million. We assigned the mapped reads to annotated genes with HTSeq (*Anders et al., 2015*). We normalized the counts and calculated log two fold changes between different conditions with the 'normal' shrining algorithm in DESeq2 (*Love et al., 2014*). To perform GSEA, we input the shrunken log2 fold change values into WebGestalt (http://www.web-gestalt.org) (*Liao et al., 2019*), selecting GSEA (*Subramanian et al., 2005*) as the method and HALLMARK50 (*Liberzon et al., 2015*) as the functional database. To call differentially expressed genes, we set the threshold as the adjusted p-value<0.05 and the shrunken log2 fold change >0.5 with DESeq2. We visualized the shrunken log2 fold changes and the base means as the MA-plots using the ggplot2 package in R. The Venn diagram was drawn with the VennDiagram package in R (*Chen and Boutros, 2011*). The GO term enrichment analysis was performed with the topGO package in R (*Alexa et al., 2006*), where Fisher's exact test was employed for the statistical test. The data were visualized with the ggplot2 package in R.

## Data analysis of 4C-seq

We only employed a representative clone for each genomic configuration for the 4C-seq assays. However, for each viewpoint in the most cases, we prepared a couple of replicate libraries that were separately prepared from different dishes, to confirm the reproducibility of the experiment (*Figure 1—figure supplement 1B* and *Figure 3—figure supplements 2A* and *3A*).

We first combined separately sequenced reads of the same libraries from different lanes as fastq files. The sequences of the viewpoint fragment up to the restriction sites were removed with FASTX-Toolkit. Then we mapped the rest of the sequences to the human genome (hg19) using Bowtie2 mostly with the default settings except that the –score-min option was set as 'L,−0.1,–0.1' (*Langmead and Salzberg, 2012*). The generated SAM files were converted to BAM files, indexed and sorted with SAMtools (*Li et al., 2009*). We used the FourCSeq package to normalize the counts as reads per million (RPM), smooth them with the window size of seven fragments, and produce BedGraph files (*Klein et al., 2015*). We visualized the tracks in Integrative Genomics Viewer (version 2.4.6) (*Robinson et al., 2011*). The data ranges are indicated by counts per million. Counting the number of reads mapped to given regions was performed with BEDTools (version 2.26.0) (*Quinlan and Hall, 2010*). To calculate contact frequencies, we divided the read numbers in a given region by the total read numbers mapped to the defined locus except for the 10 kb region from the viewpoint fragment. When analyzing the directionality of chromatin folding, we combined the read numbers of replicates from the same viewpoints. To perform PCA, we first counted reads in defined bins. We took 30 kb and 10 kb as the sizes of the bins for VP-MYC1/2 and VP-NEUROG2, respectively. We combined the read numbers of replicates from the same viewpoints (either VP-MYC1 or VP-MYC2). Then we calculated ratios of reads in each bin within the region of interest (whole locus, the left 900 kb region, or the right 600 kb region for VP-MYC1/2). Then we performed PCA using the data sets with the prcomp function in R. The component loadings were calculated using the sweep function in R. The R codes used for the analyses are shown in *Figure 3—source code 1* and *Figure 8—source code 1*. To perform the correlative analysis between the 4C-seq counts and gene expression levels, we also combined reads of replicates and calculated contact frequencies first. Then, the linear regression was performed against the log-log plot to obtain the slope in R. The Spearman's rank correlation coefficients were also calculated using a function in R. The log-log plots were visualized using the ggplot2 package in R.

## Data analysis of nChIP-qPCR assay

We always took input samples for every nChIP and calculated enrichment as ratios to the input samples. Our replicates mean different nChIP samples derived from separately cultured cells in different dishes. In order to cancel the inevitable variance in the total enrichment efficiency of nChIP experiments, we normalized the enrichment at *MYC* to those at control regions, which were the *ACTB* region for the active H3K4me3 mark and the *T* region for the repressive H3K27me3 mark. As the treatment with EPZ causes an epigenetic change in genome-wide, we did not do the normalization in *Figure 7G and H*. To test statistical significance between different conditions in *Figure 7C–F*, we performed one-way ANOVA with Tukey's multiple-comparison post hoc test. To assess statistical significance between treatments with DMSO and EPZ, we performed Welch's two-sample t-test in *Figure 7G and H*.

## Data analysis of nChIP-seq

The reads from the same libraries were first combined as a fastq file when they were sequenced in different lanes. We mapped the data to the human genome (hg19) using Bowtie2 with the same options as the 4C-seq (*Langmead and Salzberg, 2012*). Then, we generated BedGraph files for visual inspection with HOMER (*Heinz et al., 2010*). Peak calling was also performed with HOMER. We also mapped reads to a synthetic genomic DNA carrying the STITCH sequence inside. For this purpose, we first retrieved unmapped reads and reads that are likely to be unique from the mapped BAM file with SAMtools, with scripts of 'samtools view -b -f 4' and 'samtools view -b -q 10', respectively, and combined them together, in order to remove reads that can be potentially mapped to repeat sequences. Then we re-mapped the reads against the custom reference genome. The subsequent generation of BedGraph files was carried out as above with HOMER (*Heinz et al., 2010*). We visualized the BedGraph tracks in IGV (*Robinson et al., 2011*). The data ranges are indicated by counts per 10 million.

To calculate the log2 fold change for H3K4me3, H3K27me3, H3K27ac, we first obtained a list of peaks that are called at least two among the four experiments (two replicates from the Hap and two from the STITCH+30kb). Next, we counted the read counts mapped to the peaks for each experiment. Then, we calculated the log2 fold change for each peak normalized by the size factors determined by the read counts in all the peaks, using the framework of DESeq2, without the shrinking algorithm. We ranked the peaks according to the values and plotted with the ggplot2 package in R.

Similarly, for CTCF nChIP-seq, we obtained a list of peaks that are called at least two among the six experiments (two replicates from STITCH+30kb, STITCH/KRAB with DOX, and STITCH/KRAB without DOX). We determined the orientations of the CTCF binding using GimmeMotifs (*van Heeringen and Veenstra, 2011*) with the position weight matrix from the HOCOMOCO database (*Kulakovskiy et al., 2018*), with the threshold of false discovery rate <0.1. To calculate the log2 fold change between plus and minus of DOX, we counted the read counts mapped to the peaks for each experiment. For the binding at STITCH, we separately count the reads against the synthetic genome. Then, we calculated the log2 fold change and the base means for each peak normalized by the counts in all the peaks, as described above. We ranked the peaks according to the log2 fold changes. The rank plot and the MA-plot were generated with the ggplot2 package in R.

## Acknowledgements

We thank Prof. Shinya Yamanaka (Kyoto University) for providing us the hiPSC line. We would also like to thank Drs. Sumihiro Maeda, Kent Imaizumi, Tsukasa Sanosaka, and Tomohiko Akiyama for their scientific advice and generous supports in revising the manuscript.

## Additional information

**Competing interests**

Taro Tsujimura, Hideyuki Okano, Keiichi Hishikawa: An inventor on the Japanese patent application (2018-154577, filed on 21 August 2018) and the PCT international patent application (PCT/JP2019/

032106, filed on 16 August 2019) in respect of the STITCH/KRAB system by Keio University. The other authors declare that no competing interests exist.

## Funding

| Funder | Grant reference number | Author |
| --- | --- | --- |
| Japan Society for the Promotion of Science | Grants-in-Aid for Young Scientists (B) (17K16072) | Taro Tsujimura |
| Japan Society for the Promotion of Science | Grants-in-Aid for Scientific Research (B) (15H03001) | Keiichi Hishikawa |
| Japan Society for the Promotion of Science | Grants-in-Aid for Scientific Research (C) (16K09602) | Osamu Takase |
| Japan Society for the Promotion of Science | Grants-in-Aid for Scientific Research (C) (15K09244) | Masahiro Yoshikawa |
| Mutou Group | | Hideyuki Okano |
| APA Group | | Hideyuki Okano |
| IMS Group | | Hideyuki Okano |
| Alba Lab | | Hideyuki Okano |
| Kobe One Medicine, One Health | | Hideyuki Okano |
| Japan Agency for Medical Research and Development | Acceleration Program for Intractable Disease Research Utilizing Disease-specific iPS Cells | Hideyuki Okano |
| Keio University | Program for the Advancement of Research in Core Projects on Longevity of KGRI | Hideyuki Okano |

The funders had no role in study design, data collection and interpretation, or the decision to submit the work for publication.

## Author contributions

Taro Tsujimura, Conceptualization, Data curation, Formal analysis, Funding acquisition, Investigation, Visualization, Methodology; Osamu Takase, Masahiro Yoshikawa, Resources, Funding acquisition, Methodology; Etsuko Sano, Investigation, Methodology; Matsuhiko Hayashi, Project administration, MH made critical contribution to the decision of starting the project, and supported the set-up. MH have read the manuscript and approved it to be published; Kazuto Hoshi, Tsuyoshi Takato, Resources, Project administration; Atsushi Toyoda, Data curation, Investigation, Methodology; Hideyuki Okano, Resources, Funding acquisition, Validation, Investigation, Methodology, Project administration; Keiichi Hishikawa, Conceptualization, Resources, Funding acquisition, Validation, Investigation, Methodology, Project administration

## Author ORCIDs

Taro Tsujimura https://orcid.org/0000-0002-3281-0150
Atsushi Toyoda http://orcid.org/0000-0002-0728-7548
Hideyuki Okano http://orcid.org/0000-0001-7482-5935

## Decision letter and Author response

Decision letter https://doi.org/10.7554/eLife.47980.sa1
Author response https://doi.org/10.7554/eLife.47980.sa2

# Additional files

## Supplementary files

• Supplementary file 1. Tables for DNA sequences of oligo DNAs, of gRNA target sites, of the STITCH construct, and of indexes for NGS libraries. (**A**) List of guide RNAs for CRISPR genome editing used in the study. (**B**) The DNA sequences of the elements composing STITCH. (**C**) List of primers used to prepare the targeting cassettes. (**D**) List of primers used for the genotyping. (**E**) List of primers used in the qPCR assays. (**F**) List of primers used for the 4C 1st PCR. (**G**) List of primers used to prepare the NGS libraries. (**H**) List of the NGS libraries.

• Transparent reporting form

## Data availability

 Allthe deep sequencing data of the 4C-seq, RNA-seq and nChIP-seqlibraries analyzed in this study were deposited in ArrayExpress:E-MTAB-7668, E-MTAB-7669, E-MTAB-7670, E-MTAB-8492, andE-MTAB-8957.

The following datasets were generated:

| Author(s) | Year | Dataset title | Dataset URL | Database and Identifier |
|---|---|---|---|---|
| Tsujimura T | 2019 | RNA-seq of wild type (Hap), insulation (STITCH+30kb) and deletion (del(30-440)) of the MYC enhancer in human iPS cells. | https://www.ebi.ac.uk/arrayexpress/experiments/E-MTAB-7669 | ArrayExpress, E-MTAB-7669 |
| Tsujimura T | 2019 | 4C-seq from viewpoint at MYC promoter (VP-MYC1 and VP-MYC2), in wild type (Hap) and variously modified alleles around the locus in human iPS cells. | https://www.ebi.ac.uk/arrayexpress/experiments/E-MTAB-7668 | ArrayExpress, E-MTAB-7668 |
| Tsujimura T | 2019 | nChIP-seq for CTCF, H3K4me3, H3K27me3 and H3K9me3, in wild type (Hap), STITCH+30kb and STITCH/KRAB clones of human iPS cells. | https://www.ebi.ac.uk/arrayexpress/experiments/E-MTAB-7670 | ArrayExpress, E-MTAB-7670 |
| Tsujimura T | 2019 | 4C-seq to show the effects of insertion of STITCH into MYC+30kb and NEUROG2-65kb positions on the chromatin conformation in human iPSCs and differentiated neural progenitor cells | https://www.ebi.ac.uk/arrayexpress/experiments/E-MTAB-8492 | ArrayExpress, E-MTAB-8492 |
| Tsujimura T | 2020 | 4C-seq to show the effects of insertion of STITCH into NEUROG2-65kb positions on the chromatin conformation in neural progenitor cells differentiated from human iPSCs | https://www.ebi.ac.uk/arrayexpress/experiments/E-MTAB-8957 | ArrayExpress, E-MTAB-8957 |

The following previously published datasets were used:

| Author(s) | Year | Dataset title | Dataset URL | Database and Identifier |
|---|---|---|---|---|
| Barakat TS, Halbritter F, Zhang M, Rendeiro AF, Bock C, Chambers I | 2016 | Functional dissection of the enhancer repertoire in human embryonic stem cells | https://www.ncbi.nlm.nih.gov/geo/query/acc.cgi?acc=GSE99631 | NCBI Gene Expression Omnibus, GSE99631 |
| Lister R, Pelizzola M, Dowen RH, Hawkins RD, Hon G, Tonti-Filippini J, Nery JR, Lee L, Ye Z, Ngo Q, Edsall L, Antosiewicz-Bourget J, Stewart R, | 2011 | Reference Epigenome: ChIP-Seq Analysis of H3K27ac in Neural Progenitor Cells; renlab.H3K27ac.NPC.02.01 | https://www.ncbi.nlm.nih.gov/geo/query/acc.cgi?acc=GSM767343 | NCBI Gene Expression Omnibus, GSM767343 |

| | | | | |
|---|---|---|---|---|
| Ruotti V, Millar AH, Thomson JA, Ren B, Ecker JR | | | | |
| Dixon JR, Jung I, Selvaraj S, Ren B | 2015 | Global Reorganization of Chromatin Architecture during Embronic Stem Cell Differentiation | https://www.ncbi.nlm. nih.gov/geo/query/acc. cgi?acc=GSE52457 | NCBI Gene Expression Omnibus, GSE52457 |

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
