## [Decision Letter]

**Acceptance summary:**

This study describes development of a inducible insulator cassette, STITCH, that can act as boundary element that blocks long-range chromatin interactions. This can be a very valuable tool to dissect rules of long-range promoter – enhancer communication, and chromosome folding through mechanisms such as loop extrusion.

**Decision letter after peer review:**

Thank you for submitting your article "Controlling gene activation by enhancers through a drug-inducible topological insulator" for consideration by *eLife*. Your article has been reviewed by three peer reviewers, including Job Dekker as the Reviewing Editor and Reviewer #1, and the evaluation has been overseen by Jessica Tyler as the Senior Editor.

The reviewers have discussed the reviews with one another and the Reviewing Editor has drafted this decision to help you prepare a revised submission.

Summary:

In this paper, the authors present a new tool to modify chromosome structure and enhancer-promoter interactions. The major advance over previously identified insulators is that this tool (STITCH) is designed to include not only divergent tandem arrays of CTCF sites to create a boundary, but also interspersed tetO arrays to allow inducible regulation of CTCF binding. The tetO arrays are added between the CTCF sites such that with the addition of a tetR-KRAB transgene to the cell line, CTCF binding can be disrupted by inducing heterochromatin formation at STITCH in a doxycycline repressible manner. Disruption of CTCF binding to STITCH leads to increased interactions and enhancer-promoter contacts across the STITCH insertion site, and can modify gene expression. Therefore, STITCH is an inducible insulating element, which would be a broadly useful tool in the field of chromosome structure and beyond, such as in genome editing applications.

Essential revisions:

1) Given that STITCH is presented as a tool, all reviewers felt that the approach should be used for 2-3 case studies, ideally in at least 2 different cell types. This will help show how generally applicable the tool is and how robust the results are.

2) All reviewers felt that the PCA analysis was confusing and did not make a critical contribution. Please thoroughly revise the corresponding text and figures, or remove this analysis from the manuscript.

3) It is critical to show whether TetR binding alone, without the KRAB domain, affects CTCF binding and/or CTCF-mediated insulation.

4) The H3K27 ChIP experiments need to be repeated to provide better statistics to support the conclusion that changes in the levels of this mark can be accurately quantified. There are concerns this data set is not of high quality. Also, in Figure 4C-D positive and negative controls should be added.

5) The transcriptional analysis In Figure 2 could be improved. The cutoffs used to identify differentially expressed genes with DESeq2 are very loose (adjusted p-value = 0.1, no limitation is imposed on fold changes). We suggest to repeat the analysis with more than one clone per condition.

6) Please extend the discussion about the relationship between structural changes induced by STITCH and potential new loops induced by the ectopic CTCF sites in the light of the current understanding of CTCF orientation (and loop extrusion?). Reciprocal 4C viewpoints based on endogenous CTCF sites could also help clarifying this matter.

7) Upon publication we feel it is critical that the tools are made available to the community (e.g. putting plasmids on AddGene).

Reviewer #1:

The authors overstate the novelty of their results with respect to insulating elements. Insulating elements have been previously identified, and usually contain CTCF sites, similar to the STITCH sequence (Bell et al., (1999); Liao et al., (2018); Emery, (2011)). This should be more adequately referenced and introduced. In addition, it has already been shown that CTCF sites that form TAD boundaries will block enhancer-promoter interactions, yet this is presented as a novel result (subsection “Titrating blocking activity of STITCH by serial mutations of the CTCF binding sites”) (Hou et al., (2008); Guo et al., (2015); Braikia et al., (2017)).

The conclusion that the enhancer blocking activity and chromosome interaction activity are due to separate mechanisms is not sufficiently explained or supported. It is unclear why loop extrusion and enhancer blocking are introduced as separate mechanisms (Introduction), when the current understanding in the field is that enhancer blocking by CTCF sites is likely due to the creation of new TAD boundaries, which are formed by loop extrusion being blocked by CTCF sites (Recently reviewed in Schoenfelder and Fraser, (2019)).

Similarly, there appears to be confusion about the various mechanisms of long-range chromatin interactions in the Discussion section. Subsection “Mechanism of the STITCH insulation and its control by heterochromatin induction”, second papragraph is very confusing. In the current work CTCF is removed, from the locus, but cohesin is not. Therefore, there is now unblocked loop extrusion throughout the locus. Given that the interaction between the enhancer and MYC is blocked by CTCF, it is much more likely that the increased interaction between the super-enhancer and MYC when CTCF is removed is driven by loop extrusion.

Technical details and biological conclusions are not adequately explained in the text throughout this manuscript. The manuscript would benefit from improving the explanations of why specific analyses are used, and what the results signify biologically, beyond just stating the observations. In particular, the explanations of PCA analysis, component loading plots, and power law scaling between gene expression and 4C-seq contact frequency should be clarified. As it is presented now it is entirely unclear what the PCA analysis really adds. Additional technical details of the computational methods used to analyze the sequencing data is also needed, preferably an online repository for the code used to generate the plots and run the statistical tests should be included with the manuscript.

While 4C-seq is a useful technique for studying the specific interactions between MYC and surrounding loci, it would be beneficial to also compare this to an all-by-all or many-by-many chromosome conformation capture method such as Hi-C or 5C to show the endogenous organization of this region. Putting the STITCH insertion in the context of the landscape of genomic architecture (where are TADs or compartments found in this region?) would strengthen the manuscript and might help to understand how the different directionalities of the CTCF motifs in STITCH are working, as the current explanation is unclear. In addition, it would strengthen the manuscript to compare the contact frequency changes in the STITCH mutants at the MYC locus to changes in the endogenous TZ locus with similar modifications from previous publications by these authors, to determine how variable this behavior is at different genomic loci.

Reviewer #2:

The role of CTCF, cohesin, TADs and 3D genome organization in regulating gene expression and enhancer-promoter (E-P) contacts is currently being intensely studied. In particular, whether CTCF sites and TADs really regulate E-P contacts and gene expression has recently become controversial, with some studies claiming that CTCF and TADs have no role in regulating gene expression (e.g. see https://www.biorxiv.org/content/10.1101/609941v1).

Most studies (including the preprint above) have taken a "deletion" approach to this issue: take a natural E-P pair and TAD and then go in and start deleting or inverting etc. CTCF binding sites and see how gene expression is affected. What is nice about this study is that they take the opposite approach – a kind of "addition" approach. They add the STICH array of CTCF binding sites to the MYC gene at different locations and see how contact frequency (4C) and gene expression (qPCR) of MYC in hiPSCs is affected. The 2 most important points in my opinion are:

1) CTCF sites really can block E-P contact and strongly (~20-fold) affect gene expression. At least for the MYC gene.

2) Histone modifications can be used to turn ON and OFF CTCF insulation with quite high temporal resolution.

I believe the authors get about as close to causality as is realistic with the current tools of molecular biology, which is nice.

Beyond some important but addressable concerns (poor writing, at times confusing figures and presentation, occasionally poor referencing, tool availability), my major concern is this: The authors report STICH as a tool. 2 key features in a tool that are desirable to have are: (1) robustness and (2) generality. But because the authors only apply STICH to one locus (MYC) in one cell type, we cannot really tell if STICH is likely to block E-P contacts in general and robustly in many other loci and other cell types. The impact of STICH would have been greatly increased if the authors could have applied it to 2-3 case studies, ideally in at least 2 different cell types.

So overall, I believe this is a nice contribution with some really important insights, but that the general interest and impact could have been substantially improved if the authors had applied STICH to at least 2-3 different systems and if they can improve their presentation.

Specific issues:

Writing: the paper is for the most part reasonably written, but there are at least >25 cases of poor English and/or syntax/grammar issues. This is too many for a reviewer to fix and I suggest that the authors go through and clean up the issues.

Discuss results in context: First sentence of the Abstract and in the Introduction suggest that "regulation of gene-enhancer interaction is better understood,". I would argue that this is not true. In fact, recently several people (e.g. https://www.biorxiv.org/content/10.1101/609941v1) have begun arguing that CTCF plays essentially no role in the regulation of gene expression. The fact that the authors see such clear results on MYC, in my opinion only increases the impact and value of this study. Therefore, it would be nice if the authors could discuss their MYC result a little bit more clearly in the Discussion section in the context of the many recent studies arguing that CTCF plays no or only a minor role in regulating E-P contacts and gene expression.

Key resources should be available: First of all, my apologies if the authors already did this. But I tried to find this information and was unable to. The authors must put the key STICH plasmids on AddGene for the community, since the value of a tool is largely derived from it being readily accessible for the community. The DNA sequences of STICH must also be available with the paper. I could not find the DNA sequences of the full STICH sequence nor could I find the sequences of the specific CTCF binding sites. These must be available.

RNA-seq Results: The RNA-Seq studies in Figure 2 were really nice. But I could not understand why the STICH +30kb cell line would have ~2-3x more deregulated genes than the del(30-440) cell line. Although STICH is powerful, deleting the enhancer should still have a stronger effect on expression than just blocking it. Could the authors better explain this?

PCA-analysis: In Figure 3A-B, the analysis of how 4C reads in the different regions depend on the STICH construct was really nice. It was also very interesting to see the highly non-linear scaling between 4C contact frequency and gene expression (Figure 3I-J). Both of these are really important contributions in my opinion.

But the PCA-analysis was extremely confusing and convoluted. I really tried to follow the text and the figures, but it was very difficult for me to understand what the point was. In the Results section, the authors spend a lot of text and a huge number of figure panels on this, but I really could not understand it. My suggestion would be to remove all the figures and text pertaining to PCA or at least radically simplify the figures and the text to make it easier to understand. What is the major biological insight coming from this PCA analysis? What is component loading?

Subsection “Insulation and deletion of the enhancer resulted in similar transcriptome profiles”: authors out-of-the-blue reference VP-MYC1 and VP-MYC2, without any figure REF. I could not understand this.

tetR-KRAB studies: The Tet-R KRAB studies were very nice. I may have missed prior studies, but to my knowledge this is the first clear and causal demonstration that histone modifications can turn OFF CTCF insulation. However, one control I was missing was a DNA-binding control. TetR-KRAB binding could disrupt CTCF binding and insulation through 2 ways: DNA-binding competition (e.g. TetR-binding outcompetes CTCF binding) or KRAB-deposition of histone modifications. I would have liked to see a control showing that TetR binding alone – without the KRAB-domain – does not affect CTCF-mediated insulation.

But pretty neat to see that STICH insulation directly affects cell proliferation (subsection “Titrating blocking activity of STITCH by serial mutations of the CTCF binding sites”).

Figure 6. F and G have errors bars, but Figure 6 C, D, and E do not. Need to add errors bars to these.

Otherwise, the time-course results were also pretty cool.

Reviewer #3:

The manuscript describes a strategy to modulate chromosomal contacts in the vicinity of the endogenous MYC gene in human iPS cells through the ectopic insertion of an array of CTCF sites. The approach (named STITCH) seems to be able to alter MYC transcription levels, which correlates with changes in interaction frequencies between the MYC locus and a super-enhancer region downstream. The authors further monitor chromatin states at the engineered locus upon the induction of H3K9 trimethylation by targeted recruitment of a KRAB domain at the STITCH cassette, which is shown to disrupt CTCF binding and restore wild-type chromosomal contacts. The authors conclude that CTCF-mediated modulation of chromosome interactions is the driver of transcriptional changes.

The study is interesting and well designed, and has the potential to bring insight into how gene expression could be modulated by manipulating chromosome structure. However, it suffers from several major drawbacks that should be thoroughly addressed.

1) Many of the native ChIP-seq experiments in the manuscript are difficult to interpret and it is often difficult to agree with the authors on the changes they describe. The zoom level in all Figures is way too low to visually appreciate any local changes at the MYC locus, the STITCH cassette and the neighboring region. More importantly, some crucial experiments (notably the H3K27ac and H3K27me3 ChIP-seq reported in Figure 1, Figure 4 and Figure 4—figure supplement 1) appear to be strongly suboptimal. It is hard to imagine that a local increase of ~2 counts in a 0-6 range as reported in Figure 1 and Figure 4 really does correspond to a specific enrichment as opposed to technical noise.

The authors should perform new H3K27ac/me3 ChIP-seq experiments, provide statistics to support the notion that changes in these chromatin marks can really be quantified and discuss their findings in the light of the new experiments. Crosslinking ChIP-seq would be a viable option in this context – in fact I found it quite unclear why native ChIP was required in this particular study.

2) The correlation between MYC transcription levels and contact frequencies with the super-enhancer region (Figure 3) in mutant STITCH cell lines are interesting, and well supported by the large number of independent clones analyzed. Unfortunately, the structural changes induced by ectopic CTCF sites were not correlated with the position and orientation of endogenous CTCF sites. MYC itself is highly bound by CTCF, as can be more or less seen (again at regrettably low resolution) in Figure 5F. I would suggest the authors to thoroughly discuss the relationship between structural changes induced by STITCH and potential new loops induced by the ectopic CTCF sites in the light of the current understanding of CTCF orientation (and loop extrusion?). Reciprocal 4C viewpoints based on endogenous CTCF sites could also help clarifying this matter.

3) It is unclear how many copies of the STITCH cassette have been integrated at the MYC locus. The authors should provide evidence that a single insertion of 6 CTCF sites is actually responsible for the observed structural changes, as opposed to multiple tandem repeat insertions (especially since lipofection -and hence large amounts of DNA per cell- was used to generate the Cas9 assisted knock-in).

4) The transcriptional analysis In Figure 2 could be improved. The cutoffs used to identify differentially expressed genes with DESeq2 are very loose (adjusted p-value = 0.1, no limitation is imposed on fold changes). In the absence of differential gene expression analysis on more than one STITCH and del(30-440) clones, it is difficult to assess what the >1000 genes detected as differentially expressed under these loose criteria actually represent. I would suggest to repeat the analysis including more than one clone per condition and using more stringent criteria (e.g. padj<0.01, |log2(FC)|>1) in order to identify mis-regulated genes more robustly and reliably. Also, a qPCR validation of significantly up- or down-regulated genes is missing.

Finally, there is no explanation for the fact that the effect on transcription in the deletion mutant is smaller than in the STICH mutant. If the changes are indeed due to the insulation of the super-enhancer region from the MYC gene, then deletion of the super-enhancer region should lead to an even stronger effect on transcription.

5) It would be nice to prove that transcriptional changes in the STITCH and del(30-440) lines are really caused by downregulation of MYC, which could be done notably by overexpressing MYC and testing if normal expression programs are rescued.

6) The PCA analysis is Figure 3 is in principle interesting and laudable as an attempt to quantify differences in 4C profiles in a quantitative and unbiased way. However, the text is somewhat obscure and panels 3C-H are difficult to interpret. It is unclear why the results shown in Figure 3E-H, where PCA is performed on a subset of the data, are so different from panel 3D. These differences are acknowledged in the main text but I did not understand how they are interpreted by the authors. I would actually suggest that the text relative to Figure 3 is entirely re-written and clarified (e.g. please explain what "component loading" means in this context). In addition, the PCA results should be integrated with a discussion of whether they correlate or not with the position and orientation of endogenous CTCF sites (see point 2 above).

7) Transcriptional downregulation of MYC is attributed to changes in contact frequencies due to the presence of ectopic CTCF sequences at the STITCH cassette, which is supported by the strong correlation observed in Figure 3I. If this is really the case, and is due to CTCF looping from STITCH onto endogenous CTCF sites, then it should be possible to recapitulate the phenotype by deleting the endogenous partner CTCF sites. This would significantly strengthen the interpretation of the data.

8) in Figure 4C-D, negative and positive controls are missing (i.e. one or more regions where H3K4me3 should not be detected, and a region that is heavily bound by H3K27me, such as a poised gene, or a Hox gene). This is a very important control though, because one of the most interesting observations in the manuscript is that the transcriptional downregulation of MYC correlates with higher H3K27me3 levels. However, how much H3K27me3 is deposited? How does it compare with poised and/or inactive loci?

9) In Figure 5, it is impossible to understand which changes are occurring at the MYC promoter in terms of H3K9me3 and CTCF levels. This is nonetheless crucial to interpret the gene expression changes upon Dox induction and how they are related to targeted recruitment of KRAB. It seems that the CTCF signal is decreased also in the MYC promoter in the absence of Dox, and not only at the STITCH region. A zoom-in and quantification of ChIP-seq experiments (peak calling, integrated intensities of signals) should be provided, and the results should be discussed accordingly.

10) Along the same line, in Figure 6 a crucial missing information is how CTCF binding evolves in time at the STITCH cassette and at the MYC locus.

11) It is unclear what 'control' in Figure 7 refers to.

12) One very interesting observation is that MYC gains H3K27me3 upon STITCH insertion, which correlates with the observed level of insulation in the various mutants and with transcriptional activation/deactivation in time course experiments. However why is it so? If this happens as a consequence of physical insulation from the super enhancer, how do the authors interpret it? An alternative explanation is that PRC2 is recruited by sequences in the STITCH cassette, and helps repressing transcription. The delay observed between MYC deactivation/reactivation and the corresponding differences of H3K27me3 are not large enough to exclude this second hypothesis. Based on the experiment shown in Figure 7J it cannot be excluded that H3K27me3 levels are unchanged upon treatment with EPZ, in the absence of a carefully quantified ChIP experiment.

[Editors' note: further revisions were suggested prior to acceptance, as described below.]

Thank you for submitting your article "Controlling gene activation by enhancers through a drug-inducible topological insulator" for consideration by *eLife*. Your article has been reviewed by Jessica Tyler as the Senior Editor, a Reviewing Editor, and two reviewers. The reviewers have opted to remain anonymous.

The reviewers have discussed the reviews with one another and the Reviewing Editor has drafted this decision to help you prepare a revised submission.

Summary:

This study describes development of a inducible insulator cassette, STITCH, that can act as boundary. This can be a very valuable tool to dissect rules of promoter – enhancer communication, and chromosome folding through mechanisms such as loop extrusion.

We request that the authors address the following issues.

Essential revisions:

1) The authors were requested to put the structural changes induced by STITCH in the context of the overall CTCF binding patterns in the MYC region. They addressed this point on the one hand by performing new 4C experiments using the STITCH sequence as a viewpoint. These experiments, now in Figure 1, unfortunately do not seem to reveal how the various endogenous CTCF sites could be used to make new connections with the ectopic STITCH cassette and even a re-analysis of the 4C data with the MYC promoter as a viewpoint are inconclusive. The authors conclude vaguely that "It might be extrapolated from these previous results that there are not very specific endogenous regions that singly form loops with STITCH to organize the conformational changes induced by STITCH". The interpretation of the data in the rebuttal is also highly speculative and does really not address the reviewers' request that structural changes are evaluated in the light of a more global view of chromosome contacts such as the one provided by Hi-C data. In 4C it is always hard to detect loop extrusion-associated structural features such as loops and stripes (or flares), and it is not surprising that specific connections between CTCF sites might be missed without performing matched Hi-C or 5C experiments. The authors need to at least these limitations of their 4C-based analyses.

2) In the new Figure 8, new experiments are provided to support the applicability of STITCH to additional contexts. However, the data do not appear to fully support the conclusion that STITCH-mediated modifications of chromosome interactions are at the basis of the observed transcriptional effects. First, 4C experiments in panel F do not allow to conclude in any manner that STITCH alters conformation at the targeted allele. Certainly, the presence of the wild-type allele confounds the readout, but even considering this, the fluctuating small-% differences observed do not seem to be robust (also, information on replicates is not provided unless I am mistaken). The right experiment to address this point would have been to perform 4C from the STITCH cassette itself, which would allow to detect mainly contacts within the mutant allele and which is apparently technically possible given that similar experiments are reported in the new version of Figure 1. It would be important to add such 4C analysis.

Second, it appears that NEUROG2 downregulation is only shown for one population of cells following piggyBac-mediated insertion of the TetR-KRAB transgene. It is unclear whether these results would hold true if the transposition of the transgene is repeated in independent experiments. Given that piggyBac insertions typically occur in multiple genomic locations simultaneously in every cell, a possibility that cannot be excluded is that the transcriptional effect on NEUROG2 is a secondary effect of mis-regulation of one or more upstream genes that are accidentally targeted by the transposon. This should be discussed.

3) Introduction and Discussion section: the authors seem to confuse several concepts of loop extrusion, insulation and contact domains. Insulation by CTCF is the result of its ability to block loop extrusion. Insulation does not necessarily involve CTCF sites to loop with each other (in fact such loops barely insulate). Insulation and contact domains can occur even when interactions are not strictly divergent at boundaries: insulation can occur in only one direction when CTCF sites are all in the same direction. Such unidirectional insulation can demarcate contact domains. The authors are asked to consider these issues when revising the Introduction and Discussion section.

4) All reviewers found the manuscript extremely difficult to read. Please do not use track changes. The manuscript should be carefully edited.

5) All plasmids should be made available through AddGene.

---

## [Author Response]

Summary:In this paper, the authors present a new tool to modify chromosome structure and enhancer-promoter interactions. The major advance over previously identified insulators is that this tool (STITCH) is designed to include not only divergent tandem arrays of CTCF sites to create a boundary, but also interspersed tetO arrays to allow inducible regulation of CTCF binding. The tetO arrays are added between the CTCF sites such that with the addition of a tetR-KRAB transgene to the cell line, CTCF binding can be disrupted by inducing heterochromatin formation at STITCH in a doxycycline repressible manner. Disruption of CTCF binding to STITCH leads to increased interactions and enhancer-promoter contacts across the STITCH insertion site, and can modify gene expression. Therefore, STITCH is an inducible insulating element, which would be a broadly useful tool in the field of chromosome structure and beyond, such as in genome editing applications.Essential revisions:1) Given that STITCH is presented as a tool, all reviewers felt that the approach should be used for 2-3 case studies, ideally in at least 2 different cell types. This will help show how generally applicable the tool is and how robust the results are.

We agree that generality as a tool is very important for readers. As suggested, we newly tested the functionality of STITCH at another locus, namely near *NEUROG2*, in human iPSCs and neural progenitor cells (NPCs). Thanks to the suggestions, we could now include a discussion regarding this issue in the manuscript, as explained below.

Firstly, the previous studies have shown that the TZ at the mouse *Tfap2c-Bmp7* locus is constantly bound by CTCF and organize chromatin contacts in various tissues (Tsujimura et al., 2015; 2018). In the present study, we show that the binding elements extracted from the TZ are also bound by CTCF and organize the chromatin conformation as a reconstituted cassette (STITCH) at a different locus (*MYC*) in a different species (human). These results well argue that STITCH, as well as the TZ, should be able to control chromatin contacts in various contexts through the binding of CTCF.

To further show the generality of the STITCH function, we inserted STITCH near *NEUROG2* in human iPSCs and integrated the tetR-KRAB transgene into the cells (Figure 8). Then, we differentiated the iPSCs to NPCs where *NEUROG2* is expressed. Our results clearly show that STITCH at this position also recruits CTCF and blocks the chromatin contact and that the KRAB induction again controls the CTCF binding and the chromatin in a drug-dependent manner in both cell types. Further, our data show that the expression is significantly down-regulated by STITCH in differentiating NPCs on day 4. We believe these results well support that the functionality of STITCH is quite robust and encourage researchers to apply the system for their researches.

Also, many other studies, as pointed out by the comment of the Reviewer#1 below, have already shown that inserting CTCF binding sequences insulate gene-enhancer interaction. Therefore, we firmly think that STITCH should be applicable to many genomic and cellular contexts, as presented in this study. However, as discussed below in our response to essential revision 6 and the revised manuscript, and also as remarked by reviewer#2, it is still quite elusive how CTCF manages this insulation process. In this sense, to fully describe the generality of the system, we may still need to wait for an accumulation of knowledge in the CTCF function.

2) All reviewers felt that the PCA analysis was confusing and did not make a critical contribution. Please thoroughly revise the corresponding text and figures, or remove this analysis from the manuscript.

We wish to apologize for not having explained well the methodology in our previous manuscript. Based on the reviewers' comments, we thoroughly simplified and revised both texts and figures. We also add the R codes that we used to analyze the data. So it should be now much more comprehensible than the previous version.

In brief, we think that applying PCA to analyze 4C-seq is a simple and powerful approach to extract crucial information from the data. Our PCA could, in fact, describe the preferential contact of *MYC* with the super-enhancer region more than with other non-enhancer regions in the vicinity in the absence of the STITCH insertion. As discussed in the following, we believe this finding should contribute to understanding how CTCF regulates the gene-enhancer interaction. Therefore, we would like to insist that this part is worth being kept and reported broadly to the community.

Improved parts in the revision

First, to improve the readability, we mainly revised the following four points:

1) We improved the appearance of PCA plots. We now use consistent colors and shapes to plot each allele in different panels, as suggested by Reviewer#1. We illustrate CTCF configurations nearby the plots to better compare the results. We also add interpretation of the PCA plots so our text should become more comprehensible while referencing the figure panels.

2) We simplified the organization of this PCA part and removed four panels in total from the main and supplementary figures.

3) We now explain how component loadings are calculated and what they mean, while providing the R code that we used to calculate the values and make the plots.

4) We thoroughly rewrote the texts to make what we think PCA could provide more precise and understandable.

Applying PCA to 4C-seq

Next, we would like to discuss what PCA provides. PCA allows us to compare 4C-seq data from multiple conditions (alleles) at once and at the same time to extract genomic regions that characteristically change contact patterns in correlation to the conditions.

Plotting the PC1 and PC2 values of each sample as a PCA plot illustrates how much the different alleles affect the contact profiles in comparison with each other (Figure 3C). If different alleles are arranged on the plot according to compositions of CTCF binding sequences, it means that the compositions of the CTCF binding sites are the major drivers to change the contact profiles within a given region. For example, in Figure 3C, the "non-blocking" alleles, namely WT(Hap) and del(L1-R3), show the lowest values of PC1, while the other alleles with CTCF binding sequences show higher values of PC1, illustrating that the largest variance among the samples is most likely due to the arrangements of the CTCF binding elements.

Then the component loadings of PC1 show which genomic regions (bins) are more loaded onto the PC1 values. The component loadings of PC1 are calculated as the product of the eigenvector and the square root of the eigenvalue of PC1. Component loadings correspond to the correlative coefficients between the component values of PC1 and the original frequency values of the binned genomic region. For example, in Figure 3D, the component loadings of the bins in the left 900-kb region are all nearly 1, showing that the contact frequencies of these bins well correlate positively with the PC1 values. It means that the samples with higher PC1 values have higher contact frequencies with the 900-kb region, which is, in fact, the case (Figure 3B). On the other hand, the bins in the right region, particularly the immediate 570-kb region, have component loadings of minus values, meaning that the contact frequencies of these bins correlate negatively with the PC1 values. In fact, the alleles with lower values of PC1 show higher contact frequencies with the right 570-kb region (Figure 3B). We think these data show a proof-of-concept that PCA can collectively represent the 4C contact patterns of multiple samples as a simple and powerful analytical method.

What PCA adds to this work

The PCA plot in Figure 3C indicates that the arrangements of CTCF differentiate the samples by two different effects: One is the blocking effect by the presence of CTCF against the non-blocking alleles; the other is the directionality effect segregating the CTCF arrays pointing to leftward from those to rightward. To disentangle the intermingled effects, we next performed PCA for subsets of the alleles. First, we removed the non-blocking alleles, namely WT(Hap) and del(L1-R3), leaving only the STITCH and the mutant alleles (Figure 3—figure supplement 4). This subset should reduce the blocking versus non-blocking effect. In fact, in Figure 3—figure supplement 4, the segregation is mainly seen between the leftward and the rightward alleles. We think this segregation well represents the differences of the directionality scores from VP-MYC1/2 in different alleles, which we now add as Figure 3—figure supplement 3B in this revision (for the directionality score, please refer to our response to essential revision 6 and the revised manuscript).

Next, we only used the non-blocking alleles and the non-directional blocking alleles, namely the original STITCH and inv(L1-R3), to reduce the effect of the directionality and enhance the blocking effect. Then the two groups are segregated along PC1 (Figure 3E). Also, the component loadings show complete switching at the insertion site of STITCH, which, of course, makes sense (Figure 3F).

Then, we asked if the left 900-kb or the right 600-kb regions contain internal regions that specifically associate with *MYC* depending on the absence and presence of CTCF. For this, we performed PCA using the contact frequencies only within the left or right regions (Figure 3G, H). Note that the contact frequencies are re-normalized only among the bins subject to PCA. If all the bins in either left or right side change the contact frequency more or less equally in response to CTCF, which perhaps might be a predominant expectation from the current understanding, there should not be clear segregation in PCA by the compositions of CTCF. In fact, we see no clear segregation in PCA for the left side region, suggesting that the binned regions in the left 900-kb region behave more or less equally with each other (Figure 3G).

However, for the right 600-kb region, this was not the case (Figure 3H). The plot shows segregation between blocking and non-blocking alleles. Most notably, the component-loading plot shows that the non-blocking alleles are more associated with the region corresponding to the super-enhancer. This means that STITCH does not just block the contact as a whole beyond the insertion site. STITCH instead seems to interrupt the interaction of *MYC* with the super-enhancer actively. As discussed later in our response to essential revision 6, we think this finding is important to understand how CTCF insulates the gene-enhancer interaction. Without PCA, at least for us, it would have been impossible to describe this feature. Thus, PCA can (and in fact did) extract valuable information from the 4C-seq data.

3) It is critical to show whether TetR binding alone, without the KRAB domain, affects CTCF binding and/or CTCF-mediated insulation.

Thanks for raising this point. We agree that this control is important. We newly performed nChIP-qPCR to confirm that CTCF binds to STITCH when the tetR-3xFLAG-HA was induced (Figure 5—figure supplement 1J). This result strongly suggests that KRAB induction, beyond simple binding of protein, is essential to disable the insulation by STITCH. Unfortunately, we failed to see significant enrichment of tetR-3xFLAG-HA at STITCH by several protocols of ChIP. We tried various conditions with help from Dr. Tomohiko Akiyama (Keio University School of Medicine), who has expertise in doing ChIP against FLAG-tagged transcription factors (Akiyama et al., 2015; Yukawa et al., 2014), but we could not establish a protocol to detect the binding of tetR at STITCH in the timeframe of the revision. We are aware of previous reports showing binding of tetR at tetO using similar tags (Moussa et al., 2019; Pourfarzad et al., 2013; Ragunathan et al., 2015). We think that, when compared to these studies, the difficulty we encountered might be attributable to the fact that our STITCH includes a total of only four elements of tetO that are sparsely arranged with each other within the 1.3-kb length of STITCH, while those successful studies used a clustered array of 7x or 10x tetO sequences.

Instead, we further tested if the STITCH at the same location but with the puromycin resistant cassette (*PURO^r^*), which is the one before Cre recombination was carried out, would be bound by CTCF. Here, the *PURO^r^* is transcribed from the cassette, so there should be binding of certain proteins at STITCH. Our nChIP-qPCR shows that CTCF still binds to STITCH at similar levels, and *MYC* is repressed (Figure 5—figure supplement 1K, L). These results strongly support that the KRAB induction is essential to expel the CTCF binding. We believe these data clarify the concerns raised by the reviewers.

4) The H3K27 ChIP experiments need to be repeated to provide better statistics to support the conclusion that changes in the levels of this mark can be accurately quantified. There are concerns this data set is not of high quality. Also, in Figure 4C-D positive and negative controls should be added.

To clarify the concerns in the revised manuscript, we first re-analyzed the ChIP-seq data (now Figure 4A-E). We also added positive and negative controls in Figure 4C-D (now Figure 4F-G). Further, we repeated the experiments in Figure 6F, G, Figure 7A-F, and obtained the same conclusion.

Reanalysis of nChIP-seq for H3K4me3, H3K27me3, and H3K27ac (Figure 4A-E)

Firstly, to demonstrate that the H3K4me3 and H3K27me3 profiles only changed at *MYC*, but not other loci, we presented magnified views of the profiles at *MYC, T* (repressive), *ACTB* (active), *HOXD13* (repressive), and *DPPA4* (repressive) loci (Figure 4A, Figure 4—figure supplement 4C). These panels well illustrate that while the epigenetic states at *MYC* were considerably changed, those at the other loci are unchanged at all. We also think that the magnified views well show that the enrichment we detect is well more than the backgrounds. Also, these figures should support that our assays are good enough to distinguish changed and unchanged epigenetic states in different conditions. Of note, the *MYC* locus lacks one allele, so we adjusted the count ranges to show in the tracks accordingly (0-6 for MYC, but 0-12 for others, as for H3K27me3).

Next, to further provide more statistic support to our observation, we calculated fold changes of read counts over peaks in genome-wide in STITCH^+^30kb against Hap. As shown in Figure 4C-E, the H3K4me3 and H3K27me3 peaks are ranked as one of the top peaks showing the most extensive changes. By contrast, the same analysis shows that the H3K27ac peaks detected in the super-enhancer/*PVT1* region did not change much by the STITCH insertion. We think these results adequately support our conclusion that the epigenetic states were only altered at *MYC*.

Positive and negative controls in Figure 4F-G

We newly quantified the enrichment at *DPPA4* and *T* as positive and negative controls for H3K4me3, respectively (Figure 4F). Similarly, the enrichment at *HOXD13* and *ACTB* was added as positive and negative controls, respectively, for H3K27me3 (Figure 4G).

Repeated analysis of the time-course change of H3K4me3 and H3K27me3 at *MYC* (Figure 6F, G, Figure 7A-F)

We repeated the time-course experiments in Figure 6F, G, and Figure 7A-F. For Figure 6F, G, in this revision, we extended the timepoint up to 48 hours after the addition or removal of DOX, to compare the 24hour point with the later point. As we have shown in the previous manuscript, we could see again that the change of the H3K4me3 level is rapid, while the change of the H3K27me3 level is slow.

In Figure 7, we again sampled cells at 24 and 72 hours after DOX addition/removal together with DOX plus/minus controls, and examined the histone mark levels. Then again, we could reproduce our previous results that 24 hours is not enough for H3K27me3 to be entirely switched. Thus, our finding that H3K27me3 level only follows the gene expression change should be very robust.

5) The transcriptional analysis In Figure 2 could be improved. The cutoffs used to identify differentially expressed genes with DESeq2 are very loose (adjusted p-value = 0.1, no limitation is imposed on fold changes). We suggest to repeat the analysis with more than one clone per condition.

Thanks a lot for the suggestions. In the previous version of our manuscript, we used the threshold (p-adjusted < 0.1), as it was used for demonstration and benchmarking in the paper reporting the development of DESeq2 (Love et al., 2014). In this revision, we re-analyzed the data with a tighter threshold (p-adjusted < 0.05, log2-fold-change > 0.5). This setting called less number of differentially expressed genes (Figure 2B-D, Figure 2—figure supplement 2B, C). Nonetheless, the GO enrichment analysis among the commonly down-regulated genes still shows enrichment of categories such as cholesterol synthesis and rRNA processing, as we described in the previous version (Figure 2—figure supplement 2B, C).

As everyone might agree, deciding on the threshold is a little bit arbitrary. In this sense, we found Gene Set Enrichment Analysis (GSEA) is attractive, because this algorithm does not impose threshold setting (Subramanian et al., 2005). Also, as pointed out by reviewer#1's comment, the primary aim of this analysis should be on whether the target of *MYC* is affected by the mutations or not. We found that the "HALLMARK 50" from MSigDB includes categories of *MYC* targets (Liberzon et al., 2015). Therefore, in this revised manuscript, we newly performed GSEA against HALLMARK 50. Figure 2E and F show that the analysis detected quite strong enrichment of the *MYC* target categories in down-regulated genes in both STITCH^+^30kb and del(30-440). The other enriched categories are shared between the two mutations, suggesting that the tendency of the transcriptional change is quite similar between them (Figure 2E, F). Moreover, many of the enriched categories have already been suggested to be subject to *MYC* function in various cell types. Thus, the reanalysis could show that *MYC* was effectively down-regulated by the mutations to affect the expression of its target genes that seem to be shared in many cell types.

Regarding the suggestions to use different clones per condition, the three replicates that we used were actually from three different (sub-)clones, as described in the Materials and methods section in the previous manuscript. We now also indicate it in the corresponding part in the revised Results section. As indicated here, the used clones were: as for Hap the parental clone and two isolated sub-clones derived from the parental one; as for STITCH^+^30kb and del(30-440), three different clones isolated upon the final Cre recombination step, respectively. To analyze completely independent clones for each should have been ideal for fully controlling the clonal variations. However, it would be pragmatically very complicated to do so here, because introducing these mutations needs quite a few steps of cloning.

We still think our results are valid enough for the following reasons. Firstly, we did obtain different clones at one step for each condition, so much of the cloning effects should have been well controlled. Secondly, we confirmed that the deletion of STITCH or the KRAB induction well recovered the *MYC* expression level back to the normal level (Figure 3A and Figure 5C), so at least the repression of *MYC* observed in this study should not be attributed to the clonal difference. Therefore, we do not think that our data are suffering much from variations due to the cloning processes.

6) Please extend the discussion about the relationship between structural changes induced by STITCH and potential new loops induced by the ectopic CTCF sites in the light of the current understanding of CTCF orientation (and loop extrusion?). Reciprocal 4C viewpoints based on endogenous CTCF sites could also help clarifying this matter.

Thanks a lot for this suggestion. In the previous version, the 4C-seq was only from viewpoints at *MYC*. So, we could not discuss much how STITCH itself was involved in the chromatin structure. In this revised manuscript, we newly performed 4C-seq from the flanking sites of STITCH as viewpoints, as we thought it should be more direct to discuss this matter (now Figure 3—figure supplement 3). We also put the CTCF binding sites and their orientations, as well as the endogenous domain structures along some of the 4C-seq tracks (Figure 1B, C, Figure 3—figure supplement 3A). With these data, we extended the discussion in the context of CTCF loops. In summary, we cannot conclude that any specific loops or loop/contact domains are essential for STITCH insulation. We instead claim that considering a model linking the functionality of CTCF (and loop extrusion) directly to the disruption of gene-enhancer interaction should be required, as explained below.

Formation of loops?

We visually inspected the new 4C-seq plots and found that there seem to be several peak-like bumps in some of these plots (Figure 3—figure supplement 3). These bumps may represent new loops that STITCH creates. However, we cannot tell if there is anything special that would be regulating the insulation process, simply because the bumps are not very striking. To clarify if there is a newly formed loop that would be important, it is necessary to delete endogenous CTCF binding sites (or whatever) that are involved in this loop formation, as suggested by reviewer#3. However, we could not perform this experiment for this revision, firstly because the new 4C-seq plots did not indicate encouraging peaks to test and secondly because it would be certainly impossible to finish the analysis in a reasonable timeframe here. Also, please note that the deletion of one region might only result in reestablishment of another loop with some remaining regions, possibly the CTCF binding sites next to the deleted one, which would make the experimental design very complicated.

However, we would like to point out that we had already performed analogous experiments in our previous study (Tsujimura et al., 2018). In Figure 5 of this paper, we analyzed contact profiles of *Tfap2c*, which carries CTCF binding sites nearby, upon several mutations of the TZ. First, the inversion experiment of *Tfap2c* indicated that the directional folding around *Tfap2c* greatly depends on the locally associated CTCF binding sites. On the other hand, the mutations around the TZ did not affect the folding directionality of *Tfap2c* almost at all. This result suggests that the conformational change imposed by a CTCF binding site (near *Tfap2c*) does not rely on looping with another CTCF binding site located distantly (around the TZ). Also, the recent paper, which was kindly introduced by reviewer#2, deleted CTCF binding sites that bridge loops to establish the *Shh* loop domain. Interestingly, it did not result in an appreciable change of the *Shh* expression, strongly suggesting that a specific loop may not be relevant to genome regulation (Williamson et al., 2019).

Based on these results, we now have a view that discussing the formation of new stable loops with specific endogenous sites might not be directly fruitful to gain insights into how CTCF insulates the gene-enhancer interaction. In the revised manuscript, we describe these thoughts along with the presentation of Figure 3—figure supplement 3.

Of important note, we believe that loops anchoring the insulating CTCF sites should be rather crucial for the disruption of the gene-enhancer interaction, as explained in the following. However, according to what we think, it might not matter much whether the loops are formed stably with specific CTCF binding sites or dynamically/promiscuously with any other sites, as long as they are anchored at the insulating CTCF site (now Figure 9E). In this sense, we do NOT think that the current understanding of the insulation-by-looping model should be excluded.

Loop/contact domains?

We also have extended discussion regarding the current dogma of enhancer regulation by contact domains. As discussed right above, our methodology could not identify specific loops created by STITCH. Therefore, this study is not able to tell the overall re-organization of domains induced by STITCH. As suggested by reviewer#1, Hi-C or 5C should be required to do so. However, domains are basically defined by "boundaries" that exhibit diverging directionality of chromatin folding. So, it was possible for us to analyze the transition of folding directionality around the insertion site of STITCH (Figure 3—figure supplement 3B). We found that the divergence of folding directionality is not evident across STITCH, particularly in the delL allele, suggesting that the formation of domains or establishment of domain boundaries is not prerequisite for the STITCH insulation (Figure 3—figure supplement 3B).

We are quite aware that many previous studies (including (Tsujimura et al., 2015)) supports the currently recognized model that contact domains restrict enhancer targets (Schoenfelder and Fraser, 2019), as indicated by reviewer #1. However, we now wonder if this is really the case. We even find that questioning or neglecting this dogma seems helpful to interpret the data presented in this study and others. So please allow us to explain what we think regarding this idea here.

As far as we understand, what have been shown so far are primarily the followings:

1) Boundaries/CTCF binding sites exhibit the orientation-dependent directionality of chromatin folding most likely through the loop extrusion.

2) Contact domains emerge as a consequence of the directional folding (or the loop formation) by the function of the boundaries/CTCF binding sites (and cohesin loops).

3) Enhancer allocation is mostly restricted within contact domains.

4) Genetic manipulation of boundaries/CTCF binding sites (but not directly of contact domains) alters the gene-enhancer interaction.

These data underlie the current understanding that CTCF creates (boundaries of) domains, which then restrict the enhancer allocation. However, it should be questioned if taking the contact domains into account is genuinely essential to explain the enhancer regulation because there do not seem studies directly showing the causative role of the domains *per se*, but not the CTCF/loop extrusion. The above data should also corroborate another idea that CTCF/loop extrusion limits the enhancer targets and at the same time, establish contact domains (or chromatin organization that can be called contact domains). What has been shown so far is the only correlation between the presence of contact domains and enhancer regulation through analysis of CTCF binding sites.

For this reason, we had put loop extrusion and enhancer blocking as separate mechanisms in the Introduction. As our text was pointed out as unclear by reviewer #1, we have now added more of these explanations in the revised manuscript for it.

From this perspective, it should not be surprising that we did not find evidence of domain boundaries at STITCH from the analysis of folding directionality. Of course, absence of evidence is not evidence of absence. However, at least our data does not encourage the domain-centric view.

Besides, we would like to point out some data of this work and other studies that the regulation-by-domain model does not explain well. First, (Bintu et al., 2018) has shown that contact domains represent averaged projection of various domain-like structures. This means that CTCF/cohesin allows domain-like association across domain boundaries. Then, why does the gene-enhancer interaction not take place across boundaries?

Similarly, we (and surely many other studies), in fact, detect inter-domain contacts between genes and enhancers. For example, our 4C-seq results show that STITCH reduces the contacts with the enhancer region only by half. However, the change in the expression level is more than 20 times. We think this discrepancy is quite puzzling. Is there really any difference between inter-domain and intra-domain contacts? Is there any evidence for it? Then what was the domain-like structure across the boundary observed by DNA FISH (Bintu et al., 2018)?

CTCF-centric view

We think the most straightforward interpretation of these observations should be that CTCF/loop extrusion somehow interrupts the gene-enhancer interaction when inserted in between, regardless of the formation of loop/contact domains. In this sense, we think our PCA has provided valuable insights. The analysis shows that the presence of CTCF more pronouncedly decreases the contacts of *MYC* with the enhancer region more than the other non-enhancer regions around. Based on this finding, we have proposed two models for enhancer regulation by CTCF, as illustrated in Figure 9D-E, which is added for this revision. One is that the gene-enhancer interaction is boosted upon an increase of overall contact frequency, possibly by an enhanced phase-separation process (Figure 9D). The other is that (either dynamic or stable) loop extrusion anchored at the insulating CTCF efficiently disrupts the gene-enhancer interaction (Figure 9E). Of course, these models are elusive. However, we believe they are reasonable enough to be proposed in this manuscript. Along this line, it should be pointed out that the regulation of gene-enhancer interaction by CTCF is not fully understood. Therefore, as stated in our response to the essential revision 1, the generality of STITCH cannot be described entirely for the moment.

7) Upon publication we feel it is critical that the tools are made available to the community (e.g. putting plasmids on AddGene).

Thanks a lot for this suggestion. It will be our pleasure if many people use our system for their researches. Accordingly, we have deposited the three plasmids carrying STITCH, tetR-KRAB-2A-Puro, and tetR-3xFLAG-HA-2A-Puro, respectively, to AddGene. They should become soon available to the broad community from them. Please also see the Key Resources Table in the revised manuscript.

Reviewer #1:The authors overstate the novelty of their results with respect to insulating elements. Insulating elements have been previously identified, and usually contain CTCF sites, similar to the STITCH sequence (Bell et al., (1999); Liao et al., (2018); Emery, (2011)). This should be more adequately referenced and introduced.

Thanks for this comment. It was not our intention to claim novelty for it. We have now introduced some of these studies in the Introduction to explain previous works regarding the CTCF insulation.

In addition, it has already been shown that CTCF sites that form TAD boundaries will block enhancer-promoter interactions, yet this is presented as a novel result (subsection “Titrating blocking activity of STITCH by serial mutations of the CTCF binding sites”) (Hou et al., (2008); Guo et al., (2015); Braikia et al., (2017)).

Thanks a lot for bringing our attention to these previous studies in this respect. Notably, we realized that Hou et al., (2008) in Figure 5A and Figure S4 similarly claims that the CTCF insertion specifically reduces the gene-enhancer interaction, but not the contacts with the other regions in the vicinity. We newly mentioned about their finding in this part.

The conclusion that the enhancer blocking activity and chromosome interaction activity are due to separate mechanisms is not sufficiently explained or supported. It is unclear why loop extrusion and enhancer blocking are introduced as separate mechanisms (Introduction), when the current understanding in the field is that enhancer blocking by CTCF sites is likely due to the creation of new TAD boundaries, which are formed by loop extrusion being blocked by CTCF sites (Recently reviewed in Schoenfelder and Fraser, (2019)).

Please see our response to the essential revision 6. We have tried to explain this better in the revised Introduction.

Similarly, there appears to be confusion about the various mechanisms of long-range chromatin interactions in the Discussion section. Subsection “Mechanism of the STITCH insulation and its control by heterochromatin induction”, second papragraph is very confusing. In the current work CTCF is removed, from the locus, but cohesin is not. Therefore, there is now unblocked loop extrusion throughout the locus. Given that the interaction between the enhancer and MYC is blocked by CTCF, it is much more likely that the increased interaction between the super-enhancer and MYC when CTCF is removed is driven by loop extrusion.

Depletion of cohesin enhances compartmentalization (Rao et al., 2017; Schwarzer et al., 2017). Therefore, we think it is less likely that the unblocked loop extrusion is required to establish the gene-enhancer interaction here. Rao et al., (2017) has shown that the links between super-enhancers are established even between different chromosomes, suggesting this can be achieved without any loop extrusion. We have mentioned this in the revised discussion. We think it is more likely that the loop extrusion interferes with the gene-enhancer interaction when CTCF is present (Figure 9), as explained in our response to the essential revision 6.

Technical details and biological conclusions are not adequately explained in the text throughout this manuscript. The manuscript would benefit from improving the explanations of why specific analyses are used, and what the results signify biologically, beyond just stating the observations. In particular, the explanations of PCA analysis, component loading plots, and power law scaling between gene expression and 4C-seq contact frequency should be clarified. As it is presented now it is entirely unclear what the PCA analysis really adds. Additional technical details of the computational methods used to analyze the sequencing data is also needed, preferably an online repository for the code used to generate the plots and run the statistical tests should be included with the manuscript.

Thanks a lot for the suggestion. We have tried to make the aims, the methodology, and the biological conclusions clearer throughout the revised manuscript. Particularly, for PCA, we now have added an extensive explanation of the methodology and clarified the story. We also put the R codes as supplementary files.

While 4C-seq is a useful technique for studying the specific interactions between MYC and surrounding loci, it would be beneficial to also compare this to an all-by-all or many-by-many chromosome conformation capture method such as Hi-C or 5C to show the endogenous organization of this region. Putting the STITCH insertion in the context of the landscape of genomic architecture (where are TADs or compartments found in this region?) would strengthen the manuscript and might help to understand how the different directionalities of the CTCF motifs in STITCH are working, as the current explanation is unclear.

Thanks for the comment. Now we have added Hi-C data and TADs organization in human ESCs (Dixon et al., 2015) to Figure 1B to illustrate how the domains are organized around the locus. We see that *MYC* is located inside a vast domain. Accordingly, we have introduced this data at the beginning of the Results section. Moreover, we newly performed 4C-seq from flanking regions of the insertion as viewpoints to see how STITCH affects the organization, as explained in our response to the essential revision 6. Also as explained in the response, we believe that we could provide a reasonable model of how STITCH insulates gene expression.

In addition, it would strengthen the manuscript to compare the contact frequency changes in the STITCH mutants at the MYC locus to changes in the endogenous TZ locus with similar modifications from previous publications by these authors, to determine how variable this behavior is at different genomic loci.

Thanks a lot for the suggestion. It is very true. As mentioned above, we have performed new 4C-seq experiments and analyzed the directionality of chromatin folding around the STITCH insertion, as was performed in our previous work (Tsujimura et al., 2018). We did similarly observe skewed transition of the folding directionality across inserted STITCH as in the endogenous TZ. This strongly suggests that the CTCF binding sites behave similarly in both endogenous and synthetic contexts. Interestingly, however, the divergence of folding directionality around STITCH was not evident in contrast to the TZ. As explained in our response to the essential revision 6, this may suggest that the creation of diverging directionality of chromatin folding is not an essential prerequisite for the insulation of gene-enhancer interaction.

Reviewer #2:The role of CTCF, cohesin, TADs and 3D genome organization in regulating gene expression and enhancer-promoter (E-P) contacts is currently being intensely studied. In particular, whether CTCF sites and TADs really regulate E-P contacts and gene expression has recently become controversial, with some studies claiming that CTCF and TADs have no role in regulating gene expression (e.g. see https://www.biorxiv.org/content/10.1101/609941v1).Most studies (including the preprint above) have taken a "deletion" approach to this issue: take a natural E-P pair and TAD and then go in and start deleting or inverting etc. CTCF binding sites and see how gene expression is affected. What is nice about this study is that they take the opposite approach – a kind of "addition" approach. They add the STICH array of CTCF binding sites to the MYC gene at different locations and see how contact frequency (4C) and gene expression (qPCR) of MYC in hiPSCs is affected. The 2 most important points in my opinion are:1) CTCF sites really can block E-P contact and strongly (~20-fold) affect gene expression. At least for the MYC gene.2) Histone modifications can be used to turn ON and OFF CTCF insulation with quite high temporal resolution.I believe the authors get about as close to causality as is realistic with the current tools of molecular biology, which is nice.Beyond some important but addressable concerns (poor writing, at times confusing figures and presentation, occasionally poor referencing, tool availability), my major concern is this: The authors report STICH as a tool. 2 key features in a tool that are desirable to have are: (1) robustness and (2) generality. But because the authors only apply STICH to one locus (MYC) in one cell type, we cannot really tell if STICH is likely to block E-P contacts in general and robustly in many other loci and other cell types. The impact of STICH would have been greatly increased if the authors could have applied it to 2-3 case studies, ideally in at least 2 different cell types.So overall, I believe this is a nice contribution with some really important insights, but that the general interest and impact could have been substantially improved if the authors had applied STICH to at least 2-3 different systems and if they can improve their presentation.

Thanks a lot for this comment. Regarding the generality and robustness of STITCH, please see our response to essential revision 1.

Specific issues:Writing: the paper is for the most part reasonably written, but there are at least >25 cases of poor English and/or syntax/grammar issues. This is too many for a reviewer to fix and I suggest that the authors go through and clean up these issues.

We apologize for our poor language. We have tried to correct mistakes and to improve readability.

Discuss results in context: First sentence of the Abstract and in the Introduction suggest that "regulation of gene-enhancer interaction is better understood,". I would argue that this is not true. In fact, recently several people (e.g. https://www.biorxiv.org/content/10.1101/609941v1) have begun arguing that CTCF plays essentially no role in the regulation of gene expression.

We agree with this comment. While recent studies have provided valuable insights into the genome regulation, there remain still many puzzling phenomena. We changed the sentence as "While regulation of gene-enhancer interaction is intensively studied,".

The fact that the authors see such clear results on MYC, in my opinion only increases the impact and value of this study. Therefore, it would be nice if the authors could discuss their MYC result a little bit more clearly in the Discussion section in the context of the many recent studies arguing that CTCF plays no or only a minor role in regulating E-P contacts and gene expression.

Thanks for this suggestion. We have now extended our Discussion section. For the details, please see our response to essential revision 6.

Key resources should be available: First of all, my apologies if the authors already did this. But I tried to find this information and was unable to. The authors must put the key STICH plasmids on AddGene for the community, since the value of a tool is largely derived from it being readily accessible for the community. The DNA sequences of STICH must also be available with the paper. I could not find the DNA sequences of the full STICH sequence nor could I find the sequences of the specific CTCF binding sites. These must be available.

Thanks a lot for this suggestion. We have now deposited the plasmids to AddGene. Please see our response to essential revision 7. The DNA sequences of STITCH were provided in Supplementary file 2. The sequences should also become available from AddGene soon.

RNA-seq Results: The RNA-Seq studies in Figure 2 were really nice. But I could not understand why the STICH +30kb cell line would have ~2-3x more deregulated genes than the del(30-440) cell line. Although STICH is powerful, deleting the enhancer should still have a stronger effect on expression than just blocking it. Could the authors better explain this?

Thanks for the comment. We speculate that the deletion allowed contact of *MYC* with regions with some enhancer activity located further than the +440kb position (Figure 1E), which led to a slight upregulation of *MYC*. We now discuss this in the corresponding part of the Results section.

PCA-analysis: In Figure 3A-B, the analysis of how 4C reads in the different regions depend on the STICH construct was really nice. It was also very interesting to see the highly non-linear scaling between 4C contact frequency and gene expression (Figure 3I-J). Both of these are really important contributions in my opinion.

Thanks a lot for the comment. We thought that the feasibility of serial mutagenesis of CTCF binding sites could be an advantage of our system, so we tried to gain as much insight as possible from this.

But the PCA-analysis was extremely confusing and convoluted. I really tried to follow the text and the figures, but it was very difficult for me to understand what the point was. Aorund lines 200-250, the authors spend a lot of text and a huge number of figure panels on this, but I really could not understand it. My suggestion would be to remove all the figures and text pertaining to PCA or at least radically simplify the figures and the text to make it easier to understand. What is the major biological insight coming from this PCA analysis? What is component loading?

We again apologize for our inadequate explanation. We have revised this part thoroughly. Please see our response to essential revision 2.

Subsection “Insulation and deletion of the enhancer resulted in similar transcriptome profiles”: authors out-of-the-blue reference VP-MYC1 and VP-MYC2, without any figure REF. I could not understand this.

Thanks for the comment. We now add the figure references.

tetR-KRAB studies: The Tet-R KRAB studies were very nice. I may have missed prior studies, but to my knowledge this is the first clear and causal demonstration that histone modifications can turn OFF CTCF insulation. However, one control I was missing was a DNA-binding control. TetR-KRAB binding could disrupt CTCF binding and insulation through 2 ways: DNA-binding competition (e.g. TetR-binding outcompetes CTCF binding) or KRAB-deposition of histone modifications. I would have liked to see a control showing that TetR binding alone – without the KRAB-domain – does not affect CTCF-mediated insulation.But pretty neat to see that STICH insulation directly affects cell proliferation (subsection “Titrating blocking activity of STITCH by serial mutations of the CTCF binding sites”).

Thanks for the comments and suggestions. Please see our response to essential revision 3.

Figure 6. F and G have errors bars, but Figure 6 C, D, and E do not. Need to add errors bars to these.Otherwise, the time-course results were also pretty cool.

Thanks for the comments. We only performed n=1 experiments for Figure 6C, D, E as the objective here was to capture the time-course change of the system. Besides, the experiments, particularly for Figure 6D and E, give relatively robust results. Based on these observations in Figure 6, we newly performed n=3 experiments for the statistical test presented in Figure 7A-F. We made the numbers of replicates explicit in the revised manuscript.

Reviewer #3:The manuscript describes a strategy to modulate chromosomal contacts in the vicinity of the endogenous MYC gene in human iPS cells through the ectopic insertion of an array of CTCF sites. The approach (named STITCH) seems to be able to alter MYC transcription levels, which correlates with changes in interaction frequencies between the MYC locus and a super-enhancer region downstream. The authors further monitor chromatin states at the engineered locus upon the induction of H3K9 trimethylation by targeted recruitment of a KRAB domain at the STITCH cassette, which is shown to disrupt CTCF binding and restore wild-type chromosomal contacts. The authors conclude that CTCF-mediated modulation of chromosome interactions is the driver of transcriptional changes.The study is interesting and well designed, and has the potential to bring insight into how gene expression could be modulated by manipulating chromosome structure. However, it suffers from several major drawbacks that should be thoroughly addressed.

Thanks a lot for appreciating our work.

1) Many of the native ChIP-seq experiments in the manuscript are difficult to interpret and it is often difficult to agree with the authors on the changes they describe. The zoom level in all Figures is way too low to visually appreciate any local changes at the MYC locus, the STITCH cassette and the neighboring region. More importantly, some crucial experiments (notably the H3K27ac and H3K27me3 ChIP-seq reported in Figure 1, Figure 4 and Figure 4—figure supplement 1) appear to be strongly suboptimal. It is hard to imagine that a local increase of ~2 counts in a 0-6 range as reported in Figure 1 and Figure 4 really does correspond to a specific enrichment as opposed to technical noise.The authors should perform new H3K27ac/me3 ChIP-seq experiments, provide statistics to support the notion that changes in these chromatin marks can really be quantified and discuss their findings in the light of the new experiments. Crosslinking ChIP-seq would be a viable option in this context – in fact I found it quite unclear why native ChIP was required in this particular study.

Thanks for the comment. As the *MYC* locus was made haploid, the locus demanded more coverage than the rest of the genome, which might have made our ChIP data look as if compromised. However, we re-analyzed our data more quantitatively and now could provide statistical support for our conclusion (Figure 4C-E). Please see our response to essential revision 4 for more details. Also, we think our nChIP for H3K27me3 assays are good enough to discuss the following assays in Figure 5, Figure 6, Figure 7. In fact, we repeated H3K27me3 and H3K4me3 experiments in Figure 6F, G, and Figure 7A-F, and obtained the same conclusions. As indicated in Figure 4A, the background does not look very high when inspected in magnified views. It seems that the zoomed-out view of our tracks exaggerates the background levels.

ChIP with crosslinking in principle can enrich regions that are only indirectly associated with the chromatin mark through 3D association with regions that possess the mark genuinely, as exemplified in (Skene and Henikoff, 2017). Our work tries to challenge the chromatin organization with STITCH. Therefore, we think it should be more appropriate to utilize nChIP to discuss local chromatin states, as it avoids detecting pseudo-epigenetic changes due to the re-organization of chromatin conformation.

2) The correlation between MYC transcription levels and contact frequencies with the super-enhancer region (Figure 3) in mutant STITCH cell lines are interesting, and well supported by the large number of independent clones analyzed. Unfortunately, the structural changes induced by ectopic CTCF sites were not correlated with the position and orientation of endogenous CTCF sites. MYC itself is highly bound by CTCF, as can be more or less seen (again at regrettably low resolution) in Figure 5F. I would suggest the authors to thoroughly discuss the relationship between structural changes induced by STITCH and potential new loops induced by the ectopic CTCF sites in the light of the current understanding of CTCF orientation (and loop extrusion?). Reciprocal 4C viewpoints based on endogenous CTCF sites could also help clarifying this matter.

Thanks a lot for the suggestions. Please see our response to essential revision 6.

3) It is unclear how many copies of the STITCH cassette have been integrated at the MYC locus. The authors should provide evidence that a single insertion of 6 CTCF sites is actually responsible for the observed structural changes, as opposed to multiple tandem repeat insertions (especially since lipofection -and hence large amounts of DNA per cell- was used to generate the Cas9 assisted knock-in).

We agree that tandem repeat insertions could happen. We genotyped the insertion with PCR using several combinations of primers (Supplementary file 4). One of the primer pairs was designed to anneal to the flanking sites of the insertion, which tells the total length of the inserted DNA. With this, we could safely conclude that only one copy is integrated. Besides, we carry out Cre recombination after the targeting. We, of course, confirmed the correct integration at both sides of the insertion site. In this case, it was very improbable that we could obtain insertions with multiple copies afterward, because all the extra internal copies regardless of their orientations, which we confirmed are absent, should have been deleted out by the recombination.

4) The transcriptional analysis In Figure 2 could be improved. The cutoffs used to identify differentially expressed genes with DESeq2 are very loose (adjusted p-value = 0.1, no limitation is imposed on fold changes). In the absence of differential gene expression analysis on more than one STITCH and del(30-440) clones, it is difficult to assess what the >1000 genes detected as differentially expressed under these loose criteria actually represent. I would suggest to repeat the analysis including more than one clone per condition and using more stringent criteria (e.g. padj<0.01, |log2(FC)|>1) in order to identify mis-regulated genes more robustly and reliably. Also, a qPCR validation of significantly up- or down-regulated genes is missing.Finally, there is no explanation for the fact that the effect on transcription in the deletion mutant is smaller than in the STICH mutant. If the changes are indeed due to the insulation of the super-enhancer region from the MYC gene, then deletion of the super-enhancer region should lead to an even stronger effect on transcription.

Thanks a lot for the suggestions. Please see our response to essential revision 3 for the reanalysis that we performed. The RNA-seq results are well consistent with our qPCR assays for the *MYC* expression. Therefore, we think our RNA-seq data are valid. Regarding the difference between the STITCH insertion and the enhancer deletion, we add our speculation to the revised manuscript. Please see our response to reviewer #2.

5) It would be nice to prove that transcriptional changes in the STITCH and del(30-440) lines are really caused by downregulation of MYC, which could be done notably by overexpressing MYC and testing if normal expression programs are rescued.

Thanks for the suggestion. We agree that this rescue experiment should clarify the cause of the transcriptomic changes more. However, our (re-)analysis shows that the most significantly down-regulated gene groups are those of known *MYC* target genes. Therefore, the transcriptomic alteration observed here should be reasonably attributable to the *MYC* down-regulation.

6) The PCA analysis is Figure 3 is in principle interesting and laudable as an attempt to quantify differences in 4C profiles in a quantitative and unbiased way. However, the text is somewhat obscure and panels 3C-H are difficult to interpret. It is unclear why the results shown in Figure 3E-H, where PCA is performed on a subset of the data, are so different from panel 3D. These differences are acknowledged in the main text but I did not understand how they are interpreted by the authors. I would actually suggest that the text relative to Figure 3 is entirely re-written and clarified (e.g. please explain what "component loading" means in this context). In addition, the PCA results should be integrated with a discussion of whether they correlate or not with the position and orientation of endogenous CTCF sites (see point 2 above).

Thanks a lot for the appreciation of our attempt and also for giving kind suggestions. We thoroughly revised our text and figures, as described in our response to essential revision 2. Regarding the relevance to the endogenous CTCF sites, please see our response to essential revision 6.

7) Transcriptional downregulation of MYC is attributed to changes in contact frequencies due to the presence of ectopic CTCF sequences at the STITCH cassette, which is supported by the strong correlation observed in Figure 3I. If this is really the case, and is due to CTCF looping from STITCH onto endogenous CTCF sites, then it should be possible to recapitulate the phenotype by deleting the endogenous partner CTCF sites. This would significantly strengthen the interpretation of the data.

Please see our response to essential revision 6.

8) in Figure 4C-D, negative and positive controls are missing (i.e. one or more regions where H3K4me3 should not be detected, and a region that is heavily bound by H3K27me, such as a poised gene, or a Hox gene). This is a very important control though, because one of the most interesting observations in the manuscript is that the transcriptional downregulation of MYC correlates with higher H3K27me3 levels. However, how much H3K27me3 is deposited? How does it compare with poised and/or inactive loci?

Thanks a lot for the comment and suggestion. We now have the suggested controls in the revised manuscript. Please see our response to essential revision 4.

9) In Figure 5, it is impossible to understand which changes are occurring at the MYC promoter in terms of H3K9me3 and CTCF levels. This is nonetheless crucial to interpret the gene expression changes upon Dox induction and how they are related to targeted recruitment of KRAB. It seems that the CTCF signal is decreased also in the MYC promoter in the absence of Dox, and not only at the STITCH region. A zoom-in and quantification of ChIP-seq experiments (peak calling, integrated intensities of signals) should be provided, and the results should be discussed accordingly.

Thanks for the comment. We now show a zoom-in view of these tracks, including *MYC* promoter, which clearly shows that H3K9me3 is not deposited around the promoter (Figure 5—figure supplement 1G). We also provide quantitative representations of CTCF ChIP-seq for this revision. The results show that striking changes of CTCF binding only took place at STITCH, but not in the other genomic regions, including those near the *MYC* promoter (Figure 5—figure supplement 1F).

10) Along the same line, in Figure 6 a crucial missing information is how CTCF binding evolves in time at the STITCH cassette and at the MYC locus.

We agree that this would be critical to understanding our STITCH system fully. Technically speaking, CTCF nChIP demands immediate processing and IP of chromatin after sampling, as freezing somehow abolishes most of the signals. Therefore, carrying out time-course experiments is quite difficult for our current situation. What we intend here is basically to describe the transition by the KRAB induction, which we think is well represented by our 4C-seq. We are, of course, very interested in what kind of events are serially involved in this process. However, we would like to leave this open for future works in the present study.

11) It is unclear what 'control' in Figure 7 refers to.

Thanks for the comment. As suggested by reviewer#1, we now indicate in the figure panels that they are either + or – DOX controls. They were kept in either + or – DOX for a much longer time than 24 or 72 hours (more than one passage) without switching.

12) One very interesting observation is that MYC gains H3K27me3 upon STITCH insertion, which correlates with the observed level of insulation in the various mutants and with transcriptional activation/deactivation in time course experiments. However why is it so? If this happens as a consequence of physical insulation from the super enhancer, how do the authors interpret it? An alternative explanation is that PRC2 is recruited by sequences in the STITCH cassette, and helps repressing transcription. The delay observed between MYC deactivation/reactivation and the corresponding differences of H3K27me3 are not large enough to exclude this second hypothesis. Based on the experiment shown in Figure 7J it cannot be excluded that H3K27me3 levels are unchanged upon treatment with EPZ, in the absence of a carefully quantified ChIP experiment.

Thanks for the comment. We have repeated our analysis presented in Figures 6F-G, 7A-F, and confirmed that our conclusion was reproduced, as described in our response to essential revision 4. Therefore, we think that our conclusions are valid. It has been already shown that H3K27me3 or PRC2 are not required to establish transcriptome in certain cellular contexts (Riising et al., 2014). This indicates that PRC2 is not necessarily required to repress gene expression. The delayed change of H3K27me3 was also reported before by a few studies (Hosogane et al., 2013). Our conclusions are consistent with these studies.

On the other hand, H3K27me3 levels seem to correlate with gene expression levels greatly not only at the *MYC* locus but also globally in the genome. We also wonder what this correlation would mean. Surely, as we discuss in the manuscript, PRC2 is critical for organismal development and homeostasis. Therefore, the protein complex should have essential roles. Perhaps the effects on transcriptome observed in PRC2 mutants so far might have been just an indirect effect of something more direct. However, we really cannot add speculations more. This needs to be studied in the future.

[Editors' note: further revisions were suggested prior to acceptance, as described below.]

Summary:This study describes development of a inducible insulator cassette, STITCH, that can act as boundary. This can be a very valuable tool to dissect rules of promoter – enhancer communication, and chromosome folding through mechanisms such as loop extrusion.We request that the authors address the following issues.Essential revisions:1) The authors were requested to put the structural changes induced by STITCH in the context of the overall CTCF binding patterns in the MYC region. They addressed this point on the one hand by performing new 4C experiments using the STITCH sequence as a viewpoint. These experiments, now in Figure 1, unfortunately do not seem to reveal how the various endogenous CTCF sites could be used to make new connections with the ectopic STITCH cassette and even a re-analysis of the 4C data with the MYC promoter as a viewpoint are inconclusive. The authors conclude vaguely that "It might be extrapolated from these previous results that there are not very specific endogenous regions that singly form loops with STITCH to organize the conformational changes induced by STITCH". The interpretation of the data in the rebuttal is also highly speculative and does really not address the reviewers' request that structural changes are evaluated in the light of a more global view of chromosome contacts such as the one provided by Hi-C data. In 4C it is always hard to detect loop extrusion-associated structural features such as loops and stripes (or flares), and it is not surprising that specific connections between CTCF sites might be missed without performing matched Hi-C or 5C experiments. The authors need to at least these limitations of their 4C-based analyses.

Thanks for the comment. We agree that discussing based on the “extrapolation” from other studies had been more speculative than based on actual experiments in the manuscript of the previous version. Hence, we deleted the corresponding paragraph in which we had discussed the issue in relation with the studies of the *Tfap2c* locus (Tsujimura et al., 2018) and the *Shh* locus (Williamson et al., 2019). Instead, we added the following sentence to the present manuscript (subsection “Titrating blocking activity of STITCH by serial mutations of the CTCF binding sites”).

“Also, more comprehensive analysis methods such as 5C or Hi-C are required to fully describe the locus-wide conformational change induced by STITCH.”

We also added the following sentence in Discussion section to emphasize the limitation of our study.

“Moreover, applying 5C or Hi-C might be more appropriate to describe formation of contact domains than the present 4C-based analyses.”

2) In the new Figure 8, new experiments are provided to support the applicability of STITCH to additional contexts. However, the data do not appear to fully support the conclusion that STITCH-mediated modifications of chromosome interactions are at the basis of the observed transcriptional effects. First, 4C experiments in panel F do not allow to conclude in any manner that STITCH alters conformation at the targeted allele. Certainly, the presence of the wild-type allele confounds the readout, but even considering this, the fluctuating small-% differences observed do not seem to be robust (also, information on replicates is not provided unless I am mistaken). The right experiment to address this point would have been to perform 4C from the STITCH cassette itself, which would allow to detect mainly contacts within the mutant allele and which is apparently technically possible given that similar experiments are reported in the new version of Figure 1. It would be important to add such 4C analysis.

Thanks for the suggestion. Accordingly, we newly performed 4C-seq from a viewpoint at STITCH inserted in the *NEUROG2* locus and compared the contact pattern between in the presence and absence of DOX (Figure 8—figure supplement 2D-G). The data well show that while the contact considerably extended to distant regions in the presence of DOX (i.e. with CTCF binding; see Figure 8E), the contact attenuated relatively in a short distance in the absence of DOX (Figure 8—figure supplement 2E, F). We think this change should reflect the extrusion mediated contact of the CTCF binding sites at STITCH. Thus, the new experiments could well support that the chromatin conformation is altered by the functionality of STITCH.

The viewpoint was designed at the right edge of the inserted cassette (Figure 8—figure supplement 2D). So, we also compared the directionality of chromatin folding. However, we could not detect striking changes by DOX (Figure 8—figure supplement 2E, G). We think this is because the viewpoint captures the contact pattern of not only the rightward CTCF binding sites but also the leftward ones, which are only 600-bp apart from the viewpoint. In the above analysis, indeed, the contact extension was observed on both right- and left-sides, indicating that the viewpoint captures contacts of both rightward and leftward CTCF arrays (Figure 8—figure supplement 2E, F).

We agree that the difference of the contact frequencies depicted in Figure 8F appears to be small. However, we would like to emphasize that the PCA plot in Figure 8—figure supplement 2C unbiasedly shows that the DOX alters the contact pattern exactly at the STITCH insertion site. Therefore, we think that it should be safe to attribute the conformational change to the functionality of STITCH.

Second, it appears that NEUROG2 downregulation is only shown for one population of cells following piggyBac-mediated insertion of the TetR-KRAB transgene. It is unclear whether these results would hold true if the transposition of the transgene is repeated in independent experiments. Given that piggyBac insertions typically occur in multiple genomic locations simultaneously in every cell, a possibility that cannot be excluded is that the transcriptional effect on NEUROG2 is a secondary effect of mis-regulation of one or more upstream genes that are accidentally targeted by the transposon. This should be discussed.

Thanks for the comment. Actually, we have split the cells in the same one dish, which had been transfected with the TetR-KRAB transgene in the piggyBac vector, equivalently to all the samples and replicates. Therefore, the composition/heterogeneity of the transgene insertions should be controlled well and equivalent between with and without DOX conditions. So, the possible secondary effects should also be equivalent. To clarify the procedures, we described the experiment as following (subsection “Blocking *NEUROG2* activation in differentiating neural progenitor cells with STITCH”).

“We split the NEUROG2/KRAB cells derived from a single dish equivalently to different dishes, and then either did or did not add DOX upon the start of the differentiation into NPCs.”

3) Introduction and Discussion section: the authors seem to confuse several concepts of loop extrusion, insulation and contact domains. Insulation by CTCF is the result of its ability to block loop extrusion. Insulation does not necessarily involve CTCF sites to loop with each other (in fact such loops barely insulate).

Thanks for the comment. This is actually how we think. We simplified the Introduction to prevent potential confusion.

Insulation and contact domains can occur even when interactions are not strictly divergent at boundaries: insulation can occur in only one direction when CTCF sites are all in the same direction.

We agree. Our data are also very consistent to this statement. Moreover, we believe that the direct comparison of different CTCF configurations in our serial mutagenesis experiments could contribute to this understanding.

Such unidirectional insulation can demarcate contact domains.

We agree. This is for example described as exclusion domains by Sanborn et al., 2015. As far as we understand, however, emergence of such contact domains is not conclusively shown to be a prerequisite for the specification of enhancer allocation.

The authors are asked to consider these issues when revising the Introduction and Discussion section.

Thanks for the suggestions. Based on the above comments, we have tried to improve the Introduction and Discussion section.

4) All reviewers found the manuscript extremely difficult to read. Please do not use track changes. The manuscript should be carefully edited.

We have re-edited the manuscript carefully. This time, we upload the manuscript without the track changes.

5) All plasmids should be made available through AddGene.

Now all the three plasmids used in this study are already available through AddGene.